# Rapid neurogenesis through transcriptional activation in human stem cells

Volker Busskamp[1,2,†,‡], Nathan E Lewis[1,2,3,4,†], Patrick Guye[5,†], Alex HM Ng[1,2,6], Seth L Shipman[1,2], Susan M Byrne[1,2], Neville E Sanjana[7,8], Jernej Murn[9,10], Yinqing Li[5], Shangzhong Li[11], Michael Stadler[12,13,14], Ron Weiss[5] & George M Church[1,2,*]

## Abstract

Advances in cellular reprogramming and stem cell differentiation now enable *ex vivo* studies of human neuronal differentiation. However, it remains challenging to elucidate the underlying regulatory programs because differentiation protocols are laborious and often result in low neuron yields. Here, we overexpressed two Neurogenin transcription factors in human-induced pluripotent stem cells and obtained neurons with bipolar morphology in 4 days, at greater than 90% purity. The high purity enabled mRNA and microRNA expression profiling during neurogenesis, thus revealing the genetic programs involved in the rapid transition from stem cell to neuron. The resulting cells exhibited transcriptional, morphological and functional signatures of differentiated neurons, with greatest transcriptional similarity to prenatal human brain samples. Our analysis revealed a network of key transcription factors and microRNAs that promoted loss of pluripotency and rapid neurogenesis via progenitor states. Perturbations of key transcription factors affected homogeneity and phenotypic properties of the resulting neurons, suggesting that a systems-level view of the molecular biology of differentiation may guide subsequent manipulation of human stem cells to rapidly obtain diverse neuronal types.

**Keywords** gene regulatory networks; microRNAs; neurogenesis; stem cell differentiation; transcriptomics
**Subject Categories** Methods & Resources; Stem Cells; Neuroscience

**Mol Syst Biol.** (2014) 10: 760

## Introduction

To cope with the vast complexity of the human brain with its billions of cells and trillions of synapses (Herculano-Houzel, 2009; Rockland, 2002), research efforts usually take deconstructive approaches by focusing on individual brain regions of model organisms. Ethical constraints limit the breadth of feasible research on primary human brain tissues from healthy, living subjects, and the availability of high-quality *post-mortem* tissues is limited. Thus, it is desirable to develop *in vitro* systems that mimic properties of the human brain. Advances in stem cell differentiation and transdifferentiation of somatic cells into neurons now allow the use of complementary constructive tactics to understand human brain functions (Amamoto & Arlotta, 2014). This can be done *in vitro* by generating neurons and by finding ways to connect and mature them into functional neuronal circuits. However, the lack of fast and efficient protocols to generate neurons remains a bottleneck in neuronal circuit fabrication. Moreover, successful generation of particular neuronal subtypes may also enable therapeutic cell replacement strategies for neurological disorders (Barker, 2012; Lescaudron *et al*, 2012).

Both human embryonic (ES) and human-induced pluripotent stem cells (iPS) have been successfully used to generate neurons.

1 Department of Genetics, Harvard Medical School, Boston, MA, USA
2 Wyss Institute for Biologically Inspired Engineering at Harvard University, Boston, MA, USA
3 Department of Biology, Brigham Young University, Provo, UT, USA
4 Department of Pediatrics, University of California, San Diego, CA, USA
5 Department of Biological Engineering, Massachusetts Institute of Technology, Cambridge, MA, USA
6 Department of Systems Biology, Harvard Medical School, Boston, MA, USA
7 Broad Institute of MIT and Harvard, Cambridge Center, Cambridge, MA, USA
8 McGovern Institute for Brain Research, Department of Brain and Cognitive Sciences, Department of Biological Engineering, Massachusetts Institute of Technology (MIT), Cambridge, MA, USA
9 Department of Cell Biology, Harvard Medical School, Boston, MA, USA
10 Division of Newborn Medicine, Boston Children's Hospital, Boston, MA, USA
11 Department of Bioengineering, University of California, San Diego, CA, USA
12 Friedrich Miescher Institute for Biomedical Research, Basel, Switzerland
13 Swiss Institute of Bioinformatics, Basel, Switzerland
14 University of Basel, Basel, Switzerland
*Corresponding author. Tel: +1 617 432 1278; E-mail: gchurch@genetics.med.harvard.edu
†These authors contributed equally to this work
‡Present address: Center for Regenerative Therapies Dresden (CRTD), Dresden, Germany

*In vivo*, neuronal differentiation is a complex process involving many transcription factors and regulatory cascades (He & Rosenfeld, 1991). Through the process, cells pass via progenitor cell states (Molnar & Clowry, 2012) prior to becoming neurons. Standard neuronal differentiation protocols try to mimic developmental stages by applying stepwise environmental perturbations to cells, pushing them from one state to the next. However, these differentiation protocols have been suboptimal, with multiple steps, including the application of different soluble bioactive factors to the culturing media, ultimately requiring months to complete. In addition, these protocols often suffer from high variability and relatively low yields of desired neurons (summarized by (Zhang *et al*, 2013)).

Another approach has been taken to derive neurons *in vitro* by transdifferentiating human fibroblasts with cocktails of neural transcription factors and/or microRNAs (miRNAs), yielding induced neurons (Vierbuchen & Wernig, 2012). Fibroblast-derived induced neurons are generally considered safer for transplantation because they eliminate the chance of having non-differentiated stem cells form tumors following transplantation (Vierbuchen & Wernig, 2011). However, these approaches start with slow-growing fibroblasts and suffer from low yields of induced neurons. Moreover, in transdifferentiation experiments, the neuronal differentiation process is direct; natural proliferative neuronal progenitor stages that occur during neuronal development are skipped (Liu *et al*, 2013). Culture time and neuronal yields were recently improved by induced transcription factor expression in human stem cells with a new protocol that achieved highly pure neurons from human stem cells via a selection system over 2 weeks (Zhang *et al*, 2013). This differentiation route is thought to have many similarities with transdifferentiation, although those have not been assessed directly.

To date, combinations of transcription factors and miRNAs used in differentiation protocols have been selected based on their involvement in brain development, assuming that they would function similarly in stem cells. Although resulting neurons are characterized extensively after differentiation at their endpoints, the underlying gene regulatory pathways during their differentiation are mostly unknown. Recent work in stem cell-derived neurons shed some light on potential transcriptional regulators activating various neuronal differentiation programs (Gohlke *et al*, 2008; Mazzoni *et al*, 2013; Stein *et al*, 2014; van de Leemput *et al*, 2014; Velkey & O'Shea, 2013; Wapinski *et al*, 2013), and other studies have identified key miRNA regulators in neuronal differentiation *in vivo* and *in vitro* (Akerblom *et al*, 2012; Le *et al*, 2009; Yoo *et al*, 2011).

However, we have little knowledge on the underlying gene regulatory mechanisms in stem cell-derived neurogenesis because of the aforementioned long time lines and heterogeneous neuronal populations. A coherent understanding of potential gene regulatory mechanisms would allow targeted interventions to guide, fine-tune and accelerate the differentiation processes towards neurons of interest.

To simplify neuronal differentiation protocols and facilitate the elucidation of gene regulatory mechanisms underlying stem cell-derived neurons, we present a novel rapid and robust differentiation protocol that yields highly homogeneous neurons. Neuronal differentiation in this protocol is triggered by overexpression of a pair of transcription factors (Neurogenin-1 and Neurogenin-2) in human iPS cells and results in a homogeneous population of functional bipolar neurons within 4 days. We performed RNA sequencing and quantitative miRNA profiling over the time course of differentiation to reveal regulators contributing to the rapid neurogenesis. Our results indicated that Neurogenin-mediated neurogenesis proceeds indirectly via unstable progenitor states. We elucidated a network of key transcription factors and miRNAs that contributed to differentiation. By perturbing individual members and combinations thereof, we demonstrated that while the differentiation was robust, perturbations to the network induce significant variations in resulting cell morphology.

## Results

### Neurogenin induction drives iPS cells rapidly and homogeneously to bipolar neurons

Transcription factors of the Neurogenin family are important for neuronal development *in vivo* (Morrison, 2001), and individual Neurogenins have been used previously with some success to induce neuronal differentiation from mouse cancer and ES cells (Farah *et al*, 2000; Reyes *et al*, 2008; Thoma *et al*, 2012; Velkey & O'Shea, 2013), to differentiate neurons from multipotent human neural progenitor cells (Serre *et al*, 2012), and to transdifferentiate human fibroblasts (Ladewig *et al*, 2012) and stem cells (Zhang *et al*, 2013). Furthermore, when Neurogenin-2 was induced in human stem cells and followed by glia cell co-cultures, stepwise application of bioactive factors and the usage of a selection system, high yields of neurons were achieved in only 2 weeks (Zhang *et al*, 2013). Since both Neurogenins alone can drive stem cells into neuronal lineages,

---

**Figure 1.   Rapid neuronal differentiation by induced Neurogenin overexpression in human iPS cells.**

A    General scheme of Neurogenin 1+2 induction to yield differentiated neurons from human iPS cells after 4 days.
B    Proportion of uninduced (white) and 4 days induced (black) iNGN cells analyzed by flow cytometry for the pluripotency marker Tra-1/60, demonstrating a nearly complete differentiation of iPS cells.
C    Representative transmission light microscopy image of a bipolar-shaped iNGN cell at day 4 of differentiation.
D    Quantification of bipolar-cell-shaped morphology on day 4, 78 cells analyzed in total.
E    Immunostaining for MAP2 and nuclear DAPI staining of neurons induced for 4 days (upper row) and uninduced iPS cells (lower row).
F    Quantification of MAP2-expressing cells. *n* refers to the number of cells from three independent experiments as in (E).
G    Immunostaining for SYN1 of neurons induced for 4 days (upper row) and uninduced iPS cells (lower row).
H    Quantification of SYN1-expressing cells. *n* refers to the number of cells from three independent experiments performed as in (G).
I, J   Characterization of action potentials across 10 cells recorded at 4 days (I) or 14 days (J) postinduction. Traces show response to a 20 pA injected current over 0.5 s. Inset shows a representative action potential waveform (in red) with corresponding dV/dt trace (in gray), highlighting threshold and width parameters. Left scale bar: 50 ms/20 mV. Inset scale bar gray: 5 ms/25 mV/ms, red: 25 mV.
K    Percentage spiking and non-spiking cells at 4 days and 14 days postinduction.

Data information: Scale bars (C, E, G), 20 μm. Two-sample Student's *t*-test, ***$P$-value $\leq$ 0.001. Error bars, $\pm$ SEM.

and since they are co-expressed in some neuronal progenitor cells *in vivo* (Britz *et al*, 2006), we wondered if there were beneficial effects on differentiation speed and yield from overexpressing Neurogenin-1 and Neurogenin-2 together (hereafter referred to together as Neurogenins, see also Supplementary Text). Therefore, we developed a bicistronic doxycycline-inducible Neurogenin expression cassette to trigger neurogenesis in human iPS cells (Fig 1A; Supplementary Fig S1). We used lentiviral gene delivery to introduce the inducible Neurogenin expression cassette into human

PGP1 iPS cells (Lee *et al*, 2009) leading to a stable and small molecule-inducible Neurogenin iPS line, hereafter referred to as iNGN cells. Notably, the differentiation occurred in defined stem cell media in the absence of additional selection markers or neurotrophic factors, and differentiation was successful in additional stem cell lines we tested (Supplementary Fig S1G and H).

Neurogenin protein expression in iNGN cells occurred in a doxycycline-dependent manner, and its activation triggered rapid differentiation of stem cells (Supplementary Fig S1), as demonstrated

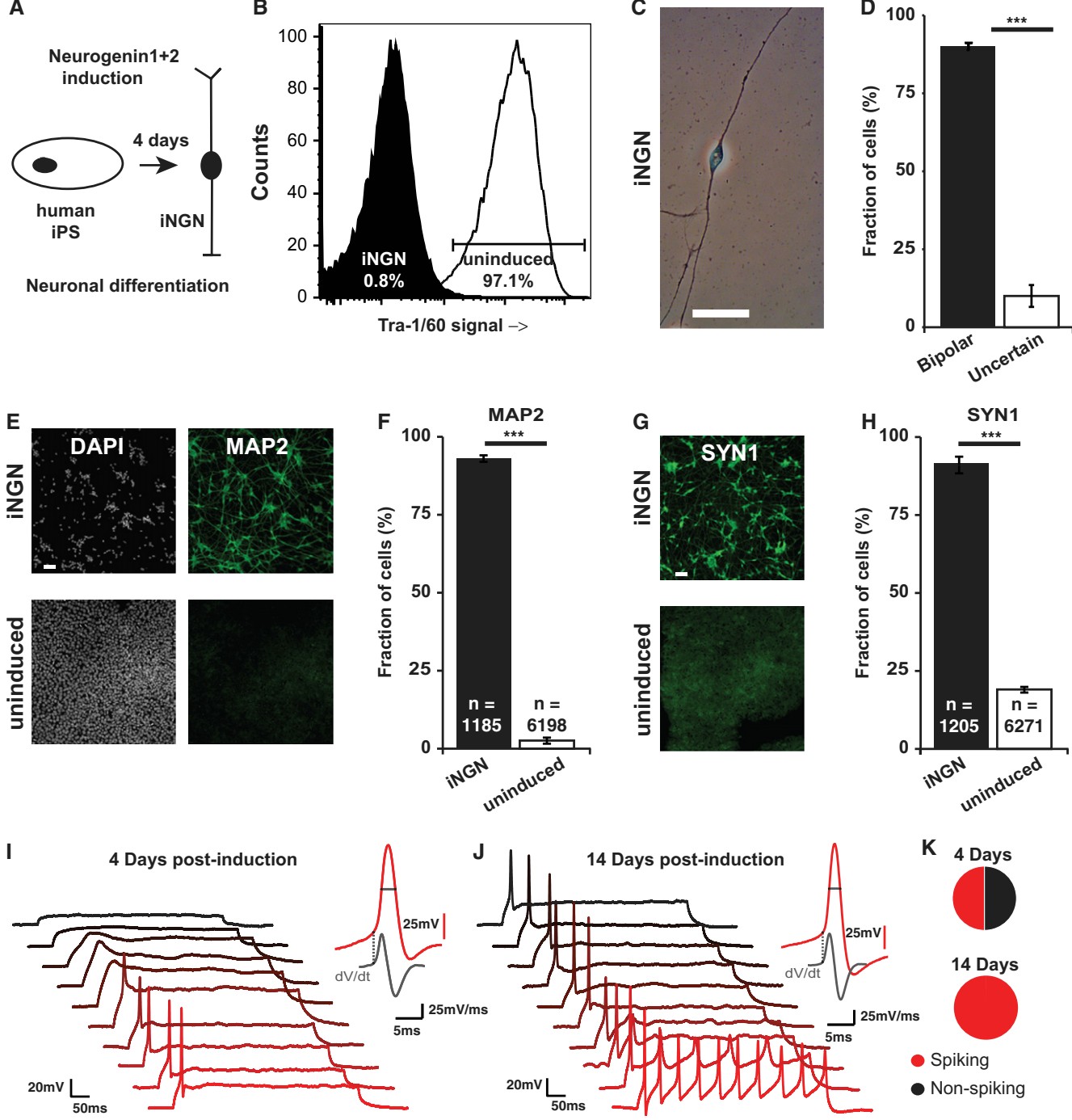

Figure 1.

by the loss of the pluripotency marker Tra-1/60. In uninduced cells, 97.1% of the cells were tested positive for this marker compared to 0.8% iNGN cells at 4 days postinduction (Fig 1B). The efficiency of neuronal conversion at day 4 was high and homogeneous; about 90 ± 4% of the induced iNGN cells had a bipolar-shaped morphology with long neurite projections on opposing sites (Fig 1C and D; Supplementary Fig S1 and Supplementary Video S1). On day 4 of induction, more than 90% of the cells stained positively for microtubule-associated protein 2 (MAP2) and Synapsin 1 (SYN1), consistent with the acquisition of neuronal identity (Fig 1E–H). These cells were also immuno-positive for several additional neural markers (Supplementary Fig S2). After induction, the proliferation rate decreased considerably and iNGN cells became postmitotic, as is common for differentiated neurons (Bhardwaj et al, 2006) (Supplementary Figs S1 and S2, Supplementary Video S1).

Next, we functionally characterized the induced neurons by whole-cell patch-clamp electrophysiology. If maintained in stem cell media, the neurons failed to fire action potentials on day 4 (Supplementary Fig S2). However, cells would fire single action potentials upon current injection when the cells were co-cultured with astrocytes (see Materials and Methods). By day 14, iNGN cells were able to fire trains of action potentials (Fig 1I and J). The number of electrically excitable cells increased from 50% at day 4 to 100% after two weeks of induction. At later time points, we detected occasional spontaneous postsynaptic currents indicating functional synaptic activity (Supplementary Fig S2). Taken together, Neurogenin induction drove human iPS cells rapidly to differentiated neurons that were competent to achieve functional maturation.

## iNGN gene expression profiles are consistent with neuronal transcription

To understand the molecular events occurring during rapid iNGN neurogenesis, we aimed to capture the transcriptomic changes over the time course of neuronal differentiation. Previous differentiation protocols have not permitted the acquisition of high-resolution temporal transcriptomic analysis of neurogenesis from human iPS cells, due to the highly heterogeneous cell populations. Our iNGN cells, on the other hand, demonstrate morphological and immuno-histochemical homogeneity (Fig 1; Supplementary Figs S1 and S2). Therefore, iNGN cells are well suited to reveal transcriptional changes during neurogenesis when analyzed in cell cohorts. We conducted RNA sequencing (RNA-Seq) experiments of iNGN cells with biological triplicates at four time points (day 0, 1, 3 and 4) (Supplementary Fig S3 and Supplementary Table S1). Cells on day 2

were morphologically similar to day 1 induced cells and were therefore not assayed (Supplementary Fig S1).

During differentiation, thousands of genes were differentially expressed ($q < 0.05$, > 1.5-fold change). Consistent with our macroscopic findings, mRNA abundance decreased for most canonical stem cell factors. For example, the stem cell markers NANOG and POU5F1 (OCT4) decreased 58- and 39-fold, respectively (Fig 2A). In line with neuronal cell fate commitment, most neural marker transcripts were significantly upregulated by day 4, including MAP2 (30-fold) and SYN1 (3.7-fold) (Fig 2B). Also, as expected, the neural repressor RE1-silencing transcription factor (REST) decreased 27-fold. In addition, many neuronal transcription factors previously used for transdifferentiation experiments (Vierbuchen & Wernig, 2012) were also upregulated more than 50-fold (Fig 2C). Thus, transcription factors that are currently used for forced neuronal induction are activated downstream of the Neurogenins. Consistent with the transcriptomic changes, we also witnessed differential protein expression, as shown by corresponding immunostainings (Fig 1E and G; Supplementary Figs S1 and S2).

In addition to the expression of proneural transcription factors, we found a rapid upregulation of transcripts that encode key neuronal components. Specifically, we found the upregulation of synaptic machinery components (Fig 2D and E) as well as those of the axon initial segment (Supplementary Fig S4), where action potentials are generated. Notably, at the presynaptic side, transcripts associated with the synthesis and secretion of the neurotransmitters glutamate and acetylcholine were upregulated. We tested protein expression at a single-cell level to see whether iNGN cells represented a mix of different neuron types or a homogeneous culture of cells showing co-transmission of glutamate and acetylcholine, which is thought to be rare, but has been previously reported in vivo (Guzman et al, 2011). When iNGN cells were subjected to immunostaining for VGLUT1 and ChAT (Fig 2F), 100% of the neurons tested positive for vGLUT1 and 98% for ChAT (Fig 2G), and stainings were co-localized (Fig 2H), suggesting that iNGN cells might be co-releasing glutamate and acetylcholine. Thus, together with the aforementioned analyses, the iNGN cells consist of a homogenous population and could express many major neuronal components within 4 days of Neurogenin induction.

### iNGN differentiation resembles in vivo processes

While differentiating, iNGN cells underwent a dramatic change in morphology (Supplementary Fig S1 and Supplementary Video S1). They first dissociated from stem cell colonies and until day 2

**Figure 2. Rapid transcriptional induction of neural markers including the synaptic machinery in iNGN cells.**

A–C   Gene expression levels of (A) stem cell markers, (B) neural markers and (C) transcription factors previously used for transdifferentiation experiments, as measured by RNA-Seq. Error bars represent the 95% confidence interval of mRNA abundance.

D, E   There was a rapid transcriptional induction of the synaptic machinery in iNGN cells over the time course of differentiation. Heatmaps represent the Z-score for expression levels for all isoforms over the four time points assayed. A schematic outline of presynaptic terminal components (D) shows a general trend of upregulation during iNGN development. Similarly, postsynaptic components and their contributions to neuronal function are shown in (E). Cellular processes are color-coded and indicated in the figure.

F   Immunostainings for vGLUT1 and ChAT of iNGN cells at day 4.

G   Quantification of vGLUT1- and ChAT-positive iNGN cells (in triplicates); error bar, SEM.

H   The signals co-localize, indicating the presence of a homogeneous neuronal population with abilities to co-transmit glutamate and acetylcholine.

Data information: Scale bars, 20 μm. FPKM, fragments per kilobase of transcript per million mapped reads.

**Figure 2.**

expanded and retracted small processes, while occasionally dividing. On day 3, larger processes emerged, finally resulting in neurons with bipolar morphology by day 4. These dynamic morphological changes showed similarities to *in vivo* differentiation steps, so we wondered whether iNGN differentiation represented a direct conversion from the stem cell lineage toward neuronal cell fate or whether the iNGN cells differentiate more 'naturally' via progenitor stages.

Thus, to obtain a global and unbiased view of which biological processes significantly changed between days 0 and 4 (Fig 3A; Supplementary Tables S2 and S8), we performed a Gene Ontology (GO) terminology analysis (Ashburner *et al*, 2000). By day 4, genes annotated as relevant for cell cycle and nucleic acid metabolism were significantly downregulated. On the other hand, GO classes relevant to neuronal differentiation, physiology and neuronal cell adhesion were significantly enriched in upregulated genes, showing that iNGN cells broadly express the necessary genes for neuronal fate commitment and the assembly of neuronal compartments such as synapses and axons. In accordance with our functional data (Fig 1; Supplementary Fig S2), the process of 'synapse assembly' was still ongoing as indicated by increasing gene expression at day 4 (Supplementary Fig S4).

Genes expressed in neuronal progenitors and classified by the GO terminology as 'regulation of neurogenesis' were highly activated (inset in Fig 3A), except for two repressors of neural genes: HES3 and HES1. The expression of NOTCH1, its ligands DLL4 and DLL1 and the NOTCH target HES5 followed a pulsed expression pattern, an initial increase followed by reduced expression by day 4. In addition, similar activation was seen for members of the 'cell fate determination' GO class (Supplementary Fig S4), suggesting that iNGN cells traversed some typical neuronal progenitor states. Additionally, many neuronal progenitor markers, such as FABP7 and NTN1, were initially upregulated and subsequently downregulated (Supplementary Fig S4), suggesting the presence of a transient progenitor identity.

Taken together, during the rapid differentiation, iNGN cells differentiated indirectly and exhibited a brief signature of neuronal progenitor cells. Hence, these cells likely take differentiation routes similar to the ones found *in vivo*.

## iNGN gene expression shows similarities to the developing human brain

Both functional and transcriptomic analyses point to a neuronal trajectory that mirrors typical developmental steps. Therefore, we investigated whether the cells resulting from Neurogenin induction exhibit similarities to neurons in the human brain. Stem cell-derived and induced neurons are generally categorized based on morphology, electrical properties and a handful of transcripts and immuno-markers gained from animal models as references. For example, bipolar neurons are found in the retina (Masland, 2001) and spinal ganglia (Matsuda *et al*, 1996). Given our wealth of transcriptomic information on these cells, we sought to refine this definition by comparing our RNA-Seq data with the BrainSpan Atlas dataset from the Allen Institute for Brain Science (Miller *et al*, 2014) (http://brainspan.org/). This dataset covers RNA-Seq data for mixed cell types of 16 cortical and subcortical structures across the full course of human brain development. This dataset lacks single-cell resolution, but it comprises the most comprehensive temporal and spatial human brain reference thus far, allowing brain mapping of *in vitro* derived neurons (Stein *et al*, 2014; van de Leemput *et al*, 2014). The transcriptomic profile of iNGN cells, 4 days postinduction correlated best with human fetal brain 12 to 26 weeks postconception (Pearson coefficient > 0.7). The correlation of the induced cells was significantly higher than seen in the uninduced iNGN cells (day 0) (Fig 3B). Furthermore, we found our cells had higher correlations with the mediodorsal nucleus of thalamus, amygdaloid complex, hippocampus and the cerebellar cortex compared to the cortical areas (Fig 3C). These higher correlations likely do not result from having higher neuronal content in the brain regions (see Supplementary Text). Thus, despite the heterogeneous composition of the BrainSpan reference samples, iNGN gene expression shows increased similarity to expression signatures of human brain tissue as compared with uninduced cells.

## miRNA profile changes support neuronal fate induction and the loss of pluripotency

Several recent studies have implicated various miRNAs as key regulators in neuron differentiation; therefore, we examined the role of miRNAs as regulators in iNGN cell differentiation by quantitatively profiling 654 different miRNAs from the same samples used for RNA-Seq (Supplementary Tables S3 and S4). Using Nanostring's nCounter technology to count individual miRNA molecules, we found that 116 and 155 miRNAs were detected above background levels at day 0 and day 4, respectively. At day 0, the uninduced iNGN samples had miRNA signatures of stem cells; the miR-302/367 cluster dominated their profile (50.3% of the total amount of miRNAs) (Fig 4A; Supplementary Fig S5) consistent with previous studies that demonstrated its role in regulating self-renewal and preserving pluripotency (Lipchina *et al*, 2012; Wang *et al*, 2013).

**Figure 3. Global neuronal cell fate commitment and spatio-temporal cell mapping of iNGN cells.**

A   Gene Ontology (GO) annotation was used to identify gene classes containing an overrepresentation of genes that were differentially expressed between day 0 and day 4 in iNGN cells. The majority of GO terms with an overrepresentation of downregulated genes (green) are related to cell cycle and nucleic acid metabolism, while GO terms with many upregulated genes (purple) include classes related to neuron development and physiology. For example, most genes in the Gene Ontology classification of 'regulation of neurogenesis' (inset) including neural progenitor markers are significantly upregulated as shown in the heatmap.

B   RNA-Seq data of uninduced iNGN (day 0, blue) and iNGN cells (day 4, black) are compared to the Allen BrainSpan data, by computing the Pearson's correlation coefficients between the iNGN cells and all brain samples at each developmental time point. This shows that iNGN gene expression is more consistent with prenatal brain gene expression, and the correlation is significantly higher for day 4 iNGN cells, compared to day 0 cells (one-sided two-sample *t*-test, with *P*-values shown in red dots).

C   Pearson's correlation coefficients were subsequently computed between day 4 iNGN cells and profiles for brain regions at each time point. The 500 most highly upregulated genes in day 4 iNGN cells were used as a neuronal signature. At each time point, Z-scores were computed for each brain region to assess their relative similarity to the iNGN signature, in comparison with the remaining brain regions. This demonstrated less similarity of iNGN cells to cortical brain structures and indicated higher similarities to the mediodorsal nucleus of thalamus, amygdaloid complex, hippocampus and cerebellar cortex. Pcw, postconception weeks.

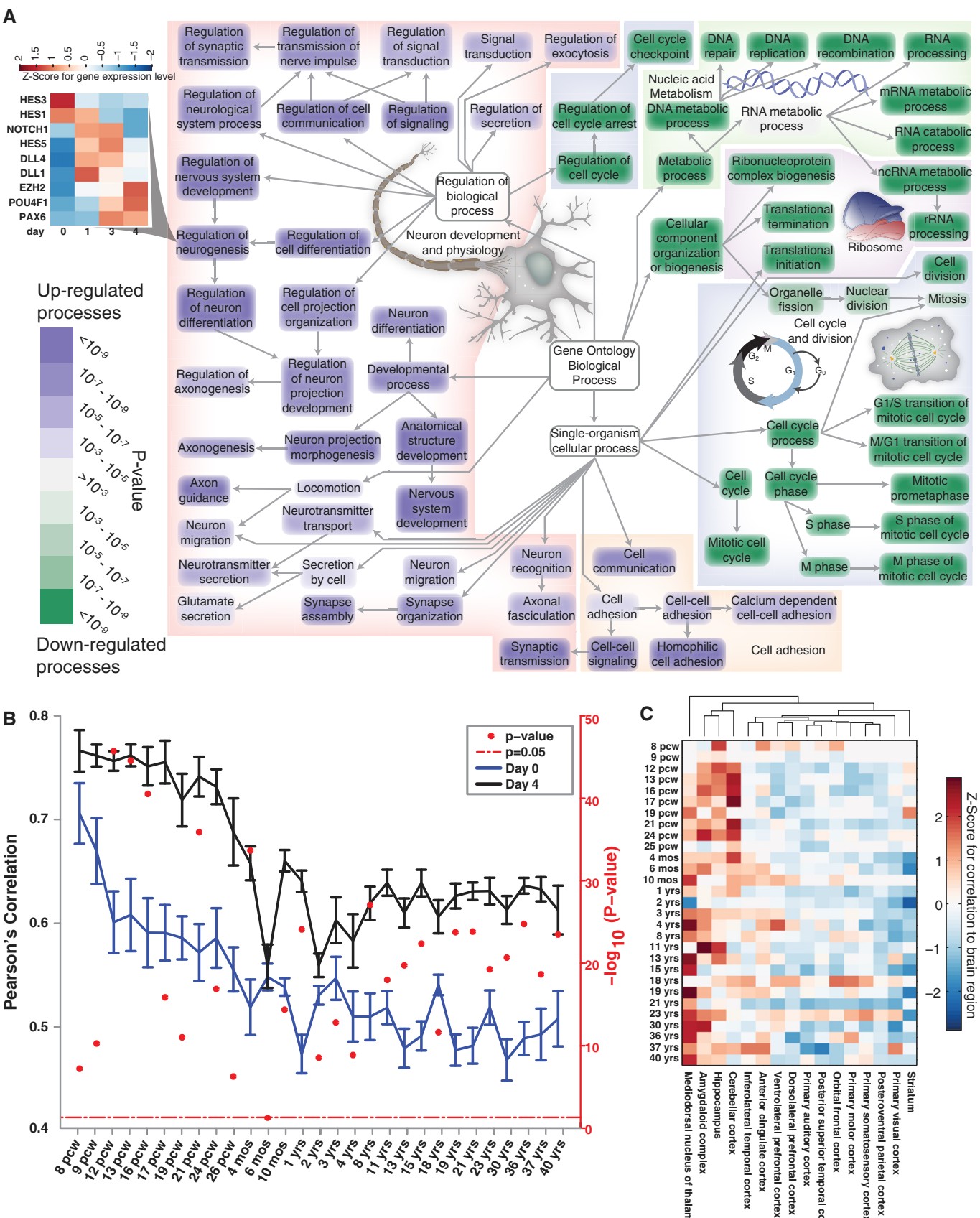

**Figure 3.**

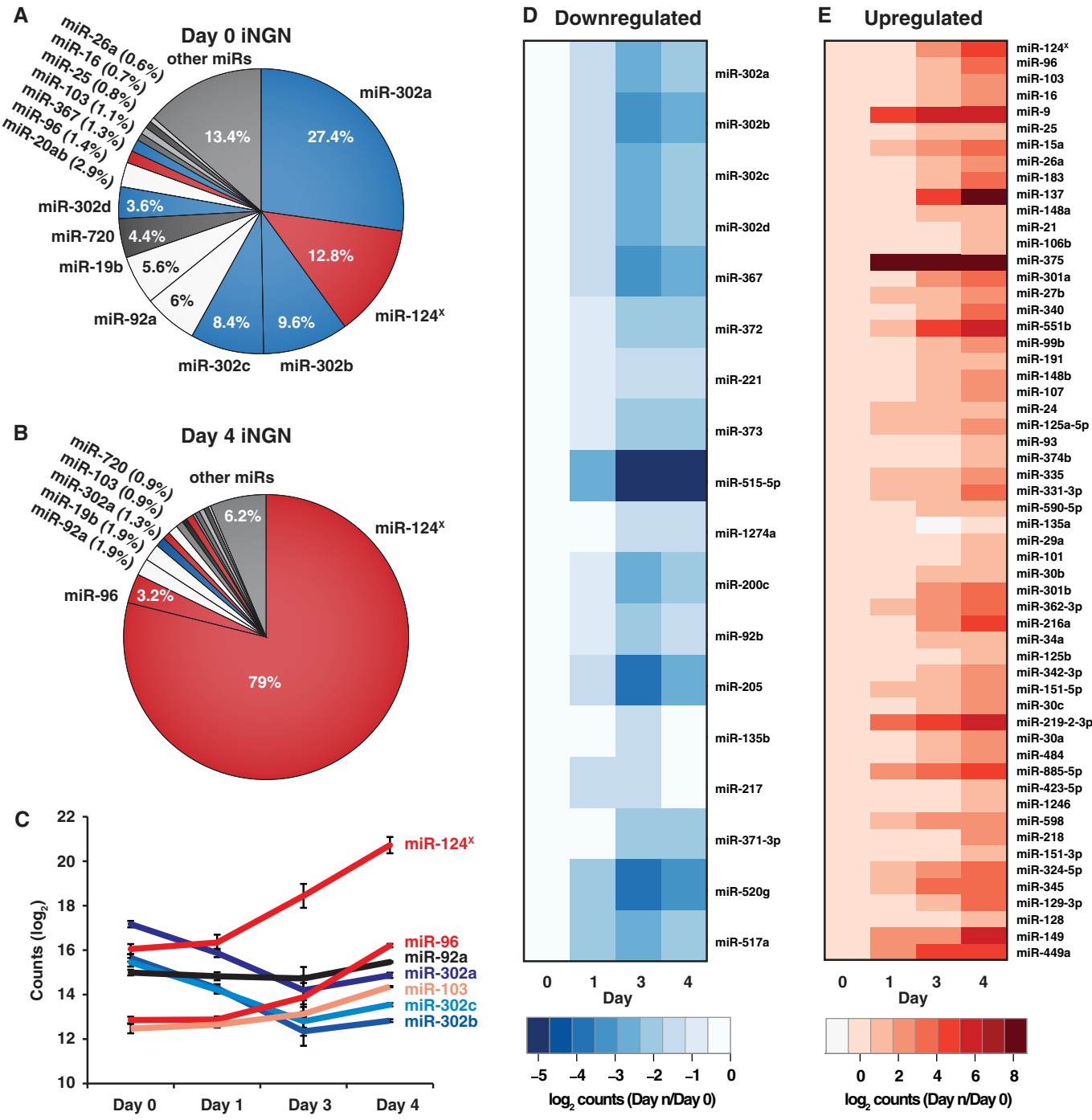

**Figure 4. Dynamic miRNA changes during iNGN differentiation.**

A, B    Relative abundance of microRNAs in (A) uninduced and (B) 4-days differentiated iNGN cells. The miRNAs associated with stem cell fate are indicated in blue, while the neuronal miRNAs are in red.

C    Dynamic miRNA changes of representative miRNAs during the differentiation of iNGN cells. Error bars, SEM. miR-124[x] refers to estimated counts from qRT–PCR.

D, E    Heatmaps of all significantly (D) down- and (E) upregulated miRNAs within the first 4 days of iNGN differentiation, rank-ordered by expression levels (ANOVA $q$-values < 0.05).

This cluster is transcriptionally regulated by NANOG, POU5F1 and SOX2 (Barroso-del Jesus *et al*, 2009), and since these pluripotency factors were downregulated, the decrease in the miR-302/367 cluster levels was expected.

We also measured miR-124, a brain-enriched miRNA (Akerblom & Jakobsson, 2013), by qRT–PCR (Supplementary Fig S5) and normalized its expression levels to nCounter results (see Materials and Methods). This miRNA is known to be important for neuronal

    

differentiation, since inhibition of miR-124 *in vivo* blocked adult neurogenesis in the mouse subventricular zone and its overexpression depleted the neural stem cell pool (Akerblom *et al*, 2012). Knockout experiments of miR-124 in mice resulted in brain abnormalities and increased apoptosis in retinal neurons (Sanuki *et al*, 2011). In our cells, miR-124 accounted for 12.8% of total miRNAs at day 0 and increased to 79% by day 4. We also observed increases in the abundance of the neuronal miR-96 (10-fold) and miR-9 (57-fold) (Fig 4B and E; Supplementary Fig S5) among others (Fig 4C). In total, by day 4, the levels of 18 miRNAs were significantly decreased in expression ($q < 0.05$) and 55 miRNAs were significantly upregulated ($q < 0.05$) (Fig 4D and E, and Supplementary Fig S5). Thus, miRNA profiles rapidly changed in the course of iNGN differentiation, consistent with the loss of pluripotency (miR-302 cluster) and the establishment of neuronal miRNA signatures (miR-124, miR-96 and miR-9).

To further identify particular miRNA contributions, we used a probabilistic modeling approach to detect dynamic regulatory networks consisting of miRNAs and transcription factors (Schulz *et al*, 2013). Cross-correlating our RNA-Seq and miRNA data over time by this probabilistic modeling method revealed additional groups of dynamically changing miRNA molecules that were likely aiding in gene expression regulation during iNGN differentiation at each measured time point (Supplementary Fig S6).

## A network of transcription factors drives the rapid neurogenesis

Homogeneous bipolar neuron cultures are achieved following Neurogenin induction, but the robust regulatory network underlying the response is not known. The GO terminology and BrainSpan analyses indicated similarities with 'natural' differentiation processes, but it is not clear which transcription factors were key players in the regulatory network driving iNGN differentiation. Thus, we analyzed the time course of mRNA expression data in the context of known transcription factor interactions in the Ingenuity Pathways Analysis (IPA) database (see Materials and Methods). To identify potential regulators, an enrichment test (Kramer *et al*, 2014) was conducted to identify transcription factors that had an overrepresentation of differentially expressed targets and had their targets changing expression in the direction consistent with the activation and repression activities of the transcription factors of interest (Supplementary Table S5). We focused here on a network of transcription factors that met these criteria and that were also connected to the Neurogenins through direct and indirect gene regulatory interactions that had been validated in other cell types and/or organisms, as catalogued in the IPA database.

Our analysis revealed a suppression of key stem cell factors by day 1. Regulatory targets of the stem cell factors POU5F1 (OCT4), NANOG and SOX2 were significantly differentially expressed ($P < 7.2 \times 10^{-4}$), consistent with the inhibition of their regulatory activities (Fig 5A). Our analysis further revealed several direct and indirect interactions through which Neurogenins likely repressed the stem cell factors (Fig 5A). Specifically, our analysis suggested that the Neurogenins inhibit SOX2, which leads to the inhibition of NANOG and POU5F1. Additional indirect interactions could further repress stem cell factors through NEUROD1, p300/CREBBP, STAT3, SPARC, FOXO1, and others, as suggested by our analysis (Fig 5A; Supplementary Text). In summary, our analysis identified pathways through which Neurogenins may repress stem cell factors and destabilize the cell's pluripotency.

As the stem cell state is inhibited, we aimed to identify portions of the network that could specifically lead to the neuronal phenotype. We identified 1,295 genes associated with neuronal GO terms (Supplementary Table S7) and found a subnetwork that could involve all transcription factors that were significantly enriched in neuronal gene targets (Fig 5B). NEUROG1 and NEUROG2 have been previously shown to directly activate NEUROD1 (Roybon *et al*, 2010), a key factor in adult neurogenesis (Gao *et al*, 2009), and our data suggest that its regulatory functions are strongly activated on day 1 and fortified each day thereafter (Fig 5B). NEUROD1 could then activate other neuronal transcription factors including NEUROD2. Our analysis further suggests that the Neurogenin expression also induces neuronal transcription factors, such as ISL1, PAX6, POU3F2, POU4F1, TLX3, and ZEB1. Furthermore, inhibitors of neurogenesis were repressed, including HES1 and REST ($P < 0.003$; Fig 5B), thus activating a few dozen neuronal genes. As the Neurogenins activate the transcription factors in our neuronal subnetwork, many downstream neuronal genes were expressed in the iNGN cells, resulting in a concerted activation of neuronal fate commitment. Thus, these transcription factors likely guide the suppression of stem cell factors and the activation of proneural factors, therein forming a connected gene regulatory network that drives human stem cells rapidly into a highly homogenous population of neurons with bipolar morphology.

## miRNAs assist in neuronal differentiation

Having identified key transcription factors, we considered the contributions of expressed miRNAs. We initially analyzed the correlations between the expression levels of miR-302a-d, miR-124, miR-96, miR-9 and miR-103 and their experimentally validated (miRTarBase 4.4 (Hsu *et al*, 2011)) mRNA targets that are expressed in iNGN cells (Supplementary Fig S8). Several mRNAs targeted by neuronal miRNAs (i.e., miR-124, miR-9 and miR-96) were downregulated upon increased miRNA expression, consistent with expectations of the role of miRNAs in repressing downstream targets, whereas for the decreasing miR-302 cluster and miR-103, similar proportions of the targets were up- and downregulated. These data suggest that during iNGN cell differentiation, miRNA functions are more biased toward *de novo* repression of upregulated miRNA targets than in disinhibition (activation) of targets of decreasing miRNAs.

We specifically found 66 miRNA interactions with the transcription factors in our regulatory network, of which 10 were significantly negatively correlated in expression with neuronal transcription factors (depicted in Fig 6A; Supplementary Figs S7 and S9). For example, REST, a validated miR-9 target (Packer *et al*, 2008), decreased in expression after day 0, consistent with the increase in miR-9 levels (Fig 6A; Supplementary Fig S8). In addition, ZEB1 and ZEB2 expression also increased over time correlating with decreased levels of corresponding miRNAs (miR-200c, miR-205 and miR-221). Beyond a couple dozen validated miRNA/transcription factor target pairs within our constrained regulatory network, we found hundreds of validated instances of miRNA regulation on neuronal and stem cell genes (Fig 6B and C).

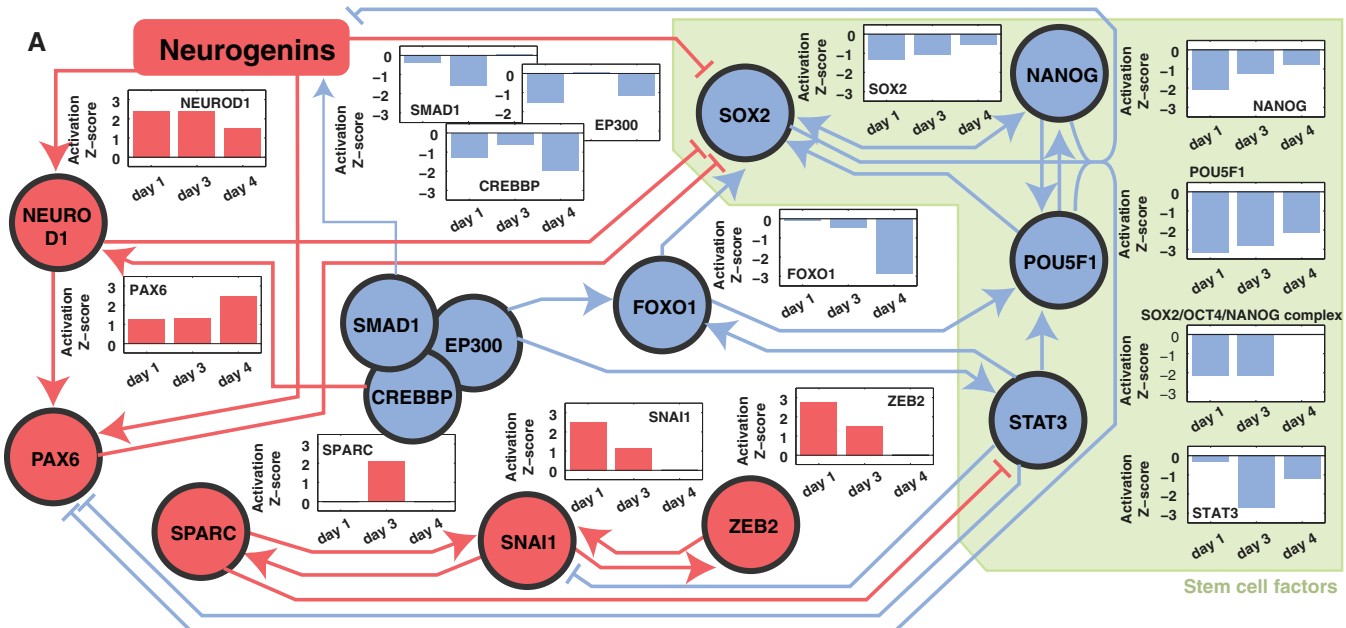

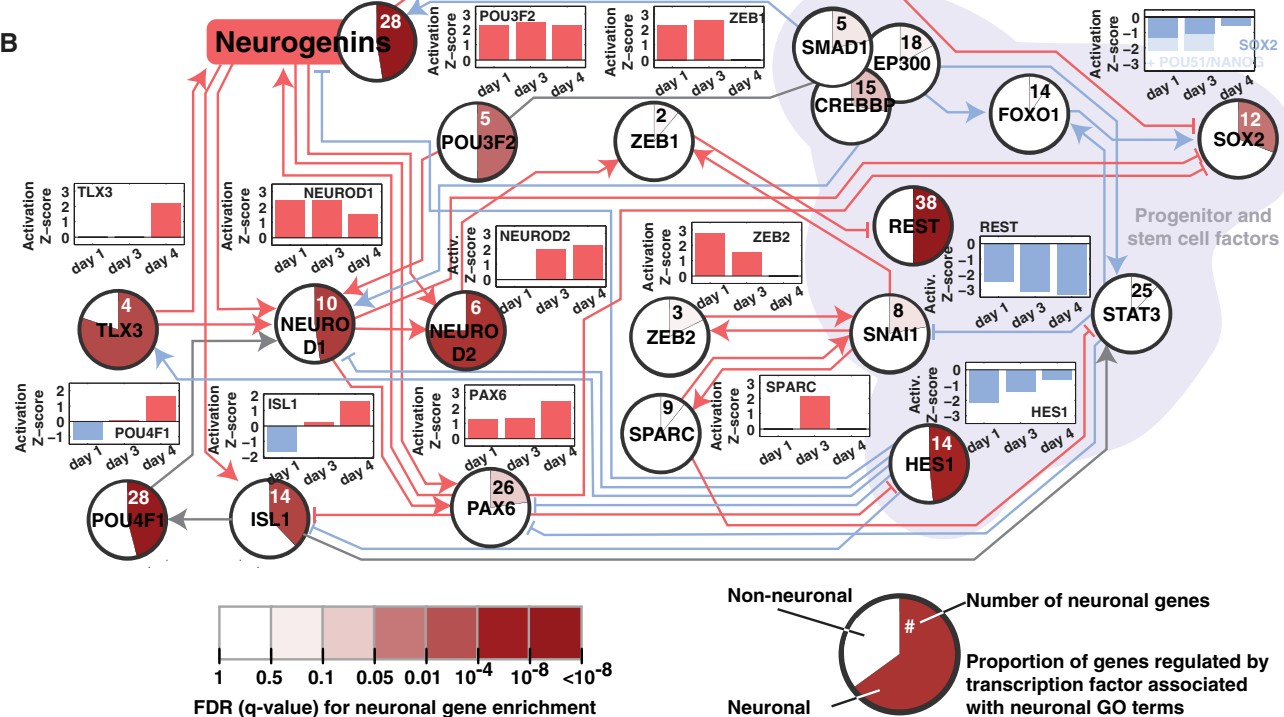

**Figure 5.  Neurogenins induce a network of transcription factors that mediate iNGN neurogenesis.**

A network of transcription factors involved in iNGN neurogenesis was elucidated from the transcription profiles using Ingenuity IPA (see Materials and Methods and Supplementary Fig S7).

A  Within this network, there is a subnetwork of transcription factors that represses stem cell factors following Neurogenin activation. The downstream genes regulated by each transcription factor were used to determine whether each transcription factor was activated (positive activation Z-score; red) or inhibited (negative activation Z-score; blue), based on differential gene expression changes seen each day (i.e., day 0 versus day 1, day 1 versus day 3, and day 3 versus day 4).

B  Neuronal transcription factors in our network were identified by looking for a significant overrepresentation (hypergeometric test, $q < 0.05$) of neuronal genes among their known target genes (using a list of 1,295 neuronal genes based on Gene Ontology). The fraction of neuronal gene targets for each transcription factor is shown in the pie charts, with the significance of overrepresentation of neuronal genes shown with color intensity. A minimal subnetwork linking all neuronal transcription factors back to the Neurogenins was identified, showing that the Neurogenins activate proneural transcription factor cascades and suppress transcription factors inhibiting neuronal genes.

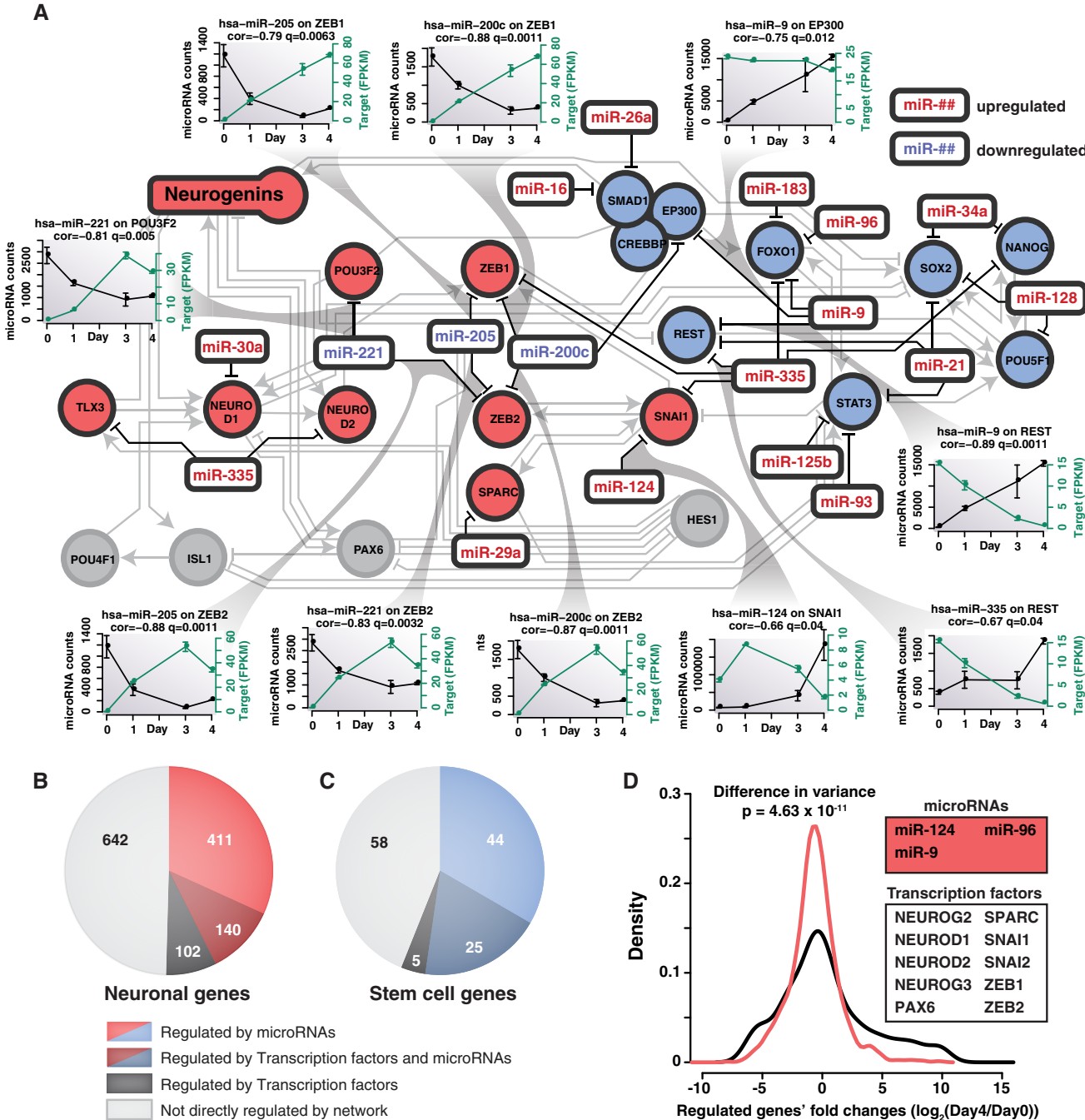

**Figure 6. miRNAs contribute to the gene regulatory network.**

A   Validated transcription factor targets for differentially expressed miRNAs were identified from miRTarBase. The miRNA interactions (black) have been superimposed on the previously generated regulatory network (light gray). Most interactions involved upregulated miRNAs that suppress stem cell factors in our network. Inset plots show cases with significant anticorrelation between miRNAs (green) and their transcription factor targets (black).

B, C   The miRNAs and transcription factors regulate many additional downstream (B) neuronal and (C) stem cell genes during iNGN differentiation. Neuronal and stem cell genes were determined based on GO terms listed in Supplementary Table S7.

D   The fold changes of downstream-regulated genes by neuronal miRNAs (red) and selected neuronal transcription factors in our network (black) were compared and indicated that regulation by transcription factors exhibits a higher impact, that is, broader range of fold change, than seen for miRNA targets (Levene's test).

Overall, miRNA-mediated repression seemed to be interweaved with transcription factor effects that occasionally must have outpaced miRNA functions, resulting in positive correlations among validated miRNA/target pairs. Consistent with this view, fold changes of validated miR-124, miR-96 and miR-9 targets were often smaller than the targets of the proneural transcription factors in our network (Fig 6D).

To further test the impact of miRNAs in iNGN cell differentiation, we knocked down the expression of the miR-302/367 cluster and miR-124 in iNGN cells by miRNA sponges (Ebert *et al*, 2007). We analyzed some of their validated targets by qRT–PCR and detected significant increases in expression levels during differentiation. However, perturbations to miRNAs did not induce noticeable changes in iNGN differentiation or iNGN cell morphology (Supplementary Fig S10). Thus, the overall regulation impact of the proneural transcription factors during iNGN differentiation appeared to be more potent compared to upregulated miRNAs.

### Validating and challenging the genetic program in iNGN cells

Our transcriptomic analysis identified several regulators that may contribute to the rapid differentiation of neurons. To verify the contribution of key factors in our network, we perturbed their expression by small hairpin (shRNA) as well as small interfering (siRNA) RNAs and assessed the morphological impact and expression of several downstream neuronal genes.

NEUROD1 is a central factor in our network and is a direct downstream target of the Neurogenins (Roybon *et al*, 2010). Its strong activation on day 1 should further activate at least 10 genes with neural annotation plus several other transcription factors, based on reported targets in IPA. We knocked down NEUROD1 with shRNAs against NEUROD1, in a construct with a GFP reporter and a puromycin selection marker to enable visualization and selection of transfected iNGN cells (Fig 7A). The shRNAs downregulated NEUROD1 levels to $22 \pm 16\%$ of the control shRNA samples. In our gene regulatory network analysis, only one gene, SLIT2, seemed to be under unique NEUROD1 control during neuronal differentiation, whereas other regulatory factors can compensate for all other NEUROD1-controlled genes following its suppression (Supplementary Fig S11). Indeed, SLIT2 expression levels were significantly reduced on day 4 as compared with a control shRNA (Fig 7C) whereas the lack of NEUROD1 resulted in non-significant expression level changes of NEUROD2 and SOX2 (Supplementary Fig S11). Since SLIT2 influences axon development and branching (Ozdinler & Erzurumlu, 2002), we assessed the morphology of iNGN cells in which NEUROD1 was knocked down (Fig 7D). Expression of the NEUROD1 shRNA significantly changed the morphology and the quantity of non-bipolar neurons but did not affect neuronal cell fate commitment (Fig 7E and F). Thus, NEUROD1 influences the bipolar-cell-shaped morphology.

To further perturb the network, we transiently transfected iNGN cells with siRNAs against additional key transcription factors. We individually targeted NEUROD1, NEUROD2, POU3F2 and ZEB1 as well as combinations for NEUROD1/NEUROD2 and NEUROD1/PAX6. The siRNAs were transfected 1 day prior to Neurogenin induction, effectively knocking down all targets (Supplementary Fig S12). Expression levels of downstream neural genes as suggested by IPA were measured by qRT–PCR at day 1 and day 3 (Fig 7G). For example, CNTN2, regulated by NEUROD2, was significantly reduced in its expression upon NEUROD2 and NEUROD1/NEUROD2 siRNA treatment. Indeed, almost all measured downstream targets showed reduced expression, except DCX, which likely was not affected since it is also directly regulated by the Neurogenins (Ge *et al*, 2006). REST and HES1 were initially reduced but showed increased expression compared with control at day 4; both are typically repressed by the targeted transcription factors (Fig 7G). Representative immunostainings for neuronal markers were conducted to assess whether transient siRNA expression interrupted neurogenesis (Supplementary Fig S11). Consistent with the NEUROD1-shRNA knockdown, siRNA treatments failed to inhibit neurogenesis, but resulted in significantly increased fractions of non-bipolar cell neurons (Fig 7H). In addition, overexpression of REST resulted in an increase in soma size (Supplementary Fig S11).

The siRNA manipulations resulted in expected changes in expression levels of downstream neural genes, suggesting that the factors in our network indeed contribute to iNGN development through the interactions suggested in our analysis. As a whole, the underlying regulatory network is robust against perturbations: Rather than grossly impeding neurogenesis, these perturbations drive the cells to morphologically altered neurons. Gaining a systems-level view of this regulatory network and altering key nodes highlights the possibilities to fine-tune the final neuronal fate.

## Discussion

Here, we demonstrated that overexpression of Neurogenin in human iPS cells yields a homogeneous population of neurons with bipolar morphology within 4 days. The homogeneity of

**Figure 7. Validating and challenging the regulatory network.**

A NEUROD1-shRNA knockdown was conducted during iNGN differentiation (A–F).

A    The NEUROD1-shRNA knockdown construct was stably integrated in iNGN cells via lentiviral gene transfer. The shRNA was under a U6 promoter, the puromycin selection marker used an SV40 promoter, and GFP was driven from a CMV promoter. Control iNGN cells were tagged with a scrambled non-functional hairpin construct.

B, C    Quantitative RT–PCR (qRT–PCR) was conducted for (B) NEUROD1 and (C) its target SLIT2 of knockdown (sh-NEUROD1, red) and control (sh-CTRL, black) samples over the time course of differentiation in biological triplicates (normalized to ACTB).

D    Immunostainings for DAPI, GFP, MAP2, and merged channels for day 4 puromycin-selected iNGN cells are shown for sh-NEUROD1 (top) and sh-CTRL (bottom).

E, F    Significant increases of non-bipolar-cell-shaped neurons were seen in sh-NEUROD1-treated iNGN cells. Three examples of altered iNGN cell morphology upon NEUROD1 knockdown (E); GFP and MAP2-staining overlay is shown. Fraction of non-bipolar iNGN cells after NEUROD1 knockdown (F); *n* refers to the number of analyzed cells of > 3 biological replicates.

G    Transient siRNA knockdowns of individual (NEUROD1, NEUROD2, POU3F2, and ZEB1) and combinations (NEUROD1/NEUROD2 and NEUROD1/PAX6) of contributing regulators result in gene expression changes of downstream targets as suggested by IPA. These were measured by qRT–PCR (column bar inlays) on day 1 (yellow) and day 3 (green) in biological triplicates and normalized to ACTB. Control iNGN cells were transfected with scrambled siRNAs.

H    All siRNA knockdowns significantly increased the fraction of non-bipolar neurons, demonstrating that the transcription factors contribute to iNGN differentiation; numbers refer to the number of analyzed cells.

Data information: Scale bars, 20 μm. Two-sample Student's *t*-test, ***P-value ≤ 0.001, **P-value ≤ 0.01, *P-value ≤ 0.05. Error bars, ± SEM.

these cells and the rapid neurogenesis allowed us to systematically characterize the neurons at the molecular level and track the transcriptional changes during the neuronal differentiation process. This was particularly valuable since it enabled us to elucidate coherent transcriptional regulatory mechanisms through which the Neurogenins inhibit stem cell maintenance/renewal and initiate a broad neuronal differentiation program. By using homogeneous differentiated cell populations, one can elucidate

gene regulatory programs contributing to the differentiation process, thus providing detailed molecular knowledge that can guide the development of additional cell populations of interest. In addition, we identified key regulators responsible for Neurogenin-mediated neurogenesis and demonstrated that miRNAs play a complementary role to neurogenesis, likely by helping to shape neuronal differentiation. It has also been recently shown that miRNAs can repress the translation of bound target mRNAs

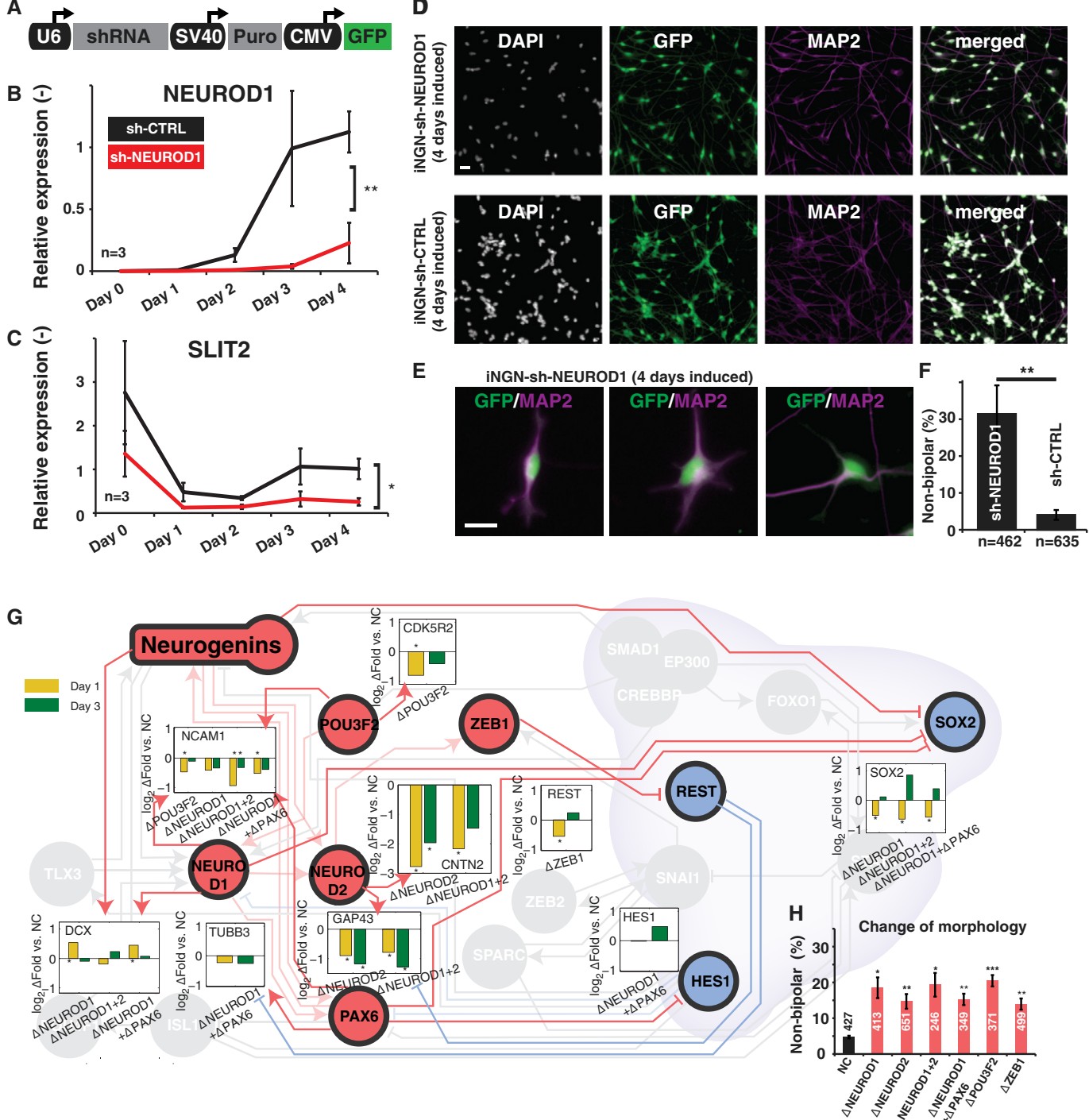

**Figure 7.**

(Meijer *et al*, 2013). Thus, it is possible that some miRNAs that did not show anticorrelation with target expression levels could be still aiding in regulation of differentiation through translational inhibition.

By perturbing key transcription factors, we found that this regulatory network is robust, but malleable, with perturbations leading to morphological variations in the resulting neurons. Using RNA-Seq, we demonstrated similarities between iNGN neurons and the transcriptomes of cells in the human developing brain.

Traditional neuronal differentiation protocols require long time lines with multistep protocols to push cells from one cellular state to the next. Here, we demonstrate the existence of differentiation pathways that continuously traverse intermediate states without additional culturing steps, thus providing the possibility of simpler and more effective differentiation protocols. In our study, the iNGN cells were kept in defined, commercially available stem cell media. Even though this medium contains growth factors that normally counteract neuronal differentiation, the Neurogenin-induced program overcame this differentiation roadblock efficiently and yielded an almost complete and homogeneous conversion to bipolar neurons. Nonetheless, neuronal maturation and electrical activity needed additional extrinsic factors despite expression of the synaptic machinery within 4 days in stem cell media. Thus, although neurogenesis can be efficiently induced even in the presence of pro-pluripotency factors, complete functional maturation still requires extrinsic neurotrophic factors.

Previous work reported that induced neurons from fibroblasts skipped neuronal progenitor states to directly become neurons (Liu *et al*, 2013). On the other hand, previous protocols using stem cells usually slowly traverse unstable progenitor states (Espuny-Camacho *et al*, 2013; Nicholas *et al*, 2013), thus usually leading to heterogeneous populations of cells and a low yield of desired neurons. The increase and subsequent rapid downregulation of neural progenitor markers and corresponding GO classes over the course of iNGN differentiation suggested a neurogenesis through progenitor states. However, SOX1, the earliest neuroectoderm lineage marker (Pevny *et al*, 1998), was not highly activated, suggesting that iNGN cells traversed later, SOX1-independent, progenitor stages in an accelerated and continuous fashion. Nevertheless, the existence of these progenitor states could present a time frame and potentially an opportunity to alter the final neuronal type, in contrast to previous transdifferentiation work where a terminal cell fate is induced directly.

One uncertainty of stem cell-derived neurons is whether fabricated neurons are relevant to *in vivo* cells. We analyzed our differentiated cells in the context of the human BrainSpan Atlas, allowing the use of hundreds of markers along with their expression levels to analyze differentiated neurons. Thus, a systematic, top-down approach (Stein *et al*, 2014; van de Leemput *et al*, 2014) can be taken to suggest which neurons resemble our iNGN cells. Direct proof of cell identity will be possible in the future as limitations from tissue heterogeneity, batch effects (Leek *et al*, 2010), and experimental variability (Robasky *et al*, 2014) are decreased by the improvement of protocols, technologies, and the development of higher resolution human brain RNA-Seq libraries, especially with single-neuron gene expression measurements (Kodama *et al*, 2012) and fluorescent *in situ* sequencing techniques (Lee *et al*, 2014). The currently available BrainSpan dataset demonstrated that day 4 iNGN

cells show greater similarity to non-cortical areas of the prenatal human brain.

Neural transcription factors used in previous stem cell differentiation and transdifferentiation protocols were also upregulated in iNGN cells, suggesting an activation of similar neuronal differentiation programs. These common regulatory elements likely drive stem cells to related neuronal cell types; consistent with this, published work shows a bias toward excitatory neurons with current protocols (Vierbuchen & Wernig, 2012; Zhang *et al*, 2013). To expand the range of neurons that can be generated *in vitro*, the genome-scale data obtained from these cells serve as a molecular blueprint of neurogenesis from stem cells, which can guide the development of additional cell populations of interest by inducing Neurogenin-decoupled transcription factors or through targeted modification of iNGN cells. For example, our interventions to the network, such as the NEUROD1 knockdown, altered the morphology of iNGN cells. One could overexpress or knockout neuronal miRNAs or use additional transcription factors or small molecules (Chambers *et al*, 2012; Ladewig *et al*, 2012). Consequently, it is possible to use iNGN cells—exploiting its speed and homogeneity—as a platform for further rational modifications to increase the variety of fabricated neurons.

Generally, each transcription factor can be considered as an important molecular 'knob' within the regulatory network, which if turned correctly, will further allow targeted engineering of differentiated cells from pluripotent cells. However, to reliably predict the outcome of subsequent perturbations to specific transcription factors, we would need additional high-resolution temporal transcriptional data of other stem cell-derived neurons. While this study successfully tied together known interactions to identify transcription factors that contribute to the regulatory network, we anticipate that as additional perturbed iNGN cells are also expression profiled in the future, more unbiased network inference algorithms can be employed to discover additional transcription factors that contribute to iNGN differentiation. Ultimately, as the network is more completely characterized, synthetic biology tools could be used to control the expression of genetic factors for targeted, rational molecular engineering of human neurons.

## Materials and Methods

### DNA constructs and lentiviral production

Mouse *Neurog1* (MMM1013-202804808, Thermo Scientific) and *Neurog2* (MMM1013-9334809, Thermo Scientific) were PCR-amplified from cDNA. A nested PCR was used to link the PCR products for *Neurog2* and *Neurog1* yielding B1_Kozak-Ngn2-2A-Ngn1_B2. This product was recombined into pDONR221 using BP clonase (11789-020, Life Technologies) to pENTR_L1_mNgn2-2A-mNgn1_L2. The cDNA-containing pENTR vectors were recombined using the LR reaction (Life Technologies) into customized lentiviral vectors based on FUW (Lois *et al*, 2002) containing a Gateway selection cassette (Life Technologies) called pLV_TRET_Ngn2-2A-Ngn1. The inducible REST overexpression vector was generated by replacing Ngn2-2A-Ngn1 by PCR-amplified REST (Addgene Plasmid 41903: LPC-flag-REST-WT, kind gift of Stephen Elledge) resulting in pLV_TRET_REST. The reverse tetracycline transactivator (rTA3)

was PCR-amplified from pTRIPZ (Thermo Scientific) and cloned into a FUW lentiviral vector containing the human EF1α promoter. The NEUROD1 shRNA vectors were purchased from Origene. Four 29-mers for NEUROD1 were applied. The following sequences were used within the pGFP-C-shLenti (TR30023, Origene) backbone including a GFP reporter and puromycin selection cassette:

a-GTCCAGAATAAGTGCTGTTTGAGATGTGA,
b-GGATCAAAGTTCCTGTTCACCTTATGTAT,
c-GCTGCTTGACTATCACATACAATTTGCAC,
d-GCCGCTCAGCATGAATGGCAACTTCTCTT.

For control transfections, a 29-mer non-effective shRNA Scrambled cassette (TR30021, Origene) within the pGFP-C-shLenti backbone was used. All shRNAs against NEUROD1 resulted in significant morphological changes of day 4 iNGN neurons. For qRT–PCR experiments, shRNA 'b' and the 29-mer non-effective shRNA Scrambled cassette were used.

The miRNA sponge sequences for hsa-miR-124 and the hsa-miR-302/367 cluster were *in silico* designed as previously described by Krol *et al* (2010), synthesized (Genewiz), PCR-amplified and placed downstream of a GFP-T2A-puromycin cassette driven by the EF1α promoter within a lentiviral vector (Addgene Plasmid 12252: pRRLSIN.cPPT.PGK-GFP.WPRE backbone, a kind gift of Didier Trono). All vector sequences were verified by sequencing. A vector containing only the GFP-T2A-puromycin cassette served as a control.

Lentiviral particles were made as previously described (Barde *et al*, 2010). For concentration, a PEG Virus Precipitation Kit (K904-50, Biovision) was used, and we determined a titer threshold by Lenti-X™ GoStix™ (631244, Clontech).

## Cell culture

The Personal Genome Project iPS cell line, derived from Participant #1 (PGP1, hu43860C), can be obtained from Coriell (GM23338, the matching primary fibroblast line is GM23248). The human embryonic stem cell line CHB-8 (NIH registration number 0007, NIH approval number NIHhESC-09-0007) was a kind gift from George Daley (Harvard Medical School, Boston, USA). PGP1 and PGP9 human iPS cells (Lee *et al*, 2009) as well as CHB-8 were cultured under sterile conditions in mTeSR media (05850, StemCell Technologies). These human stem cell lines were genetically modified by lentiviral gene transfer and genomic integration of the doxycycline-inducible Neurogenin and rTA3 vectors. The modified PGP1 cell line was named iNGN cells, and all experiments in this study were done on the PGP1 derived iNGN line unless stated otherwise. Standard tissue culture plates were coated with Matrigel hESC-qualified Matrix (354277, BD Biosciences) for 1 h at room temperature. For passaging, the cells were dissociated using TrypLE™ Express (12604013, Gibco), washed with phosphate-buffered saline (pH 7.4) (10010031, Gibco) and replated using mTeSR supplemented with 3 µg/ml InSolution™ Y-27632 Rho Kinase inhibitor (688001, EMD Millipore) and/or frozen using mFreSR media (05854, StemCell Technologies). The doxycycline (D9891-5G, Sigma) concentration for induction was 0.5 µg/ml. Even a 1-day period of doxycycline administration was sufficient to induce neurogenesis (Supplementary Fig S1). For functional studies, rat astrocytes (N7745100, Gibco) were plated on 3.5-cm poly-d-lysine-coated glass-bottom dishes (P35GC-0-14-C,

MatTek) and cultured with astrocyte medium (A1261301, Gibco) for 24 h. Next, iNGN cells were added in the presence of Y-27632 and doxycycline in mTeSR media. After 24 h, the media were changed to mTeSR containing doxycycline. After 3 days, the media were changed to (1:1) mTeSR and neurobasal A media (NBA) (10888022, Gibco) containing N-2 (17502048, Gibco) and B27 (17504044, Gibco) supplement. Notably, the supplements and the astrocyte co-cultures influenced the morphology of iNGN cells toward a higher fraction of non-bipolar shapes, and therefore, we applied these factors only for functional tests. After day 4, iNGN astrocyte co-cultures were kept in NBA (plus N-2 and B27) media.

## siRNA knockdown experiments

IDT TriFECTa™ 27-mer duplexes (three duplexes per target) HSC.RNAI.N00250.12 (NEUROD1), HSC.RNAI.N006160.12 (NEUROD2), HSC.RNAI.N000280.12 (PAX6), HSC.RNAI.N005604.12 (POU3F2) and HSC.RNAI.N001128128.12 (ZEB1) were used according to the manufacturer guidelines including the TYE-563-DS-transfection control (IDT, TriFECTa™ kit) and the negative control NC1 Control Duplex (IDT, TriFECTa™ kit). In total, 50 nM siRNA duplexes (1/4 of each duplex + 1/4 TYE-563-DS-transfection control for single siRNA targets and 1/8 of each duplex for two targets + 1/4 TYE-563-DS-transfection control) were transfected per 96-well plate (containing 10,000 iNGN cells plated 1 day prior to siRNA transfection) using the DharmaFECT siRNA transfection kit (T-2001–02, Thermo Scientific) according to the user manual (0.5 µl of DharmaFECT reagent per transfection). The transfections were performed in independent biological triplicates and related to iNGN cells transfected with 50 nM (3/4 negative control NC1 Control Duplex and 1/4 TYE-563-DS-transfection control). After 24 h, the transfections were monitored for the fluorescent TYE-563 probes and the doxycycline induction was started. Cell samples were harvested 1 and 3 days post doxycycline induction using the Power SYBR Green Cells-to-CT™ Kit (4402953, Ambion).

## Quantitative real-time PCR

20,000 iNGN cells (lentiviral transfected lines or siRNA-treated cells) were plated in Matrigel-coated 96-well plates and induced with doxycycline. The cells (< 100,000 cells per sample) were lysed at indicated time points using the Power SYBR Green Cells-to-CT™ Kit (4402953, Ambion), and RNA samples were processed for quantitative RT–PCR according to the user manual. Diluted cell lysates served as no reverse transcription (noRT) controls. The 480 SYBR Green I Master Mix (04707516001, Roche) and a LightCycler 96 System (Roche), according to the manufacturer's guidelines, were used for the quantitative PCRs. Three biological replicates were used for each condition and normalized on ACTB expression levels at indicated time points. Primers (IDT PrimeTime primer sets) used are the following:

ACTB.rev-CCTGGATAGCAACGTACATGG,
ACTB.for-ACCTTCTACAATGAGCTGCG,
REST.rev-TGGCGGGTTACTTCATGTTG,
REST.for-TGTCCTTACTCAAGTTCTCAGAAG,
NEUROD1.rev-TCCTGAGAACTGAGACACTCG,
NEUROD1.for-CCAGGGTTATGAGACTATCACTG,

NEUROD2.rev-TGGTGAAGGTGCATATCGTAAG,
NEUROD2.for-ACCACGAGAAAAGCTACCAC,
ZEB1.rev-GGCATACACCTACTCAACTACG,
ZEB1.for-CCTTCTGAGCTAGTATCTTGTCTTTC,
POU3F2.rev-GGTAGCAGGTGTAATGATGTGT,
POU3F2.for-ATCACACACTCTCCTCACTCT,
SOX2.rev-GTACAACTCCATGACCAGCTC,
SOX2.for-CTTGACCACCGAACCCAT,
CDK5R2.rev-CTCCTGTCATGTGTCACCATC,
CDK5R2.rev-GCACCTCAGTCGATCCAAA,
CNTN2.rev-ACCAGGAGGAAGCCACA,
CNTN2.rev-CTGGGAATAGCACACTGAGG,
DCX.rev-GGATCCAGGAAGATCGGAAG,
DCX.for-TTACGTTGACAGACCAGTTGG,
GAP43.rev-AGCCAAGCTGAAGAGAACATAG,
GAP43.for-TTCTTAGAGTTCAGGCATGTTCT,
C21ORF33.rev-TGTCTGGATGCGGAGTCTA (HES1),
C21ORF33.for-TCAGGAGCAAAGATCTGGAC (HES1),
TUBB3.rev-GGCCTTTGGACATCTCTTCAG,
TUBB3.for-CCTCCGTGTAGTGACCCTT,
NCAM1.rev-GACCATCCACCTCAAAGTCTT,
NCAM1.for-GAGGCTTCACAGGTAAGAGTG,
SLIT2.rev-CCTGCATCAGTAACCCATGT,
SLIT2.for-TCTCCTTCAAATCCATCAGCAC.
NGDN.rev-AGTTCAAGCTGGTGCCTATC,
NGDN.for-AGAATGAGGTGGGTCAAATCC.
GGA2.rev-TGATGCTGATGAAGAAAAGTCCA,
GGA2.for-TCCTCCTTGACCAAATTCTTGA.
KLF13.rev-ATCTTCGCACCTCAAGGC,
KLF13.for-GGGCAGCTGAACTTCTTCTC.

The data were analyzed using the $\Delta\Delta C_T$ method (Livak & Schmittgen, 2001).

## Immunohistochemistry

Cells were grown on Matrigel-coated glass coverslips and fixed for 20 min in fixation buffer (420801, Biolegend), then washed three times in phosphate-buffered saline (PBS), permeabilized in PBS containing 0.2% Triton X-100 for 15 min, and washed again three times in PBS. The coverslips were subsequently blocked for 20 min in PBS with 8% BSA and incubated for 3 h with primary antibodies in PBS containing 4% BSA followed by washing three times with PBS. Incubation with the secondary antibodies in PBS and 4% BSA was performed for 1 h, followed by washing three times in PBS. Finally, the coverslips were embedded on glass slides in ProLong Gold Antifade (P36934, Life Technologies), allowed to cure overnight, and sealed with nail polish. Primary antibodies used were the following: rabbit anti-Map2 (Abcam, ab32454), rabbit anti-Synapsin (Millipore, ab1543), chicken anti-beta-III-tubulin (Millipore, ab9354), mouse anti-NeuN (Millipore, MAB377), rabbit anti-Nanog (Cell Signaling, 3580S), goat anti-DCX (Doublecortin) (Santa Cruz, sc-8066), rabbit anti-GAT3 (GABA-transporter) (Invitrogen, 480018), mouse anti-N-Cadherin (BD, 610920), goat anti-Sox2 (Santa Cruz, sc-17319), mouse anti-Pax6 (R&D, MAB1260), rabbit anti-PSD95 (Invitrogen, 51-6900), and mouse anti-GluR2 (Invitrogen, 32-0300). Secondary antibodies/stains used were the following: 4′, 6-diamidino-2-phenylindole (DAPI, Roche, 10 μg/ml), donkey

anti-rabbit Alexa Fluor 488 (Life Technologies, A-21206), donkey anti-chicken Cy3 (Jackson Labs, 703-165-155), donkey anti-goat Alexa Fluor 568 (Invitrogen, A11057), and donkey anti-mouse Alexa Fluor 647 (Life Technologies, A31571).

## Flow Cytometry analysis

Cells were dissociated using TrypLE Express (12604013, Gibco) and washed in FACS buffer: PBS (Invitrogen) + 0.2% bovine serum albumin (Sigma). Cells were stained in FACS buffer plus anti-human TRA-1/60 antibody (clone TRA-1/60, eBioscience) and 10% fetal calf serum for 30 min at 4°C. Cells were washed twice in FACS buffer and then resuspended in FACS buffer with the viability dye SYTOX Blue (Invitrogen). Samples were collected on a BD LSRFortessa flow cytometer and analyzed using FlowJo software (Tree Star).

## SDS–PAGE and Western blotting

Whole-cell lysates of iNGN cells incubated with or without doxycycline for one or 4 days were run on SDS–polyacrylamide gels and transferred to supported nitrocellulose membrane (Bio-Rad) by standard methods. Membranes were blocked for 1 h in 5% non-fat dry milk in 1× TBS with 0.1% Tween-20 (TBST), rinsed, and incubated with primary antibody diluted in 3% BSA in TBST overnight at 4°C. The following primary antibodies were used: anti-NeuroG1 (sc-19231, Santa Cruz Biotechnology), anti-NeuroG2 (ab26190, Abcam), anti-MAP2 (AB5622, Millipore), anti-VGluT1 (ab72311, Abcam), anti-β-Actin-Peroxidase (A3854, Sigma-Aldrich), and anti-β-Actin (ACTB) (Sigma-Aldrich, A3854). Blots were washed in TBST, incubated with HRP-conjugated secondary antibodies in 5% milk in TBST for 1 h (except for anti-β-actin-peroxidase antibody), and washed again. HRP signal was detected by Enhanced ChemiLuminescence (Perkin Elmer).

## Imaging

An Observer.Z1 microscope (Zeiss) equipped with a Plan-Apochromat 20×/0.8 objective (Zeiss), a four channel LED light source (Colibri) and an EM-CCD digital camera system (Hamamatsu) as well as a Evos FL microscope (Life Technologies) equipped with DAPI and EGFP filter cubes and a Zeiss Axiovert 200 M microscope equipped with a cooled ORCA-ER charge-coupled device camera (Hamamatsu) were used. Exposure time, light intensities, and camera sensitivity were kept constant among the different samples with corresponding controls as well as image processing settings. Immunohistochemically labeled cells were automatically quantified in at least biological triplicates using Imaris software, 'Spots' in Surpass view (Bitplane AG), or manually with ImageJ v1.47 'multi-point' tool. DAPI-stained nuclei served as a reference for total cell numbers. Statistical analysis on co-localization was performed with the 'ImarisColoc' plugin (Bitplane AG).

For live cell imaging, iNGN cells were plated in a 3.5-cm glass-bottom dish and induced with doxycycline in mTeSR media for 48 h. They were then imaged on a Zeiss Axio Observer Z1 every 15 min over 48 h in an environmental chamber set to 5% $CO_2$ and heated to 37°C. The images were processed using the 'Auto Contrast' plug-in and compiled to a movie file in ImageJ v1.47. This file was converted to mpeg codec by iMovie 10.0.1 (Apple Inc.).

## Electrophysiology

Electrophysiological recordings were carried out at 20–25°C on a Nikon Eclipse TE2000-U after 4 and 14 days of treatment with doxycycline. iNGN cells were bathed in artificial cerebral spinal fluid (ACSF) containing (in mM) 140 NaCl, 2.5 KCl, 2 CaCl$_2$, 1 MgCl$_2$, 10 HEPES, and 10 glucose. Intracellular recordings were obtained using 3- to 5-MOhm glass micropipettes filled with an internal solution containing (in mM) 142 KMeSO$_3$, 5 HEPES, 0.75 MgCl$_2$, and 1.1 EGTA. Traces were collected using an Axopatch 200 amplifier (Molecular Devices), filtered with a 2 kHz Bessel filter, digitized at 10 kHz using a Digidata 1322A digitizer (Molecular Devices), stored using Clampex 10 (Molecular Devices), and analyzed off-line using customized procedures in Igor Pro (WaveMetrics). Cells were assessed for the presence of spontaneous EPSCs (sEPSCs) in voltage-clamp mode while being held at −70 mV. In current-clamp, a holding potential between −65 mV and −70 mV was maintained by constant current injection. Intrinsic properties were assessed by the injection of a set of current steps, ranging from −40 pA to 100 pA in 15-pA increments, with a duration of 0.5 s. Action potential parameters were quantified using the first action potential evoked at the lowest current injection that resulted in an action potential. Threshold was defined as the voltage at which dV/dt of the action potential waveform reached 10% of its maximum value, relative to a dV/dt baseline taken 10 ms before the peak. Action potential amplitude was defined as the difference between the threshold value (in mV) and the maximum voltage at the peak of the action potential. Width was measured at half-maximum amplitude.

## RNA sequencing

iNGN cells were plated in Matrigel-coated 6-well plates in the presence of Rho Kinase inhibitor in mTeSR media for 24 h. The media were changed to plain mTeSR, and the cells were cultured for another day until the doxycycline was added to mTeSR media. Two wells per plate were pooled for one biological replicate. In total, we generated triplicates for each time point. The day 0 samples were not treated with mTeSR plus 0.5 μg/ml doxycycline (Sigma). The cells were enzymatically dissociated, washed with phosphate-buffered saline (pH 7.4) (Gibco), and stored at 4°C overnight in RNAlater solution (Ambion). The next day, the samples were frozen at −20°C until RNA extraction. The day 1, day 3 and day 4 samples were harvested and treated accordingly. The RNA extraction was performed using the mirVana™ miRNA Isolation Kit (AM1560, Ambion) following their protocol. The protocol was interrupted after the first column purification step to obtain the total RNA. The isolated RNA was stored at −80°C and submitted to the Broad Institute (Cambridge, MA) where the quality control, library preparation (Illumina dUTP RNA-Seq Library (PolyA method)) and RNA sequencing (Illumina HiSeq (Paired End Run 101 Base)) were performed. Sequencing statistics can be found in Supplementary Fig S3 and Supplementary Table S6.

## miRNA profiling

100 ng of total RNA (aliquots of the same samples used for RNA-Seq) were used for miRNA profiling by the nCounter technology (Nanostring). A 12-reaction size kit for human miRNAs (v1) was used. All samples were processed according to the manufacturer's manual at the Broad Institute (Cambridge, MA).

Selected miRNAs were validated by miRCURY LNA™ (Exiqon Inc.) quantitative RT–PCR according to the manufacturer's manual. Briefly, 20 ng of the total RNA samples taken for RNA-Seq and nCounter experiments was used for the RT–PCR (Universal cDNA Synthesis Kit II. 8, #203301, Exiqon). A 1/80 dilution of cDNA was subsequently used for quantitative PCR using primer sets for hsa-miR-302a-3p (#204157, Exiqon), hsa-miR-124-3p (#204319, Exiqon), hsa-miR-103a-3p (#204063, Exiqon), hsa-miR-9-5p (#204513, Exiqon), and hsa-miR-96-5p (#204157, Exiqon). Each time point represented three biological replicates, and each reaction was normalized on 5S rRNA (hsa, mmu) (#203906, Exiqon). We used a 2× FastStart SYBR Green Master Mix (04673484001, Roche Applied Science) and a LightCycler 96 System (Roche), according to the manufacturer's guidelines. The data were analyzed using the ΔΔC$_T$ method (Livak & Schmittgen, 2001).

The nCounter and qRT–PCR fold changes correlated well (Supplementary Fig S5), thus allowing the reliable estimation of miR-124, which nCounter could not detect. At every time point and for each replicate, the relative miR-124 qRT–PCR expression levels were normalized to miR-302a and miR-96 and these ratios were multiplied with corresponding nCounter counts for miR-302a and miR-96 separately. We used the average value for the estimated miR-124 (miR-124$^X$) counts.

## miRNA data processing and analysis

miRNA counts were normalized to the sum of positive control probes for each replicate according to manufacturer's manual, and miRNAs with < 500 counts in all 12 samples were removed. The ANOVA test was used to find differentially expressed miRNAs with the null hypothesis that the mean count of all 4 days is the same. ANOVA *P*-values were corrected for multiple hypothesis testing using the false discovery rate method (Storey & Tibshirani, 2003), and miRNAs with *q*-values < 0.05 were considered statistically significant. miRNAs whose counts increased in day 4 compared to day 0 were considered upregulated, and those that decreased were considered downregulated. Normalized values for differentially expressed miRNAs are found in Supplementary Tables S3 and S4.

The probabilistic modeling approach for detecting dynamic miRNA contributions was performed as previously described (Schulz *et al*, 2013).

Validated miRNA targets (Hsu *et al*, 2011) were used for correlation analysis with miR-302a-d, miR-9, miR-96, and miR-103. Pearson's correlation coefficients were computed and plotted on a histogram. *P*-values were calculated and corrected using the false discovery rate method to yield *q*-values (Storey & Tibshirani, 2003).

Validated targets for active transcription factors (having positive activation score: NEUROG2, NEUROG3, NEUROD1, NEUROD2, SPARC, SNAI1, SNAI2, ZEB1, and ZEB2) and upregulated miRNAs (miR9, miR96, miR124) were combined, respectively. Expression of their targets was averaged over the triplicates and log-transformed to yield log$_2$ (day 4/day 0), then plotted as a smooth histogram with standard deviations, and variances computed. Since the variances of the two distributions were not necessarily normally distributed, Levene's test was used.

### RNA sequencing data processing and analysis

RNA sequencing reads were aligned to the human genome (Build 37, GRCh37.70). Expression levels and differential expression were determined using Cuffdiff 2 (Trapnell *et al*, 2013) in the Cufflinks package (v.2.0.2). For this study, genes were considered differentially expressed if their expression level increased by 50% in one sample, and if the *q*-value < 0.05. In total, 2,003 and 1,878 genes were significantly up- and downregulated, respectively (*q*-value < 0.05; > 1.5-fold) on day 1 compared to day 0. The number increased to 2,832 and 3,378 up- and downregulated genes by day 3, and 3,853 and 4,305 up- and downregulated genes by day 4. FPKM values are provided in Supplementary Table S1.

Gene Ontology analysis was conducted as follows. Differentially expressed genes were determined by comparing the day 0 RNA-Seq datasets to data from each subsequent day using Cuffdiff 2. All significantly up- and downregulated genes were identified. A list of background genes was also determined that included all genes for which transcripts were detected. These lists were used to look for overrepresentation of up- or downregulated genes in Biological Process Gene Ontology terms using GOrilla (Eden *et al*, 2009). All Gene Ontology Biological Process terms that were significantly enriched are reported in Supplementary Tables S2 and S8.

### Identification of GO terms containing neuronal and stem cell genes

To identify neuronal and stem cell genes that are regulated in the induced neurons, we curated the list of GO terms showing statistically significant enrichment of differentially expressed genes. Similarly, enriched GO terms associated with stem cells were also identified. All GO terms selected to identify neuronal and stem cell genes are listed in Supplementary Table S7.

### Transcription factor analysis

Analysis of transcription factor subnetwork activation was conducted using Ingenuity Pathway Analysis (IPA; www.ingenuity.com/ipa), and details of their algorithm have been published previously (Kramer *et al*, 2014). Briefly, fold change and differential expression significance were determined for each day of the experiment, compared to the previous day (e.g., day 4 versus day 3). Fold change levels for all genes were loaded into the IPA software, and upstream regulator analysis was conducted, which identifies regulators that could be active, based on differentially expressed genes. IPA was used with its default parameters, except for the following. The fold change cutoff was set at 1.5. IPA contains experimentally validated interaction data, and some predicted interactions (mostly for miRNAs). Both classes of interaction data were used for this analysis. We also used our list of expressed genes as a background list for all statistics. Lastly, since our aim was to find cascades of regulatory proteins, we did not include chemical regulators or miRNAs at this stage of the analysis. For each transcription factor or regulator, IPA first computes an overrepresentation *P*-value for each transcription factor using a one-sided Fisher's exact test to see whether more of its targets are differentially expressed than expected by random chance. Then, IPA computes an activation Z-score as described in detail previously (Kramer *et al*, 2014). Briefly, this is done by first enumerating all regulatory interactions in which the regulation directionality (i.e., activation or suppression) is well defined and then comparing up- and downregulation of each gene with the activities of an upstream transcription factor (i.e., whether the factor activates or represses a given gene). All agreements and conflicts with known regulatory mechanisms are compiled and used to compute a Z-score comparing the overlap of differential expression direction and regulation directionality, based on comparison to a null model. Thus, a quantitative measure is provided to assess how likely it is that the transcription factor is activated or repressed.

Following the identification of transcription factors that explain the patterns in differential expression, the list was analyzed to focus on transcription factors with the strongest evidence of being specific to neurogenesis in the iNGN cells. First, we focused on transcriptional regulators and regulatory complexes, which were annotated by IPA as 'transcription regulator', 'translation regulator', 'complex', 'group', and 'other' in order to capture the transcription factors involved in differentiation. Second, all regulators with an absolute activation/repression score less than 1.5 or enrichment *P*-values greater than an Benjamini FDR-corrected value of 0.05 were removed (the SOX2-OCT4 and SOX2-OCT4-NANOG complexes in IPA had scores above threshold, and so the scores for SOX2 outside of these complexes are also reported, despite being below threshold). Third, to find candidate transcription factors, those that were not significantly expressed (average FPKM < 0.5 in our datasets) were discarded, while retaining all Neurogenins. Since the mouse homologs of the Neurogenins were overexpressed here, the sequencing reads do not align to the human reference genome. Fourth, regulators were removed if there was a discrepancy in differential expression and activation state for a given day, and no further days exhibited a significant concordance. If, for example, the mRNA of the transcription factor significantly decreased, but it was predicted that the regulator was significantly increasing its activity, it would be removed unless, for another day, the mRNA was further significantly decreased with an accompanying prediction of decreasing activity. Fifth, regulators were removed if they were not connected upstream to the Neurogenins, since we were interested in finding the central factors that are specifically in cascades influenced by the Neurogenins. Since our goal was to identify local regulators that were important for iNGN differentiation, we identified more global regulators (i.e., transcription factors with more than 15 interactions within our list of transcription factors). We then repeated the fifth step without these global regulators, in order to allow the identification of pathways specific to iNGN differentiation. This resulted in a network of regulators seen in Supplementary Fig S7. See also Supplementary Table S5 for details on the network, including the aforementioned global regulators.

### BrainSpan analysis

RNA-Seq data were acquired from the Allen BrainSpan Atlas of the developing brain (http://www.brainspan.org/). The data available for download included RNA-Seq data from multiple individuals, spanning from 8 weeks postconception until 40 years old for both male and female human subjects, and from 26 different brain

structures. While expression had been acquired prior to 8 weeks postconception, these datasets were not available for download. The Pearson's correlation coefficient was computed for gene expression levels between each BrainSpan sample and the day 0 and day 4 iNGN cells. To decrease bias from unexpressed genes, all genes that had a mean FPKM level less than 0.1 were filtered out of this analysis. To test the temporal correlation of our cell lines with different developmental time points in the human brain, we computed correlation coefficients between our cell lines and each sample in the BrainSpan Atlas. Then, we computed a one-sided two-sample *t*-test to test whether the correlation coefficients for the day 4 data were higher than the day 0 correlation coefficients for all samples for each given point in the Brain Span data.

To test the brain region similarity between our iNGN cells and the human brain, we first identified 500 genes showing the largest increase in expression in our iNGN cells on day 4, with respect to day 0. Using only these 500 genes (analysis results were qualitatively robust to variations in the number of genes), we computed the correlation coefficients between the day 4 data and data for each brain region at each time point. We then computed a Z-score to see whether a given brain region correlates more highly than the remaining brain regions at each time point. Z-scores were used since they allowed the identification of brain regions that continually show higher correlation than others, and allowed enhanced comparison between brain regions since it helped to control against general transcriptomic changes seen in brain tissue over time, thus strengthening the support of a specific brain region being more similar to the iNGN cells. We note that cerebellum and cerebellar cortex data from the BrainSpan Atlas were grouped for all analyses because they did not overlap in sampling time points. However, this grouping did not qualitatively change the results of our study.

**Data availability**

Datasets have been deposited at the NCBI Gene Expression Omnibus and can be accessed with the following accession numbers: GSE60548 (Illumina RNA-Seq), GSE62145 (nCounter miRNA), and GSE62146 (Agilent microarray).

**Supplementary information** for this article is available online: http://msb.embopress.org

## Acknowledgements

We would like to thank Dr. Yoav Mayshar and Dr. Arend Hintze for commenting on the manuscript. The project was supported by NIH Grant P50 HG005550 and a grant provided by Richard Merkin to GMC. RW received support by NSF STC EBICS Grant 0939511 and NSF ERC SynBERC Grant 0540879. The Human Frontiers Science Program Organization and the Volkswagen Foundation supported VB. The Swiss National Science Foundation provided funding to VB and PG. The Ernst Schering Foundation supported PG. AHMN was supported by the Canadian National Science and Engineering Research Council, and the Lynch Foundation. SLS received support from the National Institute on Aging (5T32AG000222-22).

## Author contributions

VB conceived the project, designed and performed experiments, analyzed data, and wrote the paper. VB did not work with human ES cells. NEL analyzed RNA-Seq, microarray, and nCounter data and wrote the paper. PG developed the iNGN cell line, performed experiments, and wrote the paper. AHMN, SLS, SMB, NES, SL, and JM performed experiments and analyzed data. MS designed miRNA sponge sequences. YL provided material. RW provided material, supervised the work, and edited the paper. GMC edited the paper, supervised, and guided the research. All authors commented on the manuscript.

## Conflict of interest

The authors declare that they have no conflict of interest.

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
