## [Review Process File · Molecular Systems Biology]

Rapid neurogenesis through transcriptional activation in human stem cells

Volker Busskamp, Nathan E. Lewis, Patrick Guye, Alex H.M. Ng, Seth L Shipman, Susan M Byrne, Neville E Sanjana, Jernej Murn, Yinqing Li, Shangzhong Li, Michael Stadler, Ron Weiss and George M. Church

Corresponding author: George Church, Harvard Medical School

Review timeline:	Submission date:	14 December 2013
	Editorial Decision:	17 January 2014
	Appeal:	21 January 2014
	Editorial Decision:	07 February 2014
	Re-submission:	20 June 2014
	Editorial Decision:	30 July 2014
	Revision received:	14 October 2014
	Accepted:	16 October 2014

Editor: Maria Polychronidou

Transaction Report:

1st Editorial Decision

17 January 2014

Thank you again for submitting your work to Molecular Systems Biology. We have now heard back from two of the three referees whom we asked to evaluate your manuscript. Since their recommendations are very similar, I prefer to make a decision now rather than further delaying the process. As you will see from the reports below, the referees raise substantial concerns on your work, which, I am afraid to say, preclude its publication in Molecular Systems Biology.

In line with the increasing interest in developing direct differentiation protocols that circumvent intermediate differentiation states (Wapinski et al., Cell 2013, Liu et al., Nature Communications 2013), the reviewers acknowledge that the presented fast and efficient protocol for the differentiation of iPS cells into neurons is a potentially useful tool for directed neuronal differentiation. However, they raise significant concerns regarding the analysis and interpretation of the transcriptomics data and they are not convinced that the study provides sufficiently conclusive functional and mechanistic insights into the regulatory network underlying the neuronal differentiation of stem cells. While they find the network analysis potentially interesting, they point out that as it stands, it relies on heavy filtering (Ingenuity Pathway Analysis) and it remains unclear whether it can conclusively explain (or predict) the observed dynamics and robustness of the differentiation protocol. As such, both reviewers indicated that they would not support publication in Molecular Systems Biology.

Considering these rather substantial concerns and the overall low level of support, we feel that we have no choice but to return this manuscript with the message that we cannot offer to publish it.

Nevertheless, we recognise that, as indicated by the reviewers, your efficient differentiation protocol using PGP iPS cells represents a strong aspect of the study and understanding the systems-level mechanisms underlying the fast and robust neuronal differentiation would be interesting. As such, we would not be opposed to considering a revised and extended study that would include further experimental data to conclusively address one of the following issues:

- Provide experimentally validated insights into the mechanisms that underlie the dynamics of the direct differentiation process and its robustness. This would include a more systematical perturbation analysis of the initial/central regulatory events (including NEUROD1 and NEUROD2) and the unbiased analysis of the resulting temporal gene expression changes.
- An alternative would be to considerably strengthen the practical aspect of the study by demonstrating that the protocol can be extended/modified to efficiently generate, in a controlled manner, a broad diversity of neuronal cell types. This would require phenotypic and/or cell-type specific marker analysis of the resulting neuronal cell types and would need to go beyond the description of variations in neurons' soma diameter and axon gross morphology.

We would be happy to discuss these possibilities in more detail over the phone. Perhaps there would be other (maybe further reaching) alternatives to expand this work, such as demonstrating the power of using personal genome information in combination with personal iPS cells by functionally testing a hypothesis generated by integrating genomic (and epigenomic) data available for PGP samples with the expression data during or after differentiation.

A resubmitted work would have a new number and receipt date. We recognize that this would involve substantial additional experimentation and analysis and, as you probably understand, we can give no guarantee about its eventual acceptability. If you do decide to follow this course then it would be helpful to enclose with your re-submission an account of how the work has been altered in response to the points raised in the present review.

I am sorry that the review of your work did not result in a more favorable outcome on this occasion, but I hope that you will not be discouraged from sending your work to Molecular Systems Biology in the future. In any case, thank you for the opportunity to examine this work.

Reviewer #1:

Summary

In this manuscript, Busskamp et al describe a new method for induced differentiation of human iPS cells by overexpression of two neurogenin transcription factors, rapidly generating bipolar neurons at very high yields of purity. The paper addresses the important question of directed differentiation of stem cells. For the particular case of neurogenesis, this question has considerable potential in investigating fundamental brain functions but also application into fields in regenerative medicine. The authors analyze the morphology and functional properties of these cells and observe that the differentiation protocol leads to competent neurons able to reach maturation. They then perform transcriptional characterization for a time-course of differentiating cells through RNAseq and miRNA profiling analysis. Comparison of transcriptomic data with existing human brain reference data (the BrainSpan atlas) shows high correlation between induced differentiated neurons and prenatal cerebellar and adult thalamic neurons. The authors then use their data, as well as an existing knowledgebase to try to elucidate the core transcriptional regulatory network of differentiation, including both transcription factors and miRNAs. Finally, the robustness of the differentiation process is tested by perturbation of two known regulators of neurogenesis (NEUROD1 and REST).

General remarks

The strong point of this study is, in my opinion, the establishment of a rapid and efficient protocol for the differentiation of bipolar neurons that circumvents current limitations of differentiation from ES and iPS cells or transdifferentiation from fibroblasts, such as low purity yields or suboptimal protocols involving a large number of steps. The final section of the study also indicates that modifications of this system should provide a good basis for directed differentiation of other neural

cell types.

However, I have major reservations regarding the subsequent transcriptomic analysis and gene regulatory network construction sections of the study, and find the results insufficient to substantiate some of the author's main claims. In particular, the study does not provide sufficient detail on mechanistic aspects of network dynamics to warrant the conclusion that the regulatory network underlying the neurogenin-induced differentiation of iPS cells is deterministic, nor does it sufficiently explore network principles underlying robustness of the system. The computational methods are also not clearly described which, together with some confusing sections of the results (both text and figures) and rather brief figure legends, makes it difficult to follow the manuscript and interpret the findings. This may be partly due to space limitations but nevertheless is a major issue.

In many instances it is difficult to tell to what extent conclusions drawn are supported by the authors' own observations or are based on information taken from other published studies. As the authors say in the discussion, the data provide a valuable molecular blueprint for neurogenesis, and the cell model offers a platform for further directed differentiation of other neuronal cell types. However, the transcriptomic and computational results of the study, albeit comprehensive and interesting as a descriptive resource, are disappointing. Because of the extensive utilisation of pre-existing data, relatively little novelty emerges in terms of biological insight into mechanistic aspects of the regulatory network, and the methods used are not novel in themselves. As a result I feel the study will be of great interest to the iPS and neural audiences, but of less interest to the systems biology and computational communities. In my overall assessment of the manuscript I have tried to balance the value of the iPS differentiation model and accompanying global transcriptome data against the less compelling regulatory network analyses, bearing in mind the likely audience for Molecular Systems Biology.

Major comments

For gene regulatory network construction, the authors have used a specific functionality of the Ingenuity Pathway Analysis software (upstream regulator analysis). From my understanding, this method is heavily based on Ingenuity's own knowledgebase regarding previously described regulators and interactions. In a sense, the analysis makes sense of the experimental observations in the context of what is already known in the literature, and stored in the database. Thus, the constructed network will inevitably be limited to existing knowledge and therefore, by default, novel regulatory interactions that might stem from the author's datasets are likely to be discarded or discounted. What is gained in conformity with the literature is lost in terms of potential novelty. In the methods section, the steps for calculating activity Z-scores, crucial for reaching the presented core regulatory network, should be made clearer.

The steps for filtering the list of enriched regulons are not entirely clear, and in some cases there seem to be inconsistencies with the presented figures. In particular, the authors state that regulators were removed if there was a discrepancy in differential expression and activation site, but in figure 5B, for instance, ISL1 is included even though it significantly increases expression from day 0 to day 1 but has negative activation Z-score. In general, I found the filtration protocol to be excessively supervised (for instance, the authors set an absolute activation/repression cutoff but then open exceptions for SOX2 and SMAD1) including a final manual curation step. The analysis is already constrained by Ingenuity's stored information and the filtering further reduces the likelihood of discovering novel regulators/interactions.

Beyond the methodology, the analysis and interpretation of the regulatory network results is also not entirely clear to me. In particular the way results are described, for instance in the loss of pluripotency section of the results, makes it unclear what is predicted from the analysis and previously known from the literature. As far as I can see, the regulatory links discussed are putative links, based on the proposed network, and "targets" refers to putative targets based on the Ingenuity database, most likely in unrelated cell types (e.g. Line 266: "over-representation of differentially expressed targets"). The authors then reference published work that supports the existence of such a link in other systems. However, the text as written rather suggests (no doubt unintentionally) that these putative regulatory interactions can be clearly seen in this study, rather than that the proposed network links are simply consistent with other published data. For example, I could not see sufficient data in this study to support the statement that "By day4, the sequestered p300/CREBP complex fails to activate FOXO1" Similarly "Our data suggest that neurogenins directly inhibit

SOX2" (line275) - how have the authors drawn a conclusion about direct versus indirect regulation? This mixing of information is made worse by a couple of instances where there is a discrepancy between prediction and existing knowledge, such as where NEUROG2 is referred as activating NEUROD1 after binding a p300/CREBBP complex, but in figure 5A this interaction is not represented. Furthermore, it is not clear if the timing for the regulatory effects is inferred from the analysis, the literature or both, or how the labels "immediate repression" and "secondary repression" were derived. In order to highlight the value and novelty of their observations, the authors should first state what predictions are specifically produced by their computational analysis and then put them into context with existing literature.

Some broader, strong conclusions are drawn based on the proposed network, such as (line 288) "from our data a secondary mechanism arises in which stem cell factors are repressed primarily by inhibition of STAT3 regulation". In the absence of any functional validation experiments, I do not feel this conclusion - so forcefully stated - is supported by the data.

The conclusion to this section (lines 313-315) is, essentially, that neuronal differentiation from iPS cells involves the repression of stem cell transcription factors and activation of neuronal transcription factors. This is not a particularly surprising or novel conclusion in itself, and reinforces the need for the authors to validate at least some of the regulatory interactions that they propose, preferably ones that are novel at least in this cell type even though previously described in the literature. Since the study already utilises lentiviral expression of shRNAs this approach would not be unduly onerous or time-consuming.

After constructing the core regulatory network, the authors do perturb expression of two known neurogenesis regulators and observe that differentiation is still possible, although with some morphological differences in the obtained neurons. Although this suggests robustness of the network, the study does not sufficiently explore the impact of the perturbations in the topology and dynamics of the network to support that claim. The authors provide a possible explanation by retrospectively using the constructed network and the list of putative targets for REST and NEUROD1 for hypothesizing why differentiation was still possible. Ideally, a more comprehensive and detailed assessment of network robustness would be to obtain genome-wide transcriptomic data upon perturbation and compare the constructed networks. Given the scale of these experiments, a less costly and time-consuming alternative would be to assess whether the expression of a reasonable subset of putative REST and NEUROD1 targets is - at least initially - affected by the perturbations, and perhaps functionally test some of these targets.

In addition, the authors suggest that the limited impact of the NEUROD1 knockdown may be due to compensatory upregulation of NEUROD2. Before proposing this, the extent of knockdown achieved for NEUROD1 should be determined and shown, at both the RNA and protein levels. Furthermore, since NEUROD2 is downstream of NEUROD1 (Fig 5B), the expression of NEUROD2 should be assessed in the NEUROD1 knockdown cells, to confirm that it is still upregulated and that the proposed compensation is plausible.

In the title, and several sections of the text, the authors describe the regulatory network as being deterministic, and it is not clear to me in what respect this is meant. In the first instance, the differentiation process can be seen as deterministic given that the final outcome in around 90% of the cases is the same (bipolar neurons). However, as in my previous comment, my impression is that the authors depart from this empirical observation to make a general comment about the network. In this respect, I do not think the study has enough single-cell resolution to comment upon the deterministic/stochastic mechanistic nature of the network. In particular, it is not possible to ascertain whether the behavior of pluripotency and differentiation regulators is homogeneous and coordinated among cells at the different stages, or instead individual cells show heterogeneous expression of regulators suggestive of different paths of differentiation upon neurogenin overexpression. Having narrowed down the number of regulators in the core network (and established a basic architecture), it would now be interesting to assess (for instance by single-cell RT PCR) the level of heterogeneity of these genes in single cells, at different stages of differentiation (days 0 - 4) and relate these observations with the current picture. In such a study, Buganim et al have profiled single cells in the context of a reprogramming experiment, describing an early stochastic stage with large variability, followed by a late hierarchical phase coordinated by key regulators (Buganim et al. 2012. Cell 150: 1209-1222).

Minor comments

As a general comment, the figure legends are too brief and lack the detail required to enable the reader to easily interpret the figures. More specific recommendations on the text and figures are detailed below:-

Figure 2D, it would facilitate interpretation if labels for the different classes were presented in the same color as the dashed boxes (e.g. "cell adhesion" in orange, "RNA metabolic processes" in green, etc). Also the explanation for the selection of the genes in the inset panel should be given in the legend as well as the main text.

Fig 3A:- the p-values for the Day 0 cells should also be added to the plot.

Fig 4A legend: The meaning of "Distribution of microRNA quantities" would be clearer if worded as "relative abundance of microRNAs" or similar.

Fig 4C legend: At which timepoint are these the 5 most highly expressed miRs? Also the sentence needs to be reworded for clarity.

Fig 4D,E legend: please define "significantly"

Line 243 - please indicate here the magnitude of the increases observed in miR-96 and miR-9.

Line 245: the increase in miR-9 is linked in the text to Fig 4B but it seems to feature only in Fig 4E

Lines 253-261: the authors should include some more detail here of the approach used. In particular, Ingenuity Pathway Analysis should be mentioned somewhere other than just the methods section. Key, and non-obvious, aspects of the filtering should be briefly outlined here, in particular that regulators were removed if there was a discrepancy in their differential expression and putative activation state or if they were not connected upstream to the neurogenins.

Fig 5 - some of the plots are labelled as z-scores, not activation z-scores. Presumably this is an error or due to lack of space in the figure. If all plots are activation z-scores then it would be better to put this into the legend and remove it from all the plots. The figure would then be a little less cluttered. It is also counter-intuitive that the key describes the colours in terms of immediate and secondary repression but many of the interactions are arrowheads, which conventionally denotes activation, so this apparent contradiction should be explained briefly. There is a typo in the figure labelling of Fig 5B (progenator instead of progenitor).

Fig 5 legend: - the figure title refers to a "core gene regulatory network" and this phrase is then used repeatedly in the text, along with "core transcription factors" (e.g. line 318). Presumably this refers to the combination of Figs 5A and 5B but this should be clearly stated and the authors should check that this term is then used consistently to refer to the same set of interactions and/or regulators. The legend should also remind the reader what the total set of neuronal genes is as used in fig 5B (ie how many there are and how they were defined). Overall the legend should include more information to facilitate interpretation (e.g. it's not immediately clear to the reader that blue and red circles mean inactivation and activation regulators in 5A) and the use of colours (orange, blue and grey) for the various connectors in the network in fig 5B should be explained.

Line 290:- "In summary, our pathway analysis revealed connections of..." Throughout the manuscript it is unclear whether regulatory relationships (ie between a transcription actor and a target gene) are putative, based on the proposed network, or well-established in the literature. In places this is understandable but I would strongly argue that - in the absence of experimental validation - the authors can only claim that their pathway analysis reveals putative or proposed connections of Neurogenins with repression of stem cell factors and these terms should be used wherever relevant. Alternatively, such comments could be prefaced with "In our model/network...".

Line 305:- I found the sentence "Our transcriptomic data shows increasing inhibition of REST regulation" a little unclear. Presumably "regulation" here refers to REST activity (inferred from expression of its targets, and as shown in fig 5B) or does it actually refer to inhibition of

transcriptional regulation of REST? The transcriptomic data can really only show the latter; it has to be combined with the network analysis in order to derive the information in fig 5B. This should be clarified.

Fig 6 legend - the legend should state that the miR interactions (in black) have been superimposed on the previously generated regulatory network (in grey); because of the amount of information in the figure, many of the grey connectors appear to be coming from the miRs. Also, the key in the figure (upregulated/downregulated) should be boxed or similar to make it more obvious to the reader.

Fig 6B:- presumably the 1295 neuronal genes in the pie chart are the same set as used in Fig 5 but, if not, this needs to be stated.

Line 326:- "More neuronal microRNA targets underwent downregulation...". More than what? Non-neuronal targets of the same mIRs?

Line 338:- "The broad target range..." I don't quite understand what point is being made here, or how it explains why only a few microRNA/validated target pairs were detected in the core network

Line 348:- again, in the absence of experimental verification of microRNA interactions in this cell system, it would be more accurate to say that the changes in microRNA expression are such that they would/could assist the core TFs in repressing stem and neural progenitor factors.

Fig 7:- I missed any information regarding the extent of the knockdown achieved for NEUROD1 at the RNA and protein level. Without this information, the simplest explanation for the failure to arrest neurogenesis is simply that the knockdown was insufficient. This information should be added to Fig 7 or the supplementary information.

Line 384:- "19% of the neuronal genes were under REST or NEUROD1 regulation". In the context of discussing the REST and NEUROD1 perturbations, at first sight this suggests that the expression of these genes was seen to be affected by the perturbations. This text should be altered but, in any case, as mentioned earlier, the authors should undertake some gene expression analysis of the cells as soon as possible after the perturbations to try to verify some aspects of their network.

Line 394:- "...small changes to the accessibility of a minority of neuronal factors". I don't understand what this sentence refers to, or what data it is based on.

Reviewer #2:

The manuscript by Busskamp et al analyzes developmental program driven by Neurogenin1/2 that converts pluripotent human stem cells to nerve cells. Similar to previous report (Zhang et al, 2013), the authors demonstrate efficient conversion of human stem cells to neurons by inducible expression of Neurogenins. Expression analysis of programmed cells on day 1, 3 and 4 revealed progressive changes in cell identity and the authors assemble these changes into modular networks that represent "deterministic regulatory network" controlling neurogenesis.

1. Understanding the mechanisms of conversion of pluripotent cells to neurons by two transcription factors is of high importance. However, it is not clear whether the authors provide any new insights into the process. The networks presented in figures are heavily filtered for genes that were previously implicated in neuronal differentiation, thus eliminating potential new discoveries. Importantly, the authors do not include information about Ngn2 targets (by performing ChIP-seq) and the networks derived from time-course analysis provide little information about hierarchy of individual regulators. Overall it is impossible to determine which regulators, pathways and interactions are important and which are secondary or even irrelevant. I would therefore recommend eliminating the filters described in Methods section and reanalyze temporal expression changes in an unbiased way. Furthermore, it would be extremely helpful to compare the process of neural differentiation driven by Neurogenins with neuronal differentiation driven by extrinsic factors (e.g. Chambers et al, 2009, or retinoic acid treatment).

2. The authors picked two genes that were previously shown to play important role in neuronal

differentiation - REST and NEUROD1. They performed overexpression and knockdown studies and concluded that while these manipulations change phenotypes of resulting neurons they do not interrupt neuronal differentiation program. It is not clear whether this conclusion extends to all neuronal differentiation programs or just to the one driven by Neurogenin expression. Control experiments where genetic manipulations are performed in cells neuralized by SMAD inhibition or retinoic acid treatment should be included.

3. I assume that the authors performed expression analysis in cells overexpressing REST and in NEUROD1 knockdown cells. This seems to be the basis for figure 7G, but proper description of the experiments and more detailed analysis is missing. This needs to be rectified, otherwise sentences such as: "19% of the neuronal genes were under REST or NEUROD1 regulation. In our gene regulatory network, only one gene, SLIT2, cannot be activated by other regulatory proteins following the removal of NEUROD1", do not make much sense and sound like arbitrary cherry-picking.

4. What does it mean that the resulting cells are similar to prenatal cerebellar cortex AND postnatal mediodorsal nucleus of thalamus? Does it imply that the cells are confused and their identity cannot be matched to any known type of nerve cell?

5. Overall the manuscript reads like presentation of data without much insightful interpretation and the overall message is lost or difficult to follow.

Appeal

21 January 2014

We want to thank you for your fast communication of your decision letter and for the detailed description of your assessment. We also greatly appreciate your suggested improvements, which are in fact in line with future work in my group. At this time we would like to discuss a few points in regard to your decision.

Specifically, we were pleased that the reviewers could see the importance of the cell differentiation protocol and its advantages compared to established differentiation protocols; however, we are also quite surprised that other novel aspects of our work were ignored or taken for granted. First of all, in our manuscript, we mention the upregulation of neuronal genes and the suppression of stem cell genes, which the reviewers deem to be obvious. However, assessments of this level of molecular detail are rare for differentiated cell lines, in part because differentiation protocols usually yield only a small percentage of differentiated cells. Thus, omic-level analyses are difficult because of the high level of signal from incompletely differentiated cells, or perturbed transcription from protocols used to isolate the desired cells. Our detailed characterization was only possible as a result of the high homogeneity of our cells. Indeed, transcriptomic, immunohistochemical and functional analyses consistently indicate the presence of a >90% homogeneous neuronal population, i.e. bipolar neurons. Most differentiation studies base their success on a limited number of transcriptional, morphological and functional features, including, for example, protein markers like TUBB3, which can also frequently be detected in neuronal progenitor cells. Furthermore, these are often measured at the endpoint of the differentiation protocol. However, the high homogeneity of our cells allowed us to augment the classical measurements with RNA seq and microRNA profiling, thus providing a complete picture of neuron-related transcriptional changes in our differentiated cells, measured at multiple time points, and therefore serves as a more complete assessment of the extent of differentiation and a verification of cell type.

Second, even more importantly, we identify the molecular pathways needed for differentiation. Neuronal development has been heavily studied *in vivo* and therefore several transcription factors and their regulated genes are known to influence the differentiation of selected sets of specific neuronal types. Furthermore, the transcriptional starting points for differentiating neurons *in vivo* are different from the starting states in adult stem cells, and so several stem cell factors have been previously revealed. Given this body of knowledge, we don't claim novelty of individual interactions; however, we used an extensive database of measured interactions to reveal and assemble the pathways of regulatory interactions that lead to rapid and direct differentiation of

iNGN cells from human stem cells. This is particularly important since generally the mechanisms underlying the differentiation of stem-cell derived neurons remain as a black box, with a few cherry-picked transcription factors provided as inputs and few transcriptional, morphological and functional features as outputs, often measured at the time points when the researchers completed their experiments. However, here, for the first time we have clear and continuous pathways through which the differentiation process takes place, without requiring the use of intermediate treatments to activate different pathways. Thus, we detail the uninterrupted pathway to the differentiation of iNGN cells.

Third, many new publications link microRNAs to neuronal development. Our time course profiling of microRNA changes suggest that in iNGN cells, microRNAs do not seem to trigger neurogenesis (which is an ongoing debate in the neuronal development field). Rather, the microRNAs seem to be shaping the differentiation downstream of the core transcription factor regulators. We were surprised by the lack of reviewer comments on this substantial part of our manuscript.

While we feel the full impact of these aforementioned points were missed in the review process, the reviewer comments and your assessment raised some important and legitimate concerns that we agree must be addressed to improve its delivery and solidify our findings. We see three important areas that can be improved to strengthen this work.

First, we noticed several misunderstandings in the reviewer comments that underlay some of their concerns. While some of them hinted that the methods section was not carefully reviewed, we feel that a significant improvement can be made in our work simply by clarifying the text and methods sections. Furthermore, minor improvements to the structure of the manuscript should draw more attention to the most significant points of impact this work has.

Second, there was a concern of the novelty of the regulatory interactions. As stated above, we used a large database of experimentally-measured interactions, and therefore acknowledge that the interactions themselves are not novel. However, the pathway structure is indeed novel and has never been detailed to this extent for neurons obtained from stem cells. Thus, to demonstrate the novelty of the pathways, we propose to compare our pathways to other pathway databases. This would be a rather simple analysis for us to complete.

Third, to further validate and test the robustness of the regulatory network, we are now doing more extensive perturbations. Specifically, we aim to knockdown NEUROD1 together with NEUROD2 as well as ZEB1, ZEB2, POU3F2 and PAX6 by siRNAs individually or combinatorial. Eventually we hope to use shRNA constructs analogous to the NEUROD1 experiment in the manuscript. As readout, we plan to test direct downstream targets by real time PCR in a fast and cost-efficient way. Furthermore, morphological changes are a clear indicator of altered neuronal subtypes. While we would like to conduct a more in depth transcriptomic analysis involving multiple RNA seq analyses of individual and combinations of manipulations of iNGN cells at diverse time points, such an effort would take much longer and hinder the dissemination of the differentiation protocol. Furthermore the scale of such an effort would definitely exceed the size limitations of a MSB article, and so it would probably be best done as a separate publication. As mentioned in the cover letter, there is high competition in the field to reveal the molecular pathways mediating rapid neurogenesis in human stem cells since the molecular systems view is the key to systematically direct the generation of additional neurons including inhibitory ones.

Lastly, we want to note that this cell line is already being distributed for use by other groups and will be more deeply characterized in the coming years. Specifically, we submitted iNGN cells to the ENCODE project and they will perform their extensive (epi)genetic, transcriptomic and genomic characterizations, including ChIP-seq for hundreds of transcription factors. Additionally, we shared iNGN cells with groups studying epilepsy, developing novel optogenetic tools or studying specific epigenetic marks during neuronal development. These follow-up data will be complementary to our work and further substantially increase the value of the iNGN cell data we provide here. Thus, this manuscript will be extensively cited for its method, for the cell line itself, and for the differentiation pathways.

In summary, we are grateful for your suggestions and for Reviewer #1's very extensive evaluation.

We are happy to address most of the issues to improve the clarity and readability of our manuscript. However, in our eyes, the lack of spectacular novelty concerns by both reviewers are unjustified and technically biased. While our results seem to be obvious and therefore descriptive, improved presentation and validation while bring out the major contributions of this work, including the method, the ability to do use the cells for acquiring systems level data, the identification of regulatory pathway analysis including microRNAs involved in differentiation, and the comparison of human stem cell-derived neurons to actual cell populations in the developing human brain. As you suggested, we feel it may be fruitful to further discuss options on the phone how to proceed with our manuscript.

2nd Additional correspondence

31 January 2014

Thank you again for your reply letter on our decision with regard to your manuscript MSB-13-5058.

We appreciate that you describe a rapid and highly efficient protocol for differentiating neural cells from induced human stem cells and we do agree that the methodology in itself is of interest.

We also recognize that this experimental system enables time course transcriptomic analyses due to the high homogeneity of the differentiated (and, potentially, differentiating) population. We are thus pleased that you would consider extending the study with a more systematic analysis involving combinatorial perturbations of key regulators and measurements of the resulting response of the regulatory network. If such investigation would provide further causal insights into the core regulatory network underlying the differentiation process and its surprisingly fast and robust dynamics, we agree that it could potentially represent a significant addition to the manuscript. One central issue is perhaps to provide clues on whether the particularly robust differentiation process, as compared to previous protocols, is mostly due to the efficient repression of stem cell factors (e.g. by the inhibition of STAT3 regulation) or rather explained by a particularly strong activation of a neuronal differentiation transcriptional program.

Another major point raised by both reviewers is the fact that the analysis is currently only limited to known pathways. We understand that IPA provides a convenient way to analyze the data and describe the results in the light of known molecular and regulatory interactions. However, as you nicely highlight in your letter, your protocol provides a unique opportunity to examine the profiles of gene expression during the differentiation process in an unbiased way and at a genome-wide scale. Therefore we feel that an unbiased analysis of the transcriptomic data should also be performed and would complement the currently presented IPA 'filtered' analysis. It would also fit particularly well the scope of the journal.

Besides these major points, we are very grateful that you are considering clarifying some of the aspects of the text and we would like to emphasize that we can remain very flexible in terms of format and size limitations given that our journal is online-only.

We hope the points above are reasonable and we remain, of course, available for further discussion over the phone.

2nd Editorial Decision

07 February 2014

We have now received a late report from the third reviewer whom we initially asked to evaluate the study. As you will see below, the general points raised by this reviewer are rather similar to the other referees' comments. Given the overall recommendations and the substantial points raised, I am afraid that we cannot revert our decision.

With regard to the scope of the possible resubmission of an extended study, we would refer to our letter of 31 Jan 2014.

Reviewer #3:

Summary:

This manuscript describes a novel differentiation protocol that yields pure but immature neuronal populations after 4 days of induction (over-expression of two neurogenins), and more mature neurons after 14 days of induction. This seems to be a significant improvement over existing protocols, in terms of speed and purity. The latter half of the manuscript describes RNA-seq profiling of the cultured cells at various stages of the differentiation process. The transcriptomic data are used to deduce the most similar *in vivo* brain regions and developmental stages, and to infer the regulatory network that lies at the heart of the differentiation process.

General remarks:

One general point of frustration is that bioinformatic methods are poorly explained. In most cases, the main text and figure captions present only a very high-level description of the methodology. As a result, it is hard to judge the validity of the conclusions. Even the Materials and Methods section is vaguely worded, and decoding it requires specialized knowledge plus a bit of guesswork. For example, readers who are not familiar with Ingenuity Pathway Analysis (IPA) software would have no idea what to make of the network-related sections of the manuscript. Other bioinformatic portions of the manuscript are similarly unclear.

In general, the bioinformatic methods are ad hoc, and the conclusions do not seem robust (see below). Since validation experiments are not presented, this is a major concern.

Major points:

The analysis of iNGN similarity to human brain transcriptomes has issues. Since BrainSpan RNAseq data are derived from heterogeneous tissues, they are not ideal for inferring the neuronal subtype of cells cultured *in vitro*. The authors assume that the Pearson correlation coefficients (PCCs) shown in Figure 3 are indicative of neuronal subtype, but in reality they are probably more indicative of the relative abundance of neurons/progenitors in any particular brain region.

Another problem with this section is that Figure 3A lumps together all of the samples derived from a single developmental stage. It's not obvious how this is meaningful. At best, day 4 iNGN cells would be similar to neurons/progenitors in one specific brain region (cerebellar cortex, for example), or one subset of brain regions. This similarity could be obscured when one averages over all brain regions.

Perhaps for the reasons listed above, the results in Figure 3A look a bit surprising. Day 4 iNGNs show greatest similarity to the earliest analyzed developmental stage (8 pcw), so the obvious conclusion would be that 8 pcw brain (or some earlier time point that was not included in the analysis) is the closest *in vivo* match. But then we see that day 0 iNGN cells, which are pluripotent, also show high similarity to 8 pcw brain (PCC=0.73). This makes no sense, because there are obviously no pluripotent cells in the brain.

Figure 3B again uses an odd metric for matching iNGN cells to brain samples, and as in Figure 3A, no explanation is provided for why this metric would be a natural choice. In 3B, the PCC of a brain region is compared to the PCCs of other brain regions from the same developmental stage (z-score). It would seem more natural to use the PCC itself, rather than the z-score of the PCC. By the z-score metric, day 4 iNGN cells look like cerebellar cortex prenatally and mediodorsal thalamus postnatally, which is a peculiar switch. It is claimed that this is plausible because neurons migrate during brain development. However, this explanation can be used to justify more or less any result. Do cerebellar cortical neurons migrate to the MD nucleus of the thalamus? The other justification is also unconvincing: that the cerebellum and thalamus are functionally connected. The MD nucleus is actually connected to a very large number of brain regions - the cerebellar cortex is hardly unique in this regard.

I strongly suspect that other ways of analyzing the BrainSpan data would yield different results. A more natural approach would be to combine Figures 3A and B into a single figure: a heat map of PCCs that has brain regions along one axis and time points along the other. Bi-clustering on such a heat map should reveal brain affinities in a more natural way. Of course, there is still the problem of tissue heterogeneity. Perhaps this could be partially mitigated by replacing expression values with fold-changes (or the log of fold-change, to prevent a small number of highly-expressed genes from dominating the PCC). Each tissue could be represented by its fold change over the median of all tissues, and iNGN cells could perhaps be represented by the fold change from day 0 to day 4. Also, it would be good to see some positive and negative controls, to increase confidence in the ad hoc methodology used in the BrainSpan analysis.

The section in the main text entitled "A modular deterministic gene regulatory network drives the rapid neurogenesis" is so terse that it is unintelligible. The corresponding portions of the Materials and Methods are also elliptical and lacking in detail. As far as I can tell, the gene regulatory network was inferred from RNA-seq data by starting with the output of a software program (IPA) and then applying a series of ad hoc filters. Subsequently, the network was "manually curated," but there are no details on what this manual curation step involved. Network inference is a notoriously difficult problem. I would like to see more details and more justification in the Materials and Methods (including key aspects of the IPA algorithm and database) before placing much faith in the network models shown in Figure 5.

More generally, it's important to emphasize that the networks in Figure 5 and Figure 6 are just models. They may indeed be plausible and partially consistent with the literature (the IPA database is partially literature-derived). However, they have not been validated in iNGN cells. The portion of the manuscript entitled "The loss of pluripotency" needs to make this point clear. The same goes for subsequent portions of the manuscript. As written, they sound like a list of definitive conclusions, but they are actually only a list of bioinformatic hypotheses with moderate or marginally significant p-values and no validation. Moreover, all such network models are incomplete - they have many missing nodes and missing edges, and one should exercise caution in drawing inferences from them.

One specific concern in the network analysis is that SOX2 is held up as a key player in the neuronal differentiation cascade, even though this TF did not score well in IPA analysis. If the authors adhered strictly to their numerical cutoffs, would it even be part of the network?

Minor points:

On line 201, it is stated that BrainSpan data cover the "full course of human brain development." While this is technically true, it gives the wrong impression because the authors did not analyze the earliest BrainSpan samples. They actually started their analysis at 8 pcw, which is quite late in brain development (embryonic day 15.5 in mouse). It would have been better to include earlier developmental stages in the analysis.

The regulatory network is repeatedly described as "deterministic," but it's not clear what this word means in this context, or what the evidence is. What would constitute a non-deterministic network?

Line 265: which enrichment test is referred to here?

Line 696: what is an "enriched regulon?" Which list was filtered?

Line 698: what is the rationale behind these keywords?

Line 700: was the enrichment p-value corrected for multiple testing?

Do the red and blue colors in Figure 5 represent neuronal and pluripotency factors?

Lines 707-710: the logic of the global regulator steps is unclear. How are global regulators defined? What is the rationale behind these processing steps?

Supplementary Tables 2 and 5 are formatted in a manner that is difficult to read - rows are split up over multiple pages.

Line 687: what is the meaning of "predicted data?" Are the predictions credible?

Line 688: Table S3 seems to be about miRNAs, not upstream transcription factors.

Lines 728-730: I'm not sure the hypergeometric test is valid here, because the BrainSpan RNAseq data show intertemporal correlations.

Re-submission

20 June 2014

(see next page)

Please find below point-by-point responses to the comments of the three reviewers for manuscript MSB-13-5058R-Q. We have interspersed our responses within the referee reports, in bold font. We thank all reviewers for their insightful comments. We performed new experiments, added new figures and rewrote the manuscript to address these comments.

Reviewer #1:

Summary

In this manuscript, Busskamp et al describe a new method for induced differentiation of human iPS cells by overexpression of two neurogenin transcription factors, rapidly generating bipolar neurons at very high yields of purity. The paper addresses the important question of directed differentiation of stem cells. For the particular case of neurogenesis, this question has considerable potential in investigating fundamental brain functions but also application into fields in regenerative medicine. The authors analyze the morphology and functional properties of these cells and observe that the differentiation protocol leads to competent neurons able to reach maturation. They then perform transcriptional characterization for a time-course of differentiating cells through RNAseq and miRNA profiling analysis. Comparison of transcriptomic data with existing human brain reference data (the BrainSpan atlas) shows high correlation between induced differentiated neurons and prenatal cerebellar and adult thalamic neurons. The authors then use their data, as well as an existing knowledgebase to try to elucidate the core transcriptional regulatory network of differentiation, including both transcription factors and miRNAs. Finally, the robustness of the differentiation process is tested by perturbation of two known regulators of neurogenesis (NEUROD1 and REST).

General remarks

The strong point of this study is, in my opinion, the establishment of a rapid and efficient protocol for the differentiation of bipolar neurons that circumvents current limitations of differentiation from ES and iPS cells or transdifferentiation from fibroblasts, such as low purity yields or suboptimal protocols involving a large number of steps. The final section of the study also indicates that modifications of this system should provide a good basis for directed differentiation of other neural cell types.

We would like to thank the reviewer for the acknowledgement of the iNGN cell differentiation protocol. We also feel that the ease and efficacy of the protocol will enable many subsequent studies on the differentiation and function of diverse classes of neurons, and we note that we have already shared the cells and protocols with a number of labs who are successfully replicating our work and using the iNGN cells for their own work.

However, I have major reservations regarding the subsequent transcriptomic analysis and gene regulatory network construction sections of the study, and find the results insufficient to substantiate some of the author's main claims. In particular, the study does not provide sufficient detail on mechanistic aspects of network dynamics to warrant the conclusion that the regulatory network underlying the neurogenin-induced differentiation of iPS cells is deterministic, nor does it sufficiently explore network principles underlying robustness of the system. The computational methods are also not clearly described which, together with some confusing sections of the results (both text and figures) and rather brief figure legends, makes it difficult to follow the manuscript and interpret the findings. This may be partly due to space limitations but nevertheless is a major issue.

We appreciate the reviewer's candor, and after reading the extensive review, we have identified and eliminated the weak points of the manuscript. We also apologize for the brief and/or vague descriptions that apparently left our main claims enigmatic. Thus, we have carefully edited the manuscript to clarify the bioinformatics methods and the text in general, and also focused more on strongest points of our work. First, we toned down some claims, such as the deterministic properties of the network analysis. We also changed the title to "Transcriptomic basis of rapid neurogenesis in human stem cells". Second, in our new Figure 2, we provided more detailed analysis of the transcriptomic upregulation of the synaptic machinery. Third, we improved the discussion of the up- and downregulation of biological processes, as suggested by GO term analysis, and this now indicates many similarities with developmental steps also found *in vivo* and the brief presence of neural progenitor states (Figure 3 and Supplementary Figure 4). Furthermore, we refined and improved the BrainSpan analysis (Figure 3), following valuable suggestions from the reviewers. Fourth, we improved the description of the analysis of transcription factors contributing to the differentiation, and softened the claims of the "deterministic" nature of the regulatory network (Figure 5). We improved the text, figure legends and methods intensively towards the readability and comprehension of our findings. Fifth, we updated and validated the NEUROD1-shRNA knockdown by measuring expression levels of NEUROD1 and some targets. In addition, we performed a siRNA knockdown screen against several transcription factors that we identified in our regulatory analysis (including NEUROD1, NEUROD2, POU3F2 and ZEB1) and combinations thereof (NEUROD1+NEUROD2 and NEUROD1 and PAX6). As proposed by Reviewer #1, we measured downstream targets by qPCR. Since expression changes were detected as expected, this provided some validation of these transcription factors in the network we show. The interventions did not impede neurogenesis but altered the morphology of the neurons and thereby highlight the robustness of the regulatory program (Figure 7, Supplementary Figures 10 and 11).

Thus, we hope that the revised manuscript and the additional experiments convince Reviewer #1 of the veracity and overall importance of the systems level transcriptomic analysis of our neuronal differentiation from stem cells.

In many instances it is difficult to tell to what extent conclusions drawn are supported by the authors' own observations or are based on information taken from other published studies.

We agree that the data presentation in the text made it difficult to assess if our conclusions were based on our work or previous work. Thus, we drastically modified to text to emphasize the results from our work and how they support our claims.

As the authors say in the discussion, the data provide a valuable molecular blueprint for neurogenesis, and the cell model offers a platform for further directed differentiation of other neuronal cell types. However, the transcriptomic and computational results of the study, albeit comprehensive and interesting as a descriptive resource, are disappointing. Because of the extensive utilisation of pre-existing data, relatively little novelty emerges in terms of biological insight into mechanistic aspects of the regulatory network, and the methods used are not novel in themselves.

Reviewer #1 raises an important point here about novelty in the analysis, and we have gone to great lengths to highlight the exciting novel aspects of our analysis. It indeed is true that our analysis used Ingenuity's IPA database, which contains tens of thousands of experimentally validated transcription factor-gene interactions. IPA provided a valuable resource to help identify transcription factors that are activated or suppressed specifically in our differentiation process (which should differ from the regulatory programs seen in the differentiation of other cell types, and even different neuron types). Ideally, it would have been nice to employ network inference algorithms to predict novel regulatory interactions, but unfortunately we did not have adequate amounts of data on our iNGN cells at this time to do so reliably. However, while the individual interactions among transcription factors are not novel, the combination of transcription factors used in the regulatory program in our cell lines is completely novel and has not previously been identified during neuronal differentiation from human stem cells. Furthermore, these are important to characterize, since the transcription factors in its regulatory network will be important points where one could modulate to obtain different neuron types in the future (as we start to show with the new experimental work added in our revised manuscript). In other words, the cellular context is different from the source of the pre-existing data but we still succeeded in generating a comprehensive network in the iNGN context. Furthermore, we are excited to also present for the first time to our knowledge, the first comparison of a newly differentiated neuron cell culture to the Allen BrainSpan dataset, thus providing a spatio-temporal view of how a cell line compares to the human brain. We focused on these points in the revised manuscript and below find a summary of our novel findings:

- 1. Neuronal differentiation from stem cells is extremely efficient by transcription factor induction without any additional bioactive factors in the culturing media (Figure 1, Supplementary Figures 1 and 2)**
- 2. Neuronal maturation and electrical activity need additional extrinsic factors even if the synaptic machinery is expressed within 4 days in stem cell media (Figure 1 and Supplementary Figure 2)**
- 3. The iNGN cells differentiate rapidly and continuously via unstable progenitor states and NOT directly as frequently proposed (Figure 3 and Supplementary Figure 4)**
- 4. The BrainSpan analysis assigned iNGN cells higher correlations to prenatal human tissues. Our analysis also shows that the likelihood that iNGN cells resemble cortical neurons is very low (Figure 3)**
- 5. MicroRNA levels also dynamically change from stem cell profiles (miR-302 cluster) towards neuronal ones (miR-124, -96 and 9). Our analysis suggests that microRNA regulation takes place mostly downstream of the neuronal differentiation initiation phase (Figures 4 and 6, Supplementary Figures 5, 6, 8 and 9)**
- 6. We identified a network of transcription factors that contribute to the differentiation, and validated several central factors regulating neuronal genes by individual and combinatorial perturbations (Figure 5 and Supplementary Figure 7)**
- 7. We present the coding and non-coding transcriptomic blueprint of a neuronal differentiation program from human induced stem cells. Our data primes further targeted manipulations of iNGN cells and/or the usage of different transcription factors cells to increase the variety of human neurons. Notably, we found most transcription factors that are currently used also activated by the neurogenins explaining why most protocols result likely in similar neuronal cell types, i.e. excitatory neurons (Figure 2)**

We hope that the improvements in our text clarify the novelty of the various portions of our study and the accompanying analysis.

As a result I feel the study will be of great interest to the iPS and neural audiences, but of less interest to the systems biology and computational communities. In my overall assessment of the manuscript I have tried to balance the value of the iPS differentiation model and accompanying global transcriptome data against the less compelling regulatory network analyses, bearing in mind the likely audience for Molecular Systems Biology.

We decided to submit our work to Molecular Systems Biology because of focus it puts on high-impact biology, which is supported by omics data analysis. This venue also provides the opportunity to merge the iPS and neural communities with systems biology to demonstrate that fundamental biological processes in differentiation can be revealed by global transcriptomic analysis. Thus, using

some systems biology tools, one can begin to elucidate the molecular events over the time course of neuronal differentiation. This is particularly important since previously, a detailed analysis of stem cell-derived neurons in the course of differentiation on the molecular systems level was out of the technical reach. Now we show that we could overcome this by applying iNGN cells and we point out that a molecular systems view will likely be the key for targeted generation of neurons rather than the current approaches in the stem cell community that often “cherry-picks” a few genes for the characterization of differentiated cells.

Major comments

For gene regulatory network construction, the authors have used a specific functionality of the Ingenuity Pathway Analysis software (upstream regulator analysis). From my understanding, this method is heavily based on Ingenuity's own knowledgebase regarding previously described regulators and interactions. In a sense, the analysis makes sense of the experimental observations in the context of what is already known in the literature, and stored in the database. Thus, the constructed network will inevitably be limited to existing knowledge and therefore, by default, novel regulatory interactions that might stem from the author's datasets are likely to be discarded or discounted. What is gained in conformity with the literature is lost in terms of potential novelty.

We acknowledge that the wording in our previous submission could have evoked expectations that were not met and have remedied this by correcting the wording, adding additional analysis and experiments and modifying the focus of the paper. The unmet expectation suggested by the reviewer was the expectation to discover novel interactions, as could be obtained given enough data and gene regulatory network inference methods. Given the limited amount of data obtained from our cell lines, we instead used the well established functionalities built in Ingenuity's IPA database. Although individual interactions have been described, the pathways these interactions make up are completely novel, and successfully describe some of the major gene regulatory processes contributing to the rapid and homogeneous differentiation of our iNGN cells. Through further experimental validation, we tested several of the transcription factors to test their contribution. We are excited to report that perturbing these factors successfully changes the morphology and expression of downstream factors, thus opening up the possibility to further perturb these cells to obtain different types of neurons. To avoid the confusion in the previous submission, we have extensively modified our manuscript to clarify the important points in our study. We also changed the title and the discussion of the transcription factors.

In the methods section, the steps for calculating activity Z-scores, crucial for reaching the presented core regulatory network, should be made clearer.

We updated the methods to improve clarity. Furthermore, after we had previously submitted our work, the details of the IPA algorithms were published (see Kramer et al, 2014). We have now cited this manuscript, thus providing further details on the algorithms in their software.

The steps for filtering the list of enriched regulons are not entirely clear, and in some cases there seem to be inconsistencies with the presented figures. In particular, the authors state that regulators were removed if there was a discrepancy in differential expression and activation site, but in figure 5B, for instance, ISL1 is included even though it significantly increases expression from day 0 to day 1 but has negative activation Z-score. In general, I found the filtration protocol to be excessively supervised (for instance, the authors set an absolute activation/repression cutoff but then open exceptions for SOX2 and SMAD1) including a final manual curation step. The analysis is already constrained by Ingenuity's stored information and the filtering further reduces the likelihood of discovering novel regulators/interactions.

Indeed, the details for filtering the network were not clear enough, and at first glance it appeared that there were some inconsistencies and exceptions. However, a systematic approach was taken with no deviations, and apparent exceptions and inconsistencies had resulted from ambiguities or typos in the methods section. For example, it looked like that SOX2 was included while it seemed to not score well in IPA. SOX2 contributes its pluripotent role also in a complex with NANOG and OCT4. When the targets of the complex are considered, SOX2 scores quite well. Unfortunately this score had been erroneously omitted in the generation of the supplementary table and figure, but we have now corrected this omission. We show the complex scores in updated Figure 5A. Similarly, SMAD1 exerts its function in the context of a complex, as shown in the network with it being directly associated with EP300 and CREBBP. We note that as shown in the supplementary table, all three members were within our thresholds. The apparent discrepancy for ISL1 was not an actual discrepancy, but stems from a misinterpretation of the methods. That is, our algorithm retained it because it was consistent by day 4 when the gene expression was highest. We have rewritten this step in the analysis to correct this misunderstanding. We greatly appreciate how Reviewer #1 carefully read our methods and caught these problems, as it informed us how to improve the description of our approach. We have now rewritten much of our methods for this section to improve the description of further details of our analysis, and hope this eliminates any other possible misunderstandings that might arise. Thus, we hope that it is clearer how we identified key regulators that contribute to this rapid robust differentiation

Beyond the methodology, the analysis and interpretation of the regulatory network results is also not entirely clear to me. In particular the way results are described, for instance in the loss of pluripotency section of the results, makes it unclear what is predicted from the analysis and previously known from the literature.

Indeed, our data presentation was confusing. We significantly modified the section that discusses key regulatory proteins, emphasized the novel features or our network, and generally increased the readability of this section. Furthermore, the additional siRNA we conducted now adds more validation.

As far as I can see, the regulatory links discussed are putative links, based on the proposed network, and "targets" refers to putative targets based on the Ingenuity database, most likely in unrelated cell types (e.g. Line 266: "over-representation of differentially expressed targets"). The authors then reference published work that supports the existence of such a link in other systems. However, the text as written rather suggests (no doubt unintentionally) that these putative regulatory interactions can be clearly seen in this study, rather than that the proposed network links are simply consistent with other published data. For example, I could not see sufficient data in this study to support the statement that "By day4, the sequestered p300/CREBP complex fails to activate FOXO1" Similarly "Our data suggest that neurogenins directly inhibit SOX2" (line275) - how have the authors drawn a conclusion about direct versus indirect regulation?

We have removed much of this text in our revision, and throughout the text have emphasized where the interactions were putative.

This mixing of information is made worse by a couple of instances where there is a discrepancy between prediction and existing knowledge, such as where NEUROG2 is referred as activating NEUROD1 after binding a p300/CREBBP complex, but in figure 5A this interaction is not represented. Furthermore, it is not clear if the timing for the regulatory effects is inferred from the analysis, the literature or both, or how the labels "immediate repression" and "secondary repression" were derived. In order to highlight the value and novelty of their observations, the authors should first state what predictions are specifically produced by their computational analysis and then put them into context with existing literature.

The timing claims were based on our data, but given the changes in the focus of our work, we omitted these parts from the text and figures.

Some broader, strong conclusions are drawn based on the proposed network, such as (line 288) "from our data a secondary mechanism arises in which stem cell factors are repressed primarily by inhibition of STAT3 regulation". In the absence of any functional validation experiments, I do not feel this conclusion - so forcefully stated - is supported by the data.

We agree that this was overstated and we changed the text as following: "Additional indirect interactions could further repress stem cell factors through NEUROD1, p300/CREBBP, STAT3, SPARC, FOXO1, and others, as suggested by our analysis (Figure 5A and Supplementary Text)".

The conclusion to this section (lines 313-315) is, essentially, that neuronal differentiation from iPS cells involves the repression of stem cell transcription factors and activation of neuronal transcription factors. This is not a particularly surprising or novel conclusion in itself, and reinforces the need for the authors to validate at least some of the regulatory interactions that they propose, preferably ones that are novel at least in this cell type even though previously described in the literature. Since the study already utilises lentiviral expression of shRNAs this approach would not be unduly onerous or time-consuming.

We agree with Reviewer #1 that our findings of repression of stem cell factors and activation of neuronal transcription factors during iNGN differentiation *per se* are not surprising. Yet we have identified specific factors that are likely involved in the rapid and homogenous differentiation seen here and we tested and validated contribution of these regulators by reducing their expression levels by siRNAs. We measured the levels of the siRNA targets as well as their downstream-regulated genes by qPCR (Figure 7G and Supplementary Figure 10 and 11). Downstream gene expression changes were in line with expected changes according to our gene regulatory network. Furthermore, all manipulations resulted in higher numbers of non-bipolar neurons; hence they changed the morphology of siRNA-transfected iNGN cells, further demonstrating their contribution to iNGN differentiation (Figure 7H and Supplementary Figure 11).

After constructing the core regulatory network, the authors do perturb expression of two known neurogenesis regulators and observe that differentiation is still possible, although with some morphological differences in the obtained neurons. Although this suggests robustness of the network, the study does not sufficiently explore the impact of the perturbations in the topology and dynamics of the network to support that claim.

The authors provide a possible explanation by retrospectively using the constructed network and the list of putative targets for REST and NEUROD1 for hypothesizing why differentiation was still possible. Ideally, a more comprehensive and detailed assessment of network robustness would be to obtain genome-wide transcriptomic data upon perturbation and compare the constructed networks. Given the scale of these experiments, a less costly and time-consuming alternative would be to assess whether the expression of a reasonable subset of putative REST and NEUROD1 targets is - at least initially - affected by the perturbations, and perhaps functionally test some of these targets.

Here we agree with Reviewer #1's points and we aimed to strengthen our claims on the robustness and manipulations of the network. We added qPCR data for NEUROD1, NEUROD2, SLIT2 and SOX2 in shNEUROD1-iNGN cells over the time course of differentiation. The shRNAs prevented the expression of NEUROD1 until day 3 and on day four, NEUROD1 levels were at ~22% compared to normal levels. Still, these cells differentiated to neurons. Based

on the IPA analysis, SLIT2 was identified as a unique NEUROD1-regulated gene that was not controlled by other neuronal transcription factors. SLIT2 is highly expressed in stem cells (day 0, Figure 7C), suggesting NEUROD1-independent regulation at the stem cell state. During the course of differentiation its expression levels initially dropped and started to increase again on day 3 when most-likely NEUROD1 activated SLIT2. We measured a significant reduction in SLIT2 expression levels in the shRNA knockdown iNGN cells. NEUROD2 and SOX2 were also identified as NEUROD1 downstream-regulated genes but other regulators in iNGN cells share were also involved in their regulation (e.g., the Neurogenins). Indeed, expression levels were not significantly changed (Supplementary Figure 10) suggesting the involvement of other transcription factors.

Furthermore, we moved the REST manipulations to the supplement. The increased soma sizes were of statistical significance but it is unclear how much it would be of a biological significance. To test all potential REST-regulated genes would be quite laborious and a recent publication also indicated beneficial functions of REST in adult neurons regarding the age of neurons (Lu *et al.* Nature 2014). Outside of the scope of this manuscript, studying REST functions within iNGN cells would be an interesting story by itself.

Lastly, we expanded the number of perturbations to include siRNA knock-downs of several additional pro-neural transcription factors. Again, these led to morphological changes and changes in down-stream gene expression.

In addition, the authors suggest that the limited impact of the NEUROD1 knockdown may be due to compensatory upregulation of NEUROD2. Before proposing this, the extent of knockdown achieved for NEUROD1 should be determined and shown, at both the RNA and protein levels.

We are sorry for the misunderstanding but we assigned NEUROD2 redundant functions for NEUROD1 in the sense of taking over the regulation of NEUROD1 regulated genes as it has been shown *in vivo* by Cherry *et al.* J Neurosci 2011. We did not claim any compensatory upregulation and our added qPCR data (Supplementary Figure 10) clearly shows that it's not the case. We would also not expect a distinguishable difference on NEUROD2 protein levels.

Furthermore, since NEUROD2 is downstream of NEUROD1 (Fig 5B), the expression of NEUROD2 should be assessed in the NEUROD1 knockdown cells, to confirm that it is still upregulated and that the proposed compensation is plausible.

NEUROD2 is still expressed in sh-NEUROD1 iNGN cells over the time course of iNGN differentiation (see Supplementary Figure 10).

In the title, and several sections of the text, the authors describe the regulatory network as being deterministic, and it is not clear to me in what respect this is meant. In the first instance, the differentiation process can be seen as deterministic

given that the final outcome in around 90% of the cases is the same (bipolar neurons). However, as in my previous comment, my impression is that the authors depart from this empirical observation to make a general comment about the network. In this respect, I do not think the study has enough single-cell resolution to comment upon the deterministic/stochastic mechanistic nature of the network. In particular, it is not possible to ascertain whether the behavior of pluripotency and differentiation regulators is homogeneous and coordinated among cells at the different stages, or instead individual cells show heterogeneous expression of regulators suggestive of different paths of differentiation upon neurogenin overexpression. Having narrowed down the number of regulators in the core network (and established a basic architecture), it would now be interesting to assess (for instance by single-cell RT PCR) the level of heterogeneity of these genes in single cells, at different stages of differentiation (days 0 - 4) and relate these observations with the current picture. In such a study, Buganim et al have profiled single cells in the context of a reprogramming experiment, describing an early stochastic stage with large variability, followed by a late hierarchical phase coordinated by key regulators (Buganim et al. 2012. Cell 150: 1209-1222).

We apologize for the confusing usage of the word “deterministic” and we omitted it from the title and text. Indeed, we used “deterministic” because the iNGN cells homogeneously differentiated to bipolar neurons. We agree with Reviewer #1 that single cell analysis would be nice to have to assess if the regulatory changes we propose are coordinated among the cells, but again, this would be a manuscript by itself. Hence, we discuss the benefits of single-cell analysis either by RNAseq or FISSEQ. Notably, we submitted the iNGN cell line to the ENCODE project and the results of all their assays (unfortunately not single cell analysis) will be published in the next years and thereby increasing the data of these cell line.

Minor comments

As a general comment, the figure legends are too brief and lack the detail required to enable the reader to easily interpret the figures. More specific recommendations on the text and figures are detailed below:-

We apologize for the brevity. All figure legends were updated and expanded upon.

Figure 2D, it would facilitate interpretation if labels for the different classes were presented in the same color as the dashed boxes (e.g. "cell adhesion" in orange, "RNA metabolic processes" in green, etc). Also the explanation for the selection of the genes in the inset panel should be given in the legend as well as the main text.

We would like to thank Reviewer #1 for these suggestions. We changed the background shades of former Figure 2D (now Figure 3A) to improve the interpretation. The genes in the inset were also referenced to in the main text

and represent all genes in the associated GO class that were differentially expressed in the iNGN cells.

Fig 3A:- the p-values for the Day 0 cells should also be added to the plot.

The p-values of Figure 3A (now Figure 3B) are for the comparison between day 0 versus day 4. We have now clarified this in the figure legend.

Fig 4A legend: The meaning of "Distribution of microRNA quantities" would be clearer if worded as "relative abundance of microRNAs" or similar.

We changed the figure legend accordingly.

Fig 4C legend: At which timepoint are these the 5 most highly expressed miRs? Also the sentence needs to be reworded for clarity.

We show here the 5 most highly expressed miRNAs at day 0 (miR-302a/-302b/-302c/-124x/-92a) and at day 4 (miR-124x/-96/-92a/-19b/-302a). For simplicity we changed the text to "Dynamic miRNA changes of representative miRNAs during the differentiation of iNGN cells."

Fig 4D,E legend: please define "significantly"

We added "q-values < 0.05" to the legend text, also to Supplementary Figure 5.

Line 243 - please indicate here the magnitude of the increases observed in miR-96 and miR-9.

We changed the text as following: "We also observed increases in the abundance of the neuronal miR-96 (10-fold) and miR-9 (57-fold) (Figure 4B and E and Supplementary Figure 5) among others (Figure 4C)".

Line 245: the increase in miR-9 is linked in the text to Fig 4B but it seems to feature only in Fig 4E

This is correct; we changed the figure reference.

Lines 253-261: the authors should include some more detail here of the approach used. In particular, Ingenuity Pathway Analysis should be mentioned somewhere other than just the methods section. Key, and non-obvious, aspects of the filtering should be briefly outlined here, in particular that regulators were removed if there was a discrepancy in their differential expression and putative activation state or if they were not connected upstream to the neurogenins.

We agree with Reviewer #1's comment and changed the text to: "Thus, we analyzed the time-course of mRNA expression data in the context of known

transcription factor interactions in Ingenuity's IPA database (See Materials and methods). To identify potential regulators, a standard and non-neuronal biased enrichment test (Kramer et al, 2014) was conducted to identify transcription factors that had an overrepresentation of differentially expressed targets, and had their targets changing expression in the direction consistent with the activation and repression activities of the transcription factors of interest (Supplementary Table 5). We focused here on a network of transcription factors that met these criteria and that were also connected the Neurogenins through direct and indirect gene regulatory interactions that had been validated in other cell types and/or organisms, as catalogued in the IPA database".

Fig 5 - some of the plots are labelled as z-scores, not activation z-scores. Presumably this is an error or due to lack of space in the figure. If all plots are activation z-scores then it would be better to put this into the legend and remove it from all the plots. The figure would then be a little less cluttered. It is also counter-intuitive that the key describes the colours in terms of immediate and secondary repression but many of the interactions are arrowheads, which conventionally denotes activation, so this apparent contradiction should be explained briefly. There is a typo in the figure labelling of Fig 5B (progenator instead of progenitor).

We would like to thank Reviewer #1 for his suggestion. We omitted "immediate and secondary repression" from the Figure 5 and text and corrected the typo. All z-scores represent activation z-scores. We deliberated your suggestion to remove the labels from the plots but we decided to keep the axis labeling for consistency and clarity.

Fig 5 legend: - the figure title refers to a "core gene regulatory network" and this phrase is then used repeatedly in the text, along with "core transcription factors" (e.g. line 318). Presumably this refers to the combination of Figs 5A and 5B but this should be clearly stated and the authors should check that this term is then used consistently to refer to the same set of interactions and/or regulators. The legend should also remind the reader what the total set of neuronal genes is as used in fig 5B (ie how many there are and how they were defined). Overall the legend should include more information to facilitate interpretation (e.g. it's not immediately clear to the reader that blue and red circles mean inactivation and activation regulators in 5A) and the use of colours (orange, blue and grey) for the various connectors in the network in fig 5B should be explained.

We have gone to great efforts to address these issues and hope that the figure caption is clearer. We have also simplified the color scheme, clarified the source of the neuronal genes, and avoided the use of "core" when referring to the transcription factors.

Line 290:- "In summary, our pathway analysis revealed connections of..." Throughout the manuscript it is unclear whether regulatory relationships (ie

between a transcription actor and a target gene) are putative, based on the proposed network, or well-established in the literature. In places this is understandable but I would strongly argue that - in the absence of experimental validation - the authors can only claim that their pathway analysis reveals putative or proposed connections of Neurogenins with repression of stem cell factors and these terms should be used wherever relevant. Alternatively, such comments could be prefaced with "In our model/network...".

It is quite important to distinguish whether the regulatory interactions are validated or putative. In their database, IPA includes experimentally validated interactions with their associated publications. Furthermore, we provide some additional support through our validation experiments. We have modified the text to address this concern.

Line 305:- I found the sentence "Our transcriptomic data shows increasing inhibition of REST regulation" a little unclear. Presumably "regulation" here refers to REST activity (inferred from expression of its targets, and as shown in fig 5B) or does it actually refer to inhibition of transcriptional regulation of REST? The transcriptomic data can really only show the latter; it has to be combined with the network analysis in order to derive the information in fig 5B. This should be clarified.

We agree that this section was unclear and therefore we shortened the text to: "Furthermore, inhibitors of neurogenesis were repressed, including HES1 and REST (p < 0.003; Figure 5B)",

Fig 6 legend - the legend should state that the miR interactions (in black) have been superimposed on the previously generated regulatory network (in grey); because of the amount of information in the figure, many of the grey connectors appear to be coming from the miRs. Also, the key in the figure (upregulated/downregulated) should be boxed or similar to make it more obvious to the reader.

We changed the corresponding figure legend according to Reviewer #1's advice to: (A) Validated transcription factor targets for differentially expressed miRNAs were identified from miRTarBase. The miRNA-interactions (black) have been superimposed on the previously generated regulatory network (light grey). Most interactions involved upregulated miRNAs that suppress stem cell factors in our network. Inset plots show cases with significant anti-correlation between miRNAs (green) and their transcription-factor targets (black)".

Fig 6B:- presumably the 1295 neuronal genes in the pie chart are the same set as used in Fig 5 but, if not, this needs to be stated.

We have added the number to the caption for figure 5 and updated the figure legend to: "(B) Neuronal transcription factors in our network were identified

by looking for a significant overrepresentation (hypergeometric test; $q < 0.05$) of neuronal genes among their known target genes (using a list of 1295 neuronal genes based on Gene Ontology). The fraction of neuronal gene targets for each transcription factor is shown in the pie charts, with the significance of overrepresentation of neuronal genes shown in with color intensity”.

Line 326:- "More neuronal microRNA targets underwent downregulation...". More than what? Non-neuronal targets of the same miRs?

This is now clarified in the text; we intended to mention the targets of miR-124, -96 and -9.

Line 338"- "The broad target range..." I don't quite understand what point is being made here, or how it explains why only a few microRNA/validated target pairs were detected in the core network

Thank you for catching the typos. We have now deleted this sentence since it was a fragment of an earlier manuscript version.

Line 348:- again, in the absence of experimental verification of microRNA interactions in this cell system, it would be more accurate to say that the changes in microRNA expression are such that they would/could assist the core TFs in repressing stem and neural progenitor factors.

We changed the entire paragraph accordingly. The miRNA targets were taken from a database of validated interactions. In respect of not experimentally testing individual miRNA/target interactions, we phrased our claims more carefully, although interactions are likely if a mRNA target (RNAseq) and its corresponding miRNA (n-counter/qPCR) were expressed in iNGN cells.

Fig 7:- I missed any information regarding the extent of the knockdown achieved for NEUROD1 at the RNA and protein level. Without this information, the simplest explanation for the failure to arrest neurogenesis is simply that the knockdown was insufficient. This information should be added to Fig 7 or the supplementary information.

This information is added to Figure 7B.

Line 384:- "19% of the neuronal genes were under REST or NEUROD1 regulation". In the context of discussing the REST and NEUROD1 perturbations, at first sight this suggests that the expression of these genes was seen to be affected by the perturbations. This text should be altered but, in any case, as mentioned earlier, the authors should undertake some gene expression analysis of the cells as soon as possible after the perturbations to try to verify some aspects of their network.

Relevant gene expression verifications are added to Figure 7 and Supplementary Figure 10. Also, the paragraph is altered in the text to improve the clarity.

Line 394:- "...small changes to the accessibility of a minority of neuronal factors".
I don't understand what this sentence refers to, or what data it is based on.

We have now omitted this sentence.

Reviewer #2:

The manuscript by Busskamp et al analyzes developmental program driven by Neurogenin1/2 that converts pluripotent human stem cells to nerve cells. Similar to previous report (Zhang et al, 2013), the authors demonstrate efficient conversion of human stem cells to neurons by inducible expression of Neurogenins. Expression analysis of programmed cells on day 1, 3 and 4 revealed progressive changes in cell identity and the authors assemble these changes into modular networks that represent "deterministic regulatory network" controlling neurogenesis.

1. Understanding the mechanisms of conversion of pluripotent cells to neurons by two transcription factors is of high importance. However, it is not clear whether the authors provide any new insights into the process. The networks presented in figures are heavily filtered for genes that were previously implicated in neuronal differentiation, thus eliminating potential new discoveries.

We appreciate the Reviewer #2's acknowledgment of the importance of the rapid differentiation process and of improving our understanding of molecular mechanisms underlying. We had attempted to highlight the novelty in our analysis in the previous submission, but in retrospect the presentation of our methods and results needed much improvement, as there were many misunderstanding among the reviewer comments. We hope that our revisions in our work will adequately clarify the work and demonstrate the major impact that not only the differentiation protocol will have, but also that the network analysis indeed is novel since no previous studies have implicated the combination of pathways used in this unique cell line, which allows the cells to rapidly and directly differentiate into a homogeneous population of neurons, while traversing a progenitor state. Furthermore, we have added a number of additional experiments and analyses to strengthen our claims.

Just to immediately clarify a couple misunderstandings, we note that we indeed provide important new insights into the process of stem cell derived neuronal differentiation. First, the protocol is novel, and no other studies have achieved neurons of this homogeneity within 4 days in a single medium with only one short activation of two transcription factors. Thus, with the novelty of the iNGN cell lines, nobody has been able study which combination of regulatory pathways could contribute to this rapid and homogeneous differentiation. Thus, while we agree that using the IPA package provided individual interactions that had been seen before in other cell lines/organisms, the combination of these interactions into a connected network of transcription factors is completely novel. Furthermore, we have now added additional siRNA experiments to demonstrate the phenotypic effects of perturbing the putative network as predicted by our IPA analysis. In addition, we have significantly updated the text and methods to highlight the novelty of these factors.

Second, we acknowledge that the methods for the IPA analysis were previously inadequate, and it made it sound like the network analysis was done by choosing previously known neuronal genes. Unfortunately, this was not the case. The goal of the analysis was to identify the transcription factors that showed the strongest signature of activation or repression that were also connected to the Neurogenins. Thus, we did not heavily filter for neuronal genes; in our analysis, we removed regulators that were not expressed, or whose expression changes conflicted with their activation signatures. This helped us narrow down and elucidate several transcription factors relevant to this rapid and homogeneous differentiation. It just so happens that some (but not all) of the factors have been implicated in neurogenesis. However it is exciting and important to note that no other studies to date have shown this particular combination of factors. And more importantly, no study has shown the concerted use of these factors to obtain a highly homogenous population of neurons so robustly. We have included now, with the much improved text, experimental data in which we perturbed several regulators in our network and demonstrate that these perturbations significantly affect the cell morphology and heterogeneity of the resulting neurons, and also that downstream neuronal gene expression is affected.

We also apologize that some parts of the previous manuscript may have raised different expectations and gave the impression of a biased analysis. In the revision process, we worked on the clarity of our data presentation and updated the text, figures, figure legends and methods accordingly.

Importantly, the authors do not include information about Ngn2 targets (by performing ChIP-seq) and the networks derived from time-course analysis provide little information about hierarchy of individual regulators. Overall it is impossible to determine which regulators, pathways and interactions are important and which are secondary or even irrelevant. I would therefore recommend eliminating the filters described in Methods section and reanalyze temporal expression changes in an unbiased way. Furthermore, it would be extremely helpful to compare the process of neural differentiation driven by Neurogenins with neuronal differentiation driven by extrinsic factors (e.g. Chambers et al, 2009, or retinoic acid treatment).

We definitely agree that obtaining a comprehensive view of the DNA binding profile of the Neurogenins will be interesting. However, these experiments are non-trivial, and the reasoning for obtaining it would beg for us to obtain additional ChIP-seq experiments for all of the downstream regulators, too. Since this knowledge would provide a much deeper insight in the regulatory pathways in neuron differentiation, we have provided the cells to the ENCODE consortium, and they are beginning to generate these data for our cell line. In the future, there will be a sophisticated data set on ChIP-seq data covering hundreds of transcription factors for the iNGN cells.

However for the time being, the cell line itself is of considerable interest to the field, and should be published as soon as possible, as it is already being adopted for study by several other groups. Our work as it stands now provides a valuable protocol, and the analysis provides a series of regulators that can be perturbed to get different neuronal phenotypes. We eagerly anticipate future analyses that will build upon our work to provide a more comprehensive view of all regulatory interactions and downstream neuronal target genes, but it is likely the experiments and analyses will take a few more years and a lot of money to complete.

Again, the aims of the analysis of the regulatory network was to identify important transcription factors in the differentiation process, and the filters were standard and allowed us to narrow the search to those that were most consistent with the predicted activation/repression. Our approach during the network analysis using IPA should not have biased our analysis towards factors known to be involved in neurogenesis. We have now clarified this in several locations in the manuscript, and greatly improved the methods in the revised manuscript. For example, we introduce our results as following: "Thus, we analyzed the time-course of mRNA expression data in the context of known transcription factor interactions in Ingenuity's IPA database (See Materials and methods). To identify potential regulators, a standard and non-neuronal biased enrichment test (Kramer et al, 2014) was conducted to identify transcription factors that had an overrepresentation of differentially expressed targets, and had their targets changing expression in the direction consistent with the activation and repression activities of the transcription factors of interest (Supplementary Table 5). We focused here on a network of transcription factors that met these criteria and that were also connected the Neurogenins through direct and indirect gene regulatory interactions that had been validated in other cell types and/or organisms, as catalogued in the IPA database".

We also agree that a comparison with chemical induction protocols would gain deeper insights in understanding neuronal differentiation from stem cells. This could be useful to improve the current protocols. But the proposed method by Chambers *et al.* 2009 does not really shed light on the cellular state of induced neurons (morphology, cell types of induced neurons, formation of neuronal rosettes, >10 days incubation times, presumably mixed populations, lack of functional tests, ...). Chambers *et al.* also published in 2012 an outstanding protocol to differentiate human iPS cells to nociceptors with a 75% differentiation rate in 15 days. They also identified both Neurogenin-1 and Neurogenin-2 to be activated in the process as well as other transcription factors that we also found upregulated in iNGN cells suggesting that there are likely similar regulatory pathways during neuronal differentiation of these cell types. While the aforementioned studies were valuable work, the many differences with our approach such as different starting iPS lines, timelines (four days versus 15 days) or culturing conditions (feeder free versus cells grown feeder cells, different media, etc), and the approaches used to

characterize the cells (RNA-Seq versus gene arrays) would not allow a fair and meaningful comparison. We agree that a systematic comparison between the two protocols would be invaluable if all experimental parameters are kept comparable and we consider this as a follow up story.

2. The authors picked two genes that were previously shown to play important role in neuronal differentiation - REST and NEUROD1. They performed overexpression and knockdown studies and concluded that while these manipulations change phenotypes of resulting neurons they do not interrupt neuronal differentiation program. It is not clear whether this conclusion extends to all neuronal differentiation programs or just to the one driven by Neurogenin expression. Control experiments where genetic manipulations are performed in cells neuralized by SMAD inhibition or retinoic acid treatment should be included.

Reviewer #2 raises a good point that it would be interesting to know if the conclusions stemming from our perturbation studies are unique to the iNGN or more broadly to other differentiation systems. I think it's valuable to note here that the regulatory program we present here is probably unique to the iNGN protocol, since no other differentiation protocol achieves the high homogeneity at this speed with any additional experimental interventions. Thus, we anticipate that portions of this program are unique to our protocol. However, we agree with Reviewer #2 that comparisons to other protocols would be quite valuable. Unfortunately, as mentioned above, for technical reasons very relevant to the paper, unbiased comparisons are infeasible. That is because the protocols suggested are inefficient to compare with the highly efficient protocol presented here. It would probably take an additional couple years just to figure out how characterize the other protocols to the same depth we are doing with the iNGN cells, and then how to compare them in an unbiased manner. Thus, such a comparison would be a valuable follow-up study.

We took the suggestion, however, for more thorough experimental assessment of our cell lines and in the revised manuscript, we targeted NEUROD1, NEUROD2, NEUROD1+ NEUROD2, NEUROD1+PAX6, POU3F2 and ZEB1 by siRNAs. The knockdown of these regulators was successful as measured by qPCR (Supplementary Figure 11) and resulted in the expected reduction in target expression levels (Figure 7G). Consistently with the previous knockdown experiments, all these manipulations also altered the morphology of the resulting neurons. These data first validates the proposed gene regulatory network and second it shows that manipulations to key members can alter the final neuronal cell types.

3. I assume that the authors performed expression analysis in cells overexpressing REST and in NEUROD1 knockdown cells. This seems to be the basis for figure 7G, but proper description of the experiments and more detailed analysis is missing.

This needs to be rectified, otherwise sentences such as: "19% of the neuronal genes were under REST or NEUROD1 regulation. In our gene regulatory network, only one gene, SLIT2, cannot be activated by other regulatory proteins following the removal of NEUROD1", do not make much sense and sound like arbitrary cherry-picking.

From this comment, it was clear that the description and data presentation in the previous submission was not optimal. Indeed, we hadn't expression profiled the REST/NEUROD1 perturbed cell lines. In addition to working to substantially improve the clarity of our work, we have also now done qPCR to characterize these genes in the knockdown cells (including additional siRNA knockdowns of other pro-neuronal factors, as shown in Figure 7 and Supplementary Figures 10 and 11). The target gene selection is based on known targets of these factors taken from the IPA database, and efforts were made to select representative genes that were known to be regulated by as few regulators in our network as possible. For the sh-NEUROD1 knockdown experiments we changed the text to: "In our gene regulatory network analysis, only one gene, SLIT2, seemed to be under unique NEUROD1 control whereas other regulatory factors can compensate for all other NEUROD1-controlled genes following its suppression (Supplementary Figure 10). Indeed, SLIT2 expression levels were significantly reduced on day 4 as compared with a control shRNA (Figure 7C) whereas the lack of NEUROD1 resulted in non-significant expression level changes of NEUROD2 and SOX2 (Supplementary Figure 10)". We have moved the REST overexpression data to Supplementary Figure 10.

4. What does it mean that the resulting cells are similar to prenatal cerebellar cortex AND postnatal mediodorsal nucleus of thalamus? Does it imply that the cells are confused and their identity cannot be matched to any known type of nerve cell?

At first blush, it may sound odd that the resulting cells are most similar to cells found in two different regions in the brain. However, this isn't a major concern for a couple reasons. Since we are looking at correlations here, it isn't problematic to have our neurons similar to neurons in two different regions in the brain. Furthermore, it isn't uncommon for neuron precursors to migrate as the brain develops. Most importantly however, we note that the BrainSpan data covers entire brain areas that include different populations of neurons, and so there is noise coming from other cell types and so the spatial resolution is a suggestion at this point. The BrainSpan data lack single cell resolution and therefore our analysis can only narrow down the likelihood that iNGN cells or neurons with similar transcriptomic profiles exist in the human brain. Nevertheless, the BrainSpan dataset is currently the most adequate resource of transcriptomic data of human neurons in the brain. The reviewer's comment highlighted a need for improved analysis and text, and so we updated the BrainSpan analysis (Figure 3B and C). We also show all brain areas in the revised manuscript since we previously only highlighted the ones with the highest correlations.

5. Overall the manuscript reads like presentation of data without much insightful interpretation and the overall message is lost or difficult to follow.

We apologize for the misunderstandings and the presentation style in our previous manuscript. The revised manuscript is modified and the text, figures, figure legends and methods were updated. We worked on the clarity of our results and interpretations, and better highlight the several novel impacts and discoveries in this work, as follow:

- 1. Neuronal differentiation from stem cells is extremely efficient by transcription factor induction without any additional bioactive factors in the culturing media (Figure 1, Supplementary Figures 1 and 2)**
- 2. Neuronal maturation and electrical activity need additional extrinsic factors even if the synaptic machinery is expressed within 4 days in stem cell media (Figure 1 and Supplementary Figure 2)**
- 3. The iNGN cells differentiate rapidly and continuously via unstable progenitor states and NOT directly as frequently proposed (Figure 3 and Supplementary Figure 4)**
- 4. The BrainSpan analysis assigned iNGN cells higher correlations to prenatal human tissues whereas the spatial mapping was not as clear for one particular brain region. The likelihood that iNGN cells resemble cortical neurons is very low (Figure 3)**
- 5. MicroRNA levels also dynamically change from stem cell profiles (miR-302 cluster) towards neuronal ones (miR-124, -96 and 9). Our analysis suggests that microRNA regulation takes place mostly downstream of the neuronal differentiation initiation phase (Figures 4 and 6, Supplementary Figures 5, 6, 8 and 9)**
- 6. We identified a network of transcription factors that contribute to the differentiation, and validated several central factors regulating neuronal genes by individual and combinatorial perturbations (Figure 5 and Supplementary Figure 7)**
- 7. We present the coding and non-coding transcriptomic blueprint of a neuronal differentiation program from human induced stem cells. Our data primes further targeted manipulations of iNGN cells and/or the usage of different transcription factors cells to increase the variety of human neurons. Notably, we found most transcription factors that are currently used also activated by the neurogenins explaining why most protocols result likely in similar neuronal cell types, i.e. excitatory neurons (Figure 2)**

We believe that molecular systems analyses of the genetic programs during stem cell-derived neuronal development are the key to increase the variety of generated neurons. Herewith, we attempt to provide the molecular insights for one efficient differentiation protocol that will be of great interest to the

interdisciplinary community of neuronal differentiation. We hope our revised manuscript convinced Reviewer #2 of its relevance and novelty.

Reviewer #3:

Summary:

This manuscript describes a novel differentiation protocol that yields pure but immature neuronal populations after 4 days of induction (over-expression of two neurogenins), and more mature neurons after 14 days of induction. This seems to be a significant improvement over existing protocols, in terms of speed and purity. The latter half of the manuscript describes RNA-seq profiling of the cultured cells at various stages of the differentiation process. The transcriptomic data are used to deduce the most similar *in vivo* brain regions and developmental stages, and to infer the regulatory network that lies at the heart of the differentiation process.

General remarks:

One general point of frustration is that bioinformatic methods are poorly explained. In most cases, the main text and figure captions present only a very high-level description of the methodology. As a result, it is hard to judge the validity of the conclusions. Even the Materials and Methods section is vaguely worded, and decoding it requires specialized knowledge plus a bit of guesswork. For example, readers who are not familiar with Ingenuity Pathway Analysis (IPA) software would have no idea what to make of the network-related sections of the manuscript. Other bioinformatic portions of the manuscript are similarly unclear.

We apologize for the vague wording and inadequate descriptions in the text, figure captions and methods. Thus, we have carefully (and comprehensively) updated the text, figures, figure captions and the methods to increase the clarity of our manuscript.

In general, the bioinformatic methods are ad hoc, and the conclusions do not seem robust (see below). Since validation experiments are not presented, this is a major concern.

As we have revised our text, we noted several places where unfortunately the bioinformatic methods seemed ad hoc based on inadequate description. However, most of the bioinformatics approaches are based on standard and published protocols. We have carefully gone through and cited them as needed. Basic statistical analyses are more carefully described. In addition, we validated several of the transcription factors that were predicted to be important in regulating the iNGN differentiation. This was done with siRNA perturbations and measured expression levels of downstream regulated genes.

Major points:

The analysis of iNGN similarity to human brain transcriptomes has issues. Since BrainSpan RNAseq data are derived from heterogeneous tissues, they are not ideal for inferring the neuronal subtype of cells cultured in vitro. The authors assume that the Pearson correlation coefficients (PCCs) shown in Figure 3 are indicative of neuronal subtype, but in reality they are probably more indicative of the relative abundance of neurons/progenitors in any particular brain region.

The reviewer raises an important point here. The heterogeneity of the BrainSpan Data makes it difficult to make definitive calls on exactly the identity of our iNGN cells. However, what is clear from our analysis is that the day 4 cells show a significantly higher correlation with brain tissues than day 0 cells are. Thus, to clarify this point we have updated the text, and modified the presentation of the BrainSpan analysis. We have clarified the BrainSpan limitations in the text.

The reviewer also raises a valid concern that high correlation with distinct brain regions might be the result of higher neuron populations in those brain regions. To test this idea, we took GFAP expression levels (a proxy for glial cell populations) and looked to see if they were anti-correlated with the Z-scores for each brain region at each time point. If brain region association was the result of higher neuron populations, then one would expect that regions with a higher Z-score would have lower glial cell count (or lower GFAP expression in our analysis). However, only two time points showed anti-correlation, but the p-values were weak ($p = 0.45$ and $p = 0.2$) and insignificant after correcting for multiple hypotheses. This suggests that higher Z-scores in our comparison to the various brain regions analysis are not the result of the brain regions having higher neuron populations. We have mentioned this control in manuscript now and describe the analysis in the Supplementary text.

Another problem with this section is that Figure 3A lumps together all of the samples derived from a single developmental stage. It's not obvious how this is meaningful. At best, day 4 iNGN cells would be similar to neurons/progenitors in one specific brain region (cerebellar cortex, for example), or one subset of brain regions. This similarity could be obscured when one averages over all brain regions.

While the reviewer has a point here, it is not necessarily relevant to the interpretation of the previous Figure 3A in our paper. That is because we make no claims about similarity to specific regions from that analysis. The goal of Figure 3A in the previous submission was to just compare day 0 and day 4 cells with the brain tissues at different time points. From this analysis we only have two claims from the analysis: First, day 4 cells correlate with human brain tissue better than day 0 cells suggesting that the iNGN cells become more similar to real brain tissue after differentiation. Second, the correlation is higher in prenatal brain tissue. Through our analysis we had taken care to ensure that outliers did not skew our results, and we feel that the results are robust. That is because most brain tissues are roughly equally

represented at each time point (with a few exceptions when a brain tissue was not evaluated at a given time point), thus, a single tissue cannot skew the results too much. To demonstrate this, we have plotted the correlation between Day 4 iNGN cells and all brain regions.

Given the number of different brain regions (more than two dozen), it is difficult to make sense of the plot with all regions. Thus, in the text we show the standard deviation at each time point, and since most time points demonstrate a small span for most time points, this suggests that it is unlikely that a subset of brain regions is skewing our results in regard to 3B in the revised submission.

Perhaps for the reasons listed above, the results in Figure 3A look a bit surprising. Day 4 iNGNs show greatest similarity to the earliest analyzed developmental stage (8 pcw), so the obvious conclusion would be that 8 pcw brain (or some earlier time point that was not included in the analysis) is the closest *in vivo* match. But then we see that day 0 iNGN cells, which are pluripotent, also show high similarity to 8 pcw brain (PCC=0.73). This makes no sense, because there are obviously no pluripotent cells in the brain.

It is still a question how potent neural stem cells are but these cells exist in human brain development and also in the adult human brain (see for example Sanai *et al.* Nature 2004). The iPS cell line we used to generate iNGN cells was derived from fibroblasts, which are ectodermal. Hence, even in the pluripotent state, it's likely that they share some transcriptomic similarity with neuronal cells that are also ectodermal. However, we would suspect that day 0 iNGN cells would be more similar to younger, less differentiated and

specialized neuronal tissue as we see in Figure 3B. We agree with the reviewer that day 4 iNGN cells have highest correlation with neurons in the earliest sampled tissues.

Figure 3B again uses an odd metric for matching iNGN cells to brain samples, and as in Figure 3A, no explanation is provided for why this metric would be a natural choice. In 3B, the PCC of a brain region is compared to the PCCs of other brain regions from the same developmental stage (z-score). It would seem more natural to use the PCC itself, rather than the z-score of the PCC. By the z-score metric, day 4 iNGN cells look like cerebellar cortex prenatally and mediodorsal thalamus postnatally, which is a peculiar switch.

We agree that the rationale for this metric was not made clear. Basically, we sought to know which brain regions were most similar to our Day 4 cells. For this one would want to compare each region to each other, while controlling for the steady decrease in correlation witnessed over time. While we show the temporal profile of similarity in Figure 3A, we wanted to control for the temporal factors and show the spatial correlations in Figure 3B (in the previous submission, now replaced with a different analysis and plot in Figure 3C in this submission), which is why we show Z-scores instead of PCC (showing the PCC again was redundant in our view). We feel that the Z-scores are a far better measure since if a brain region continually shows higher correlation than others, it provides greater control against a single outlier, focuses analysis on brain regions as it controls against general transcriptomic changes that occur over time, and strengthens the support of a specific brain region being more similar. We have clarified this reasoning now in the methods section.

We agree with the reviewer that the analysis requires improved presentation, and so we plotted it as a heatmap to allow all brain regions to be seen. We also looked for a correlation of glial cells with the Z-scores to test if it merely is a correlation based on the neuronal/glial ratios in the tissues, which, as stated above did not show an increased glial presence in tissues with lower Z-scores and *vice versa*.

It is claimed that this is plausible because neurons migrate during brain development. However, this explanation can be used to justify more or less any result. Do cerebellar cortical neurons migrate to the MD nucleus of the thalamus? The other justification is also unconvincing: that the cerebellum and thalamus are functionally connected. The MD nucleus is actually connected to a very large number of brain regions - the cerebellar cortex is hardly unique in this regard.

We agree that without tracing the migration of neurons, the rationale we presented is weak. Thus, we deleted this sentence from the manuscript. We also updated the spatial mapping analysis. In the updated analysis, our results

remained consistent, and one thing that is clear is that the iNGN cells have very low correlation with cortical samples (Figure 3C).

I strongly suspect that other ways of analyzing the BrainSpan data would yield different results. A more natural approach would be to combine Figures 3A and B into a single figure: a heat map of PCCs that has brain regions along one axis and time points along the other. Bi-clustering on such a heat map should reveal brain affinities in a more natural way. Of course, there is still the problem of tissue heterogeneity. Perhaps this could be partially mitigated by replacing expression values with fold-changes (or the log of fold-change, to prevent a small number of highly-expressed genes from dominating the PCC). Each tissue could be represented by its fold change over the median of all tissues, and iNGN cells could perhaps be represented by the fold change from day 0 to day 4. Also, it would be good to see some positive and negative controls, to increase confidence in the ad hoc methodology used in the BrainSpan analysis.

The clustering idea is an interesting suggestion. The challenge with combining the two previous analyses is that it does not allow one to control for the global changes that occur in the brain over time. Thus it helps to decompose the analysis into the temporal and spatial features, thus making it easier to control for biases from the differing analyses. However, we appreciate the suggestion and have clustered the Z-scores for all brain tissues, and presented it as Figure 3C. We note that if we cluster using PCCs instead, we get qualitatively the same results. As you can see below, the PCCs were clustered (top) and compared to the Z-scores (bottom). There are some minor differences in the dendrograms, but one optimal leaf ordering is applied, the ordering of the leaves are similar.

The section in the main text entitled "A modular deterministic gene regulatory network drives the rapid neurogenesis" is so terse that it is unintelligible. The corresponding portions of the Materials and Methods are also elliptical and lacking in detail. As far as I can tell, the gene regulatory network was inferred from RNA-seq data by starting with the output of a software program (IPA) and then applying a series of ad hoc filters. Subsequently, the network was "manually curated," but there are no details on what this manual curation step involved. Network inference is a notoriously difficult problem. I would like to see more details and more justification in the Materials and Methods (including key aspects of the IPA algorithm and database) before placing much faith in the network models shown in Figure 5.

Indeed, the details in our transcription factor analysis were vague. While a systematic approach was taken we hadn't clearly explained it. We completely revamped the text in that section and the methods section. Furthermore, after we had previously submitted our work, the details of the IPA algorithms were published (see Kramer et al, 2014). We have now cited this manuscript, thus providing further details on the algorithms in their software. Also, in regard to the "manual curation", we note that this entailed just categorizing regulators in our network and grouping them as seen in Supplementary Figure 7. In a few cases links were added based on literature reports that were not included in the IPA database. These additional links are cited in the supplementary text where we describe the interactions in the network, which we discuss. The details of how the transcription factor analysis has been substantially revised and clarified in the manuscript now.

At this time, we also want to emphasize that this analysis was used to identify a series of connected transcription factors that contribute to the differentiation process. Through our analysis we did just that. We have now gone on to validate the contribution of several central transcription factors and demonstrated that their perturbation influences the homogeneity, morphology, and expression of neuronal genes regulated by these factors. Thus, with the extensive rewrite of our results and methods sections, we hope additional misunderstandings are mitigated, and that it is clearer how we identified key regulators that contribute to this rapid robust differentiation.

More generally, it's important to emphasize that the networks in Figure 5 and Figure 6 are just models. They may indeed be plausible and partially consistent with the literature (the IPA database is partially literature-derived). However, they have not been validated in iNGN cells. The portion of the manuscript entitled "The loss of pluripotency" needs to make this point clear. The same goes for subsequent portions of the manuscript. As written, they sound like a list of definitive conclusions, but they are actually only a list of bioinformatic hypotheses with moderate or marginally significant p-values and no validation. Moreover, all such

network models are incomplete - they have many missing nodes and missing edges, and one should exercise caution in drawing inferences from them.

We fully agree with this concern, and have made it clear where the transcription factors we identified and their organization is putative at this stage. We have completely changed this portion of the manuscript to address such issues. We have also emphasized that this analysis was done to identify many contributing regulators, not necessarily all, since it is true that the IPA database would be missing nodes and links. We note also that much progress will be made in the next couple years in completing these networks since the cell lines are being deeply characterized by the ENCODE project and will be subjected to many CHIP-seq experiments. Furthermore, more than 10 other groups are already using these cell lines, so we expect to gain much additional knowledge in the near future. Lastly, we also note that we have done additional validation experiments. Thus, we have updated, edited and changed the network part and included siRNA perturbations against several regulators, and through this we hope we have increased the confidence that we have identified several core regulatory factors that help induce the rapid neurogenesis upon Neurogenin activation.

One specific concern in the network analysis is that SOX2 is held up as a key player in the neuronal differentiation cascade, even though this TF did not score well in IPA analysis. If the authors adhered strictly to their numerical cutoffs, would it even be part of the network?

We acknowledge the reviewer's concern. At first glance it appeared that SOX2 did not score well in IPA. SOX2 contributes its pluripotent role also in a complex with NANOG and OCT4. When the targets of the complex are considered, SOX2 scores well. Unfortunately the complex scores had been erroneously omitted in the generation of the supplementary table, but have now been included. We have further verified the accuracy of remaining scores reported in the supplement. Thus, we in the network analysis, it should now be clear that we adhered strictly to the specified cutoffs.

Minor points:

On line 201, it is stated that BrainSpan data cover the "full course of human brain development." While this is technically true, it gives the wrong impression because the authors did not analyze the earliest BrainSpan samples. They actually started their analysis at 8 pcw, which is quite late in brain development (embryonic day 15.5 in mouse). It would have been better to include earlier developmental stages in the analysis.

We agree that it would be ideal to compare our data to expression in earlier brain tissue. Unfortunately, the 8 week data were the earliest that could be downloaded from the BrainSpan atlas website.

The regulatory network is repeatedly described as "deterministic," but it's not clear what this word means in this context, or what the evidence is. What would constitute a non-deterministic network?

A "deterministic" network would be one that once induced, is committed to one specific fate. In our study, we found that the cells, once induced all were driven to a homogeneous bipolar neuron state. However to improve clarity, we have removed this terminology in our revision.

Line 265: which enrichment test is referred to here?

It's the algorithm used in IPA (Fisher's exact test, with the comparison of differential expression direction and known regulatory functions of the transcription factors). We have now cited the paper describing the algorithm at that point in the paper. For example, at this location we updated the entire paragraph in the revised version to "Thus, we analyzed the time-course of mRNA expression data in the context of known transcription factor interactions in Ingenuity's IPA database (See Materials and methods). To identify potential regulators, a standard and non-neuronal biased enrichment test (Kramer et al, 2014) was conducted to identify transcription factors that had an overrepresentation of differentially expressed targets, and had their targets changing expression in the direction consistent with the activation and repression activities of the transcription factors of interest (Supplementary Table 5)".

Line 696: what is an "enriched regulon?" Which list was filtered?

An "enriched regulon" is a regulon that is overrepresented in the differential expression (i.e., genes regulated by a specific regulator tend to be differentially expressed more often than expected by chance. We have reworded this to improve clarity.

Line 698: what is the rationale behind these keywords?

These capture the transcription factors involved in differentiation, and helped us to remove the chemicals and drug regulatory interactions in IPA. We have now clarified this in the methods.

Line 700: was the enrichment p-value corrected for multiple testing?

We have corrected this using the Benjamini FDR of 0.05, and clarified this in the text.

Do the red and blue colors in Figure 5 represent neuronal and pluripotency factors?

We have now clarified the meaning of the colors. Basically, the colors correspond to the activation Z-score (activation = red, repression = blue).

Lines 707-710: the logic of the global regulator steps is unclear. How are global regulators defined? What is the rationale behind these processing steps?

We have rewritten much of this portion to clarify the steps, and clarified that the aim of this was to identify a core set of regulators that specifically aid in the differentiation process. We admit that the removal of global regulators might miss some important factors, but we felt that their removal would help us narrow in on the factors that are more specific to our differentiation. Regulators that are known to regulate a large number of other genes, while possibly contributing, were not considered since they act less specifically under many different conditions, target many genes in diverse processes and thus did not meet the criteria that we were aiming for (i.e., regulators that were more specifically needed for the phenotype we see). However, we still report these global regulators in supplementary table 5.

Supplementary Tables 2 and 5 are formatted in a manner that is difficult to read - rows are split up over multiple pages.

We apologize for this problem. It is a common issue when submitting xlsx files to MSB, but we will assure that the xlsx sheet is available online upon publication.

Line 687: what is the meaning of "predicted data?" Are the predictions credible?

IPA has two classes of interaction data: 1) data that are experimentally derived, and 2) interactions that are predicted (mostly predicted miRNA interactions). While setting up the analysis you can select which data classes to include. We used both. However, the downstream miRNA analyses were based on validated interactions from miRTarBase.

Line 688: Table S3 seems to be about miRNAs, not upstream transcription factors.

Thank you for catching that. That was a typo from a previous submission of this study and has now been corrected.

Lines 728-730: I'm not sure the hypergeometric test is valid here, because the BrainSpan RNAseq data show intertemporal correlations.

In our revisions, this analysis was removed and so the test was not needed anymore.

Thank you again for submitting your work to Molecular Systems Biology. We sent your manuscript to the previous reviewer #2 (current reviewer #3) and to two new reviewers (#1 and #2), who were asked to evaluate the study afresh. We have now heard back from these three referees who, as you will see from their comments below, think that the presented protocol and accompanying datasets are potentially interesting. They raise, however, substantial concerns on your work, which should be convincingly addressed in a revision of the manuscript.

The two major points that need to be convincingly addressed experimentally are the following:

- As reviewers #1 and #3 point out, the identity of the differentiated neurons is not analyzed in detail and it remains unclear whether a homogeneous population or a mixture of different neuronal subtypes is obtained. As such, single cell analyses of further neuronal markers, also including VGLUT1 and ChAT, should be performed. In our view, this is an important point both from a biological and applied point of view. Even though these experiments would not fully reveal the nature of the differentiated neurons and whether they correspond to *in vivo* subtypes, they would go a long way towards addressing the immediate issue of the homogeneity of the resulting neuronal population.
- Reviewer #2 refers to the need to demonstrate the generality of the protocol using further cell lines.

On a more editorial level, we would like to ask you to include the links and accession numbers to all datasets (i.e. RNA-Seq) and to provide the related information in the "Data Availability" section of your manuscript. For more information you can refer you our journal policies on data deposition (<http://msb.embopress.org/authorguide#a3.5>).

If you feel you can satisfactorily deal with these points and those listed by the referees, you may wish to submit a revised version of your manuscript. Please attach a covering letter giving details of the way in which you have handled each of the points raised by the referees. A revised manuscript will be once again subject to review and you probably understand that we can give you no guarantee at this stage that the eventual outcome will be favorable.

Reviewer #1:

This manuscript by Busskamp et al describes the analysis of gene expression changes during "reprogramming" of human induced pluripotent stem cells into neurons following the transient overexpression of Neurogenin transcription factors. The key finding of the manuscript is that Neurogenin-mediated induction of neuronal fate among induced pluripotent stem cells rapidly and efficiently induces neuronal fate and allows high throughput analyses to characterize the trajectories of gene expression changes following Neurogenin expression. Similar approaches have been used to generate neurons from pluripotent cells by other groups (Farah et al., 2000 Development, Reyes et al., 2008 J Neuroscience) and therefore the experimental approach does not constitute a major advance in the field. However, the high throughput analysis presented in the manuscript is of very high quality and will no doubt be a very useful reference for reprogramming protocols. The roles of Neurogenins in neuronal differentiation is very well established, but the precise molecular cascades have not been extensively investigated.

My main concern is that the identity of the neurons generated using this protocol has not been discussed in very much depth. It would be important to analyze neuronal differentiation in longer term, with respect to their possible cortical layer identity and the ability to integrate into neuronal circuits *in vivo*, not just *in vitro*.

In addition to describing the longer term development of neurons derived using this protocol, I would strongly encourage the authors to examine the dynamics of differentiation in more detail, possibly by turning off the expression of genes normally expected to mark neural stem cells *in vivo*,

such as Pax6. My concern is that the rapid induced differentiation may lead to the development of non-physiologically relevant cells, which may still correlate well at the genome-wide scale with bulk tissue profiling data, such as Brainspan, but at the single cell level may not actually correspond to any *in vivo* cell type.

Another important analysis I would highly recommend is to compare the iNGN cells gene expression profiles with standard iPSc differentiation protocols. Such data have been recently published by other labs and should be accessible (van de Leemput 2014 Neuron, and Stein et al. 2014 Neuron).

I have a minor concern regarding the analysis of microRNA expression profiles and the elucidation of the possible miRNA:mRNA interactome, specifically the conclusion that because relatively few miRNA target levels decline during differentiation, mature miRNAs may play a relatively minor role during differentiation. Although it has been proposed by several groups that changes in mRNA abundance follow microRNA interaction with target 3'UTR, recent studies have suggested that the primary mode of microRNA action may be through translational repression (see for example Meijer et al 2013 Science), which may not affect target mRNA levels. This possibility should be better reflected in the analysis as well as in the discussion.

Reviewer #2:

This manuscript describes the development of a reprogramming strategy that results in rapid conversion of human iPSc cells to neurons. The authors analyze the conversion process using mRNA and miRNA analysis, followed by various bioinformatics approaches.

The authors use co-expression of two transcription factors, Ngn1 and Ngn2. Inducible over-expression of these factors results in rapid conversion of human iPSc-cells to neurons with bipolar morphology, even if the cells are kept in stem cell media. The resulting cells express several neuronal markers. If the cells are transferred to different culture conditions they are also able to fire action potentials. The authors then perform mRNA seq and miRNA-analysis at different stages of reprogramming. Using extensive bioinformatical approaches, the authors document that the reprogramming goes via a progenitor stage. The authors then knock-down some of the transcription factors potentially involved in the formation of neurons and find that this results in immature morphologies and altered gene expression.

This study is one of many recent papers that document efficient generation of neurons using transcription factor over-expression. As such, the protocol described in this study largely confirms or extend principles already clarified by existing studies. The novelty in this study rather lies in the ambitious bioinformatical analysis of the reprogramming process. Over all, I have a sense of enthusiasm about this study. But I also have some critiques that I think are important.

Major comments:

1. The authors use a bi-cistronic vector to overexpress Ngn1 and Ngn2. The reasons for choosing these two factors are not well described in the MS. Do these factors complement each other? Do you get the same effect with only Ngn1 or Ngn2?
2. The authors rightly cite Zhang et al., 2013 which use a similar approach to generate neurons. It would be good if the authors would clarify the differences between their approach and the one used by Zhang et al?
3. All data in the manuscript is based on a single iPSc-line. It is necessary to document that the rapid reprogramming can be achieved on other iPSc-lines as well as on human embryonic stem cells.
4. The inclusion of the miRNA data is interesting since it adds another level of complexity. The authors conclude that miRNAs are "modest shapers of neurogenesis rather than powerful inducers". This strong statement is based on bioinformatical analysis that suggests that targets of upregulated miRNAs are altered less than targets of transcription factors. However, for miR-124, which is the most highly expressed miRNA in their system, only one validated target is included in their analysis. Certainly, miR-124 has many more targets... Thus, in order to make such strong a conclusion

additional experiments are necessary. A good start would be perturbation and overexpression of miR-124. It would also be good if the text were more balanced.

5. A major weakness in this study is that almost all analysis is on mRNA and miRNA despite the fact that the study is centered on TFs, which are proteins. At least some of the key findings in the study should be backed up with quantitative protein data (e.g. Western Blot).

Minor comments:

1. The title is too broad and vague

2. The authors should discuss how the choice of reprogramming factors may influence the generation of different subtypes of neurons. E.g. Ngn1/Ngn2 vs. Mash1.

Overall, this study provide an ambitious effort to document the neuronal reprogramming process. It will be of interest to the field but the above critiques need to be addressed.

Reviewer #3:

The manuscript provides a systematic analysis of molecular mechanisms underlying conversion of a pluripotent cell to a nerve cell following inducible expression of NEUROG1 and NEUROG2 in human stem cells. While the identity and nature of resulting nerve cells remains ill-defined, possibly not reflecting any real cell in the nervous system, the derived networks might be useful to other researchers studying the processes of neuronal differentiation in more physiological contexts. The revised text now more clearly describes the methodology that was used to derive the genetic networks. The findings are consistent with current state of knowledge of the effects neurogenin on gene expression. The authors validated well characterized targets of neurogenins and demonstrate that in the inducible system these targets are required for proper execution of the neuronal differentiation. While the study does not reveal conceptually novel molecular mechanisms, its re-discovery of important players in neural differentiation indicates that the expression data sets accompanying the manuscript will be useful and hopefully generally applicable.

Minor comments:

1. The authors should not conclude based on global gene expression data that resulting nerve cells express dual neurotransmitter phenotype. Simpler interpretation is that the protocol results in a mixture of nerve cells some of which are cholinergic and other are glutamatergic. To resolve this discrepancy one would have to perform single cell analysis.

2. Similarly, it is an extension to conclude based on the presented data that the cells follow normal developmental trajectory including neural progenitor stage. While several progenitor genes are induced it is unlikely that cells pass through neural plate and early neuroepithelium developmental stages. Indeed expression of the earliest and most general neural progenitor marker Sox1 is not induced during the transition.

1st Revision - authors' response

14 October 2014

Summary of changes (Revision for manuscript MSB-14-5508)

Main Text

During the revision we worked on the clarity of our manuscript and added several changes. The major changes are highlighted below:

- We added three coauthors who contributed to the work during the revision, i.e. Jernej Murn, Shangzhong Li and Michael Stadler.
- End of previous introduction: We have eliminated our statement that miRNAs were "modest shapers of neurogenesis rather than powerful inducers" as suggested by Reviewer #2.
- Lines 143ff.: As suggested by Reviewer #2, we have updated our motivation why we decided to overexpress Neurog1 and Neurog2 in human iPS cells. In addition, we highlight differences to the

protocol that Zhang et al. 2013 published.

- Lines 165f.: We mention the Western Blot data in the main text that we show in Figure E1F as suggested by Reviewer #2.
 - Lines 217ff.: We clarified the single cell immunohistochemical analysis of iNGN cells indicating the presence of a homogeneous neuronal population (Reviewer #1 and #3).
 - Lines 398ff.: We added a paragraph about the miRNA manipulations by miR- 302/367 and miR-124 sponges.
 - Lines 471ff.: As suggested by Reviewer #1, we added the possibility of translational miRNA repression to the discussion and cited Meijer et al, 2013.
 - Lines 501ff.: As suggested by Reviewer #3, we highlighted that SOX1 was not highly upregulated and therefore the neuronal progenitor states likely do not resemble early neuroectoderm lineage.
- Figures and figure legends
- Figure 2F was separated into three panels of which Figure 2G comprises a new quantification of vGLUT1 and ChAT positive iNGN cells.
 - Figure E1F: We added new Western Blot data for Neurog1, Neurog2, MAP2, vGLUT1 and ACTB.
 - Figure E1G: We present new data demonstrating that the bicistronic Neurogenin construct also works in human ES cells (CHB-8) and another iPS cell line (PGP9).
 - Figure E1H: We show the quantification of MAP2-positive neurons derived from CHB-8 and PGP9 stem cells after Neurogenin induction.
 - Figure E2B: We added a quantification of the fraction of cells shown in Figure E2A that were immune-positive for MAP2, NeuN, GLUR2, PAX6, DCX, SOX2 and GAT3.
 - Figure E10: This figure comprises new data on miRNA expression perturbations by miRNA sponges as requested by Reviewer #2. Manipulations for miR-302/367 (A-C), miR-124 (D-F) and control (G-H) are shown. Panel I shows corresponding quantifications of non-bipolar cell morphology fractions.
 - Figure E11: This was previously Supplementary Figure 10.
 - Figure E12: This was previously Supplementary Figure 11.
 - Figure E13: Here we added new data of individually overexpressed

NEUROG1 and NEUROG2 PGP1 iPS cells to address Reviewer #2's question. In Panel A, we show transmission light and MAP2-immunostained neurons. Panel B shows a pie chart of differentially expressed genes compared to iNGN cells. The scatter plots in C and D show differentially expressed genes compared to iNGN cell samples as revealed by microarray analyses. All corresponding figure legends were updated accordingly.

Materials and Methods

We updated this section according to the new experiments we performed during the revision (like Western Blots, miRNA sponges, additional human stem cell lines, etc). Furthermore, we added a "Data availability" sections with GEO accession numbers for our RNA-Seq, microarray and n-counter data.

Reformatting of the Supplementary Information

We added a paragraph about the microarray analysis of differentially expressed genes of individual NEUROGs versus iNGN cells (Figure E13) including the expanded view Materials and methods. Expanded View Tables

We added one spreadsheet, Expanded View Table E9. Changes in GO term enrichment of differentially expressed genes between cells induced with NEUROG1 or NEUROG2 vs. the Neurg1+Neurog2 iNGN cells (shown in Figure E13) are shown.

Point-by-point response (next page)

Please find below point-by-point responses to the editorial and reviewer comments for manuscript MSB-14-5508. We have interspersed our responses within the referee reports, in blue font. We thank all reviewers for their insightful comments. We performed new experiments, added new figures and rewrote the manuscript to address these comments.

Editorial response:

Dear Prof. Church,

Thank you again for submitting your work to Molecular Systems Biology. We sent your manuscript to the previous reviewer #2 (current reviewer #3) and to two new reviewers (#1 and #2), who were asked to evaluate the study afresh. We have now heard back from these three referees who, as you will see from their comments below, think that the presented protocol and accompanying datasets are potentially interesting. They raise, however, substantial concerns on your work, which should be convincingly addressed in a revision of the manuscript.

The two major points that need to be convincingly addressed experimentally are the following:

- As reviewers #1 and #3 point out, the identity of the differentiated neurons is not analyzed in detail and it remains unclear whether a homogeneous population or a mixture of different neuronal subtypes is obtained. As such, single cell analyses of further neuronal markers, also including vGLUT1 and ChAT, should be performed. In our view, this is an important point both from a biological and applied point of view. Even though these experiments would not fully reveal the nature of the differentiated neurons and whether they correspond to in vivo subtypes, they would go a long way towards addressing the immediate issue of the homogeneity of the resulting neuronal population.

Indeed, this is a very important point that we had already addressed in the previous manuscript but was likely overseen (previous Figure 2F and Supplementary Figure 2) by showing immunostainings for vGLUT1 and CHAT as well as additional neuronal markers. We used the Bitplane Imaris software package to calculate a co-localization channel resulting in 96.2% co-localization of vGLUT1 and ChAT indicating the presence of a homogeneous neuronal cell type. However, we clarified this aspect even more in the text (lines 217ff.) and did additional single-cell quantification analyses for vGLUT1 and ChAT (shown in Figure 2G) as well as for the markers MAP2, NeuN, GLUR2, PAX6 and DCX (all close to 100% of the cells were positive for these markers, see Supplementary Figure 2B) whereas 0% of the cells were positive for the GABA transporter 3

(GAT3) indicating that iNGN cells were not GABAergic, which is consistent with our transcriptomic data. Notably, all these analyses were conducted on a single cell level. Altogether, our data demonstrates that iNGN cells resemble a pure (>90%) neuronal population.

- Reviewer #2 refers to the need to demonstrate the generality of the protocol using further cell lines.

Again, this is a very important aspect of our work. Therefore, we generated additional iPS cell (PGP9) and human ES cell (CHB-8) inducible Neurogenin lines. Upon activation, these stem cells also underwent rapid neuronal differentiation as shown before in the iNGN cell line (>90%). Hence, our protocol very likely generally works in human stem cell lines. For simplicity and free access to our protocol, we will make all lentiviral vectors accessible to the scientific community at Addgene.

On a more editorial level, we would like to ask you to include the links and accession numbers to all datasets (i.e. RNA-Seq) and to provide the related information in the "Data Availability" section of your manuscript. For more information you can refer you our journal policies on data deposition (<http://msb.embopress.org/authorguide#a3.5>).

Datasets have been deposited at the NCBI Gene Expression Omnibus, and can be accessed with the following accession numbers: GSE60548 (Illumina RNA-Seq), GSE62145 (nCounter miRNA), and GSE62146 (Agilent microarray). We also included the microarray data we show in the new Supplementary Figure 13 for showing differences when using individual Neurogenins as requested by Reviewer #2. All accession numbers are listed in the "Data availability" section of the revised manuscript.

If you feel you can satisfactorily deal with these points and those listed by the referees, you may wish to submit a revised version of your manuscript. Please attach a covering letter giving details of the way in which you have handled each of the points raised by the referees. A revised manuscript will be once again subject to review and you probably understand that we can give you no guarantee at this stage that the eventual outcome will be favorable.

Yours sincerely,
Maria Polychronidou, PhD
Editor Molecular Systems Biology

Reviewer #1:

This manuscript by Busskamp et al describes the analysis of gene expression changes during "reprogramming" of human induced pluripotent stem cells into neurons following the transient overexpression of Neurogenin transcription factors. The key finding of the manuscript is that Neurogenin-mediated induction of neuronal fate among induced pluripotent stem cells rapidly and efficiently induces neuronal fate and allows high throughput analyses to characterize the trajectories of gene expression changes following Neurogenin expression. Similar approaches have been used to generate neurons from pluripotent cells by other groups (Farah et al., 2000 Development, Reyes et al., 2008 J Neuroscience) and therefore the experimental approach does not constitute a major advance in the field.

Indeed individual Neurogenins have been overexpressed previously. However, note that these previous methods used more complicated culturing techniques, resulting in lower yields of neurons. Here we report a bicistronic Neurogenin-1 and -2 with defined commercial media in human iPS cells, which did not require supplementation with bioactive factors. This package as a whole led to the high homogeneity and speed we report here. Furthermore, such a simple approach hasn't been described before. For example, Farah *et al.* used embryonic mouse P19 cells, and Neurog1 overexpression resulted in 8% neuronal induction, and Neurog2 overexpression resulted in less than 1% neuronal induction. Reyes *et al.* combined Neurog1 overexpression with two neurotrophic factors to induce neurogenesis in mouse ES cells with up to 75% success rate within 5 days. The simplicity of our approach and much higher homogeneity are just two of the major impacts of our work and crucial for our systems biological approach. However, we acknowledge the other studies with citations because they have inspired our project. Furthermore, we have already shared our iNGN cell line with 10 international collaborating laboratories that do not have extensive stem cell culturing experience. The ease of iNGN cell culturing and differentiation enabled them for example to perform further epigenetic studies during differentiation as well as biophysical characterization of optogenetic tools in human neurons.

However, the high throughput analysis presented in the manuscript is of very high quality and will no doubt be a very useful reference for reprogramming protocols. The roles of Neurogenins in neuronal differentiation is very well established, but the precise molecular cascades have not been extensively investigated.

We would like to thank Reviewer #1 for acknowledging our analyses and work on iNGN cells, which we feel is of equal importance, since no previous study has been able to acquire such high resolution data over the course of differentiation given the low homogeneity of previous protocols.

My main concern is that the identity of the neurons generated using this protocol has not been discussed in very much depth. It would be important to analyze neuronal differentiation in longer term, with respect to their possible cortical layer identity and the ability to integrate into neuronal circuits *in vivo*, not just *in vitro*.

Our main focus aims to understand the early events in iNGN cell differentiation. We have taken our cells beyond day 4 and assessed the cell line to demonstrate neuronal functions using functional recordings shown in Figure 2 and Figure E2. It would indeed be fascinating to conduct additional studies that track the continued changes in the cells. Furthermore, after day 4, a diverse range of variations in the protocol could be tested, thus likely resulting in a wider range of neuronal phenotypes. While the classification of these would be a substantial undertaking, we look forward to seeing the results of such an important follow up study.

In the meantime, we note that by day 4 we see clear features of the neurons. Our BrainSpan analysis (Figure 3B and C) indicated that day 4 iNGN cells have higher homology with prenatal non-cortical brain samples. When more detailed reference data sets are made available (i.e., data in which purified neurons from each region are profiled to reduce the level of noise from glial cells), we would likely be able to more precisely decipher the iNGN cell identity. Meanwhile, we have bolstered the characterization of our neurons via immunohistochemistry. As aforementioned, we have already shared the iNGN cell line with several labs and some collaborators focus more on iNGN cells that have been cultured for more extended periods of time, as suggested by the reviewer. Transplantation experiments into mouse brains are also anticipated, since Neurogenin-2 work by Zhang *et al.* suggests that iNGN cells likely will functionally integrate into mouse brain circuits. Given the growing interest in our cell line, we are excited about the further characterization that is underway by several other labs.

In addition to describing the longer term development of neurons derived using this protocol, I would strongly encourage the authors to examine the dynamics of differentiation in more detail, possibly by turning off the expression of genes normally expected to mark neural stem cells *in vivo*, such as Pax6.

That indeed is a valuable experiment and we have used siRNAs against Pax6 to validate our transcription factor network analysis shown in Figures 7 and S11. The Pax6 knockdown did not significantly impact the rate of iNGN cell differentiation. However, it impacted the neuronal morphology, with a higher fraction of non-bipolar cells (Figure 7).

My concern is that the rapid induced differentiation may lead to the development of non-physiologically relevant cells, which may still correlate well at the genome-wide scale with bulk tissue profiling data, such as Brainspan, but at the single cell level may not actually correspond to any *in vivo* cell type.

It is absolutely possible that the rapid differentiation of iNGN cells could resemble a non-physiologically human neuron type. However, this really is a concern of any *in vitro* differentiation protocol. Unfortunately, to prove whether or not any differentiated neuron is physiologically relevant, one would need sophisticated single cell transcriptomic reference data of all human neurons. We look forward to the future development of technologies that could provide such data. However, as is standard practice in the neuronal differentiation field, we have characterized the expression of various neuronal markers and conducted electrophysiological assessment to demonstrate the neuronal properties of the iNGN cells. For example, we show that on the single cell level vGLUT1 and ChAT co-localize in iNGN cells. Together with additional immunomarkers shown in Supplemental Figure 2 and the morphological analyses we have high confidence that iNGN cells resemble a homogeneous population of neurons. We also clarified this aspect in the text.

Another important analysis I would highly recommend is to compare the iNGN cells gene expression profiles with standard iPSc differentiation protocols. Such data have been recently published by other labs and should be accessible (van de Leemput 2014 Neuron, and Stein et al. 2014 Neuron).

We were also interested in these papers, which both were published while we finalized our work for resubmission. For the sake of completeness, we cite these two extensive resource papers. It would be indeed interesting to analyze our data in the context of those published works. Stein *et al.* was carried out using microarrays, thus the technical differences between the platforms would be difficult to resolve without having proper controls to compare. However, van de Leemput *et al.* also employed RNA-Seq, and so we looked into the possibility of doing a comparison. We downloaded the RNA-Seq reads and used the same protocols we had employed for read alignment and transcript quantification as done on our own samples. Doing this allowed us to compare the data from our study with theirs. Unfortunately, we immediately noticed that the differences between the van de Leemput data and ours were too large to make a confident comparison. We saw a very weak signature of our day 0 data showing more similarity to their day 0 samples, and our day 4 samples showing more similarity to the transcriptomes of their differentiated cells. However, the trend was very

weak since the correlations between their transcript FPKM levels and ours were much lower than the correlations within our two experiments. That is, the RNA-Seq data from their stem cells was more similar to their differentiated cells than any of our datasets. Similarly, our day 0 cells were far more similar to our day 4 cells than any of theirs. Thus it was clear that differences in experimental setup and implementation of sequencing precluded unbiased comparison in the transcriptomes of the two studies.

While we are unsure what caused the large differences between our data and theirs, there are several experimental differences that could lead to differences we saw. In the differentiation itself there were differences of sampling frequencies, lengths of the time courses, and homogeneity of cells in the culture. On the RNA sequencing end, we also noticed differences in read length (which would influence alignment), sequencing run setup, read processing (e.g., trimming), and patterns in read quality. Lastly, we note that these are two different unrelated studies that were conducted in different labs and sequenced at different centers. Unfortunately there were no standard controls for normalizing the differences between the two experiments (i.e., same cell line and culture conditions). Since the differences between the datasets were much larger than and intraexperimental variation we were unable to make a meaningful comparison at a whole transcriptome level.

Despite these clear global differences in the transcriptome readouts, we noticed some specific differences that could point to potential biological differences in how our neurons differentiated in comparison to theirs, (although, given the global differences described above, care should be taken when interpreting the following similarities and differences). For example, in van de Leemput *et al.*, the chemical induction protocol does not induce NEUROG1 and -2 until around day 60, and even then the expression is quite low. We also see several other factors being induced later on in their profiles. This suggests that there are differences in the length of time needed to activate differentiation programs. However, for most of the transcription factors highlighted as important in our study, the overall patterns of expression are qualitatively similar (e.g., they increase or decrease similarly), except, for example, REST, which remains highly expressed in their cell line, while it is quickly down-regulated in our cell line. We emphasize here that while both their study and ours provide many insights into transcription of stem cell derived neurons, the experimental differences seem to mask the ability to quantitatively compare the data sets in a reliable fashion. It would be very exciting to see a follow up study in which both protocols were conducted and all non-biological variation is carefully controlled to see how slower chemical and more rapid transcription-factor mediated induction truly differ.

I have a minor concern regarding the analysis of microRNA expression profiles and the elucidation of the possible miRNA:mRNA interactome, specifically the conclusion that because relatively few miRNA target levels decline during differentiation, mature miRNAs may play a relatively minor role during differentiation. Although it has been proposed by several groups that changes in mRNA abundance follow microRNA interaction with target 3'UTR, recent studies have suggested that the primary mode of microRNA action may be through translational repression (see for example Meijer et al 2013 Science), which may not affect target mRNA levels. This possibility should be better reflected in the analysis as well as in the discussion.

We would like to thank Reviewer #1 for mentioning this aspect. We added this possibility to our discussion. Furthermore, we added additional data on miRNA knockdowns (miRNA sponges against the miR-302/367 cluster and miR-124), which did not result in significant changes of iNGN differentiation (Figure E10). Hence, our new data further support the idea that miRNAs indeed play minor roles during the first four days of iNGN differentiation.

Reviewer #2:

This manuscript describes the development of a reprogramming strategy that results in rapid conversion of human iPS cells to neurons. The authors analyze the conversion process using mRNA and miRNA analysis, followed by various bioinformatics approaches. The authors use co-expression of two transcription factors, Ngn1 and Ngn2. Inducible over-expression of these factors results in rapid conversion of human iPS-cells to neurons with bipolar morphology, even if the cells are kept in stem cell media. The resulting cells express several neuronal markers. If the cells are transferred to different culture conditions they are also able to fire action potentials. The authors then perform mRNA seq and miRNA-analysis at different stages of reprogramming. Using extensive bioinformatical approaches, the authors document that the reprogramming goes via a progenitor stage. The authors then knock-down some of the transcription factors potentially involved in the formation of neurons and find that this results in immature morphologies and altered gene expression. This study is one of many recent papers that document efficient generation of neurons using transcription factor over-expression. As such, the protocol described in this study largely confirms or extend principles already clarified by existing studies. The novelty in this study rather lies in the ambitious bioinformatical analysis of the reprogramming process. Over all, I have a sense of enthusiasm about this study. But I also have some critiques that I think are important.

We appreciate Reviewer #2's enthusiasm and review of our work. Subsequently, we address all criticism and input.

Major comments:

1. The authors use a bi-cistronic vector to overexpress Ngn1 and Ngn2. The reasons for choosing these two factors are not well described in the MS. Do these factors complement each other? Do you get the same effect with only Ngn1 or Ngn2?

We updated the introduction and motivation for using a bicistronic Neurogenin cassette in the text. Briefly, individual Neurogenins have been shown to induce neuronal differentiation but at lesser levels (see Farah *et al.*, 2000 Development, Reyes *et al.*, 2008 J Neuroscience) and we wondered if there was a beneficial effect on differentiation speed, yield and neuron type using a bicistronic approach. After the Zhang *et al.* publication, we also tested individual Neurogenins and both transcription factors also induced neurogenesis in human iPS cells (Figure E13). We profiled these samples using microarrays and compared them with the bicistronic version highlighting thousands (NEUROG1) and hundreds (NEUROG2) of differentially expressed genes. Hence we concluded that both Neurogenins complement each other resulting in iNGN cells. We addressed this aspect in the Expanded View Text and added these data to Figure E13.

2. The authors rightly cite Zhang *et al.*, 2013 which use a similar approach to generate neurons. It would be good if the authors would clarify the differences between their approach and the one used by Zhang *et al.*?

We updated the introduction to highlight the differences. For example, Zhang, *et al.* expressed Neurog2 with bioactive factors in the culturing media and glia co-culture techniques, while our bicistronic Neurog1+2 induction was done in defined stem cell media (mTeSR). This allowed us to assess the pure Neurogenin effects on inducing neurogenesis in stem cells. As Reviewer #2 has acknowledged, we aimed to understand these early reprogramming processes whereas all bioactive and neurotrophic factors either added as a supplement or secreted by astrocytes to the culturing media affect differentiation to large extent by themselves.

3. All data in the manuscript is based on a single iPS-line. It is necessary to document that the rapid reprogramming can be achieved on other iPS-lines as well as on human embryonic stem cells.

This indeed is an important point and so we have now introduced the bicistronic construct into another iPS cell line (PGP9) as well as into a human ES cell line (CHB8). Upon induction of the Neurogenins, these cells also differentiated to neurons within four days and were MAP2 immuno-positive (both cell lines >93%). Hence, the bicistronic Neurogenin cassette works independent of the human stem cell line used. We show this data in Figure E1G, H.

4. The inclusion of the miRNA data is interesting since it adds another level of complexity. The authors conclude that miRNAs are "modest shapers of neurogenesis rather than powerful inducers". This strong statement is based on bioinformatical analysis that suggests that targets of upregulated miRNAs are altered less than targets of transcription factors. However, for miR-124, which is the most highly expressed miRNA in their system, only one validated target is included in their analysis. Certainly, miR-124 has many more targets... Thus, in order to make such strong a conclusion additional experiments are necessary. A good start would be perturbation and overexpression of miR-124. It would also be good if the text were more balanced.

Actually, we were quite a bit disappointed about miRNA functions during iNGN cell development since according to recent progress in the field, we would have thought of major contributions of these non-coding RNAs. We totally agree with Reviewer #2 on experimental miRNA perturbations and therefore we knocked down miR-124 and the miR-302/367 cluster during iNGN differentiation by using miRNA sponges. In line with our conclusions from our transcriptomic analysis, we did not detect significant global (morphological) effects by these manipulations. However, we did see some significantly increased expression levels of a few validated miRNA targets, suggesting that the sponges themselves worked. We added the data to Figure E10.

Since miR-124 and the miR-302/367 cluster are the most highly expressed miRNA species, we would have expected more significant impacts upon knockdown according to previous knowledge. As a follow up experiment, instead of an antisense knockdown strategy, we plan to knock out all three genomic miR-124 loci in iNGN cells allowing us to study this miRNA during differentiation. Alternatively, one could think of knocking out the entire miRNA machinery in iNGN cells (DGCR8 knockout) and spike in miR-124 or other miRNAs of interest (see Busskamp *et al.*, Neuron 2014).

Reviewer #2 has made a good point here. Although we have strengthened our claim with the additional data, we feel it is better to word our conclusion more carefully in the text. Thus, we have eliminated our statement that miRNAs seem to be "modest shapers of neurogenesis rather than powerful inducers".

5. A major weakness in this study is that almost all analysis is on mRNA and miRNA despite the fact that the study is centered on TFs, which are proteins. At least some of the key findings in the study should be backed up with quantitative protein data (e.g. Western Blot).

In addition to the immunohistochemistry in our paper, we have now also included additional support of doxycycline-dependent protein induction from western blot data that we show in Figure E1.

Minor comments:

1. The title is too broad and vague

Indeed, the title is somewhat broad. The current title was based on suggestion from the editor during our previous submission. If the editor agrees with the need to change the title we would suggest one of the following and welcome further input.

The transcriptional response to rapid neurogenesis in human stem cells

Rapid neurogenesis through transcriptional activation in human stem cells

2. The authors should discuss how the choice of reprogramming factors may influence the generation of different subtypes of neurons. E.g. Ngn1/Ngn2 vs. Mash1.

In a very nice study by Wapinski *et al.* 2013, the authors showed that Mash1 acts as a pioneering factor enabling other neural transcription factors to bind. Mash1 (Ascl1) works quite efficient within a cocktail of transcription factors *in vitro* (together with Brn2 and Myt1l, so-called BAM factors, Pang *et al.* 2011) but also alone in human ES cells (Chanda *et al.* 2014). Based on single cell RNA profiling of 41 markers, these neurons could not be discriminated from the Neurog2 ones reported by Zhang *et al.* suggesting that both factors induce similar neuronal lineages and for detecting differences, one would need more sophisticated analyses.

In our revision, we have suggested the potential utility of leveraging other factors to possibly obtain additional neuron types. However, we have done so carefully to avoid speculation since detailed studies need to be conducted to make reliable comparisons. We also note that with the current data and published protocols, it is very difficult to predict the neuronal outcome by using specific reprogramming factors. For example, the cellular starting point (i.e., cell type, epigenetic states, etc.) varies a lot between *in vivo* and *in vitro* systems and often one cannot

directly project *in vivo* findings into *in vitro* systems. Also between *in vitro* systems, the protocols cannot easily be compared because of different cell lines, media conditions, bioactive supplements, co-culture techniques and different readouts (e.g., immunostainings, functional recordings, gene arrays versus RNA-Seq, qRT-PCR, etc.). Another issue is that some of the published protocols are easy to reproduce and more robust whereas others are more delicate and prone to variations in the induced neurons and yields. Furthermore, most protocols to date produce a mixture of neurons. Because of these and other complexities, we need higher resolution data before we can precisely predict the neuron type induced by individual or cocktails of transcription factors.

Overall, this study provides an ambitious effort to document the neuronal reprogramming process. It will be of interest to the field but the above critiques need to be addressed.

We would like to thank Reviewer #2 for his positive evaluation of our work. We hope that we have adequately addressed of the reviewer's concerns in the revised manuscript.

Reviewer #3:

The manuscript provides a systematic analysis of molecular mechanisms underlying conversion of a pluripotent cell to a nerve cell following inducible expression of NEUROG1 and NEUROG2 in human stem cells. While the identity and nature of resulting nerve cells remains ill-defined, possibly not reflecting any real cell in the nervous system, the derived networks might be useful to other researchers studying the processes of neuronal differentiation in more physiological contexts. The revised text now more clearly describes the methodology that was used to derive the genetic networks. The findings are consistent with current state of knowledge of the effects neurogenin on gene expression. The authors validated well characterized targets of neurogenins and demonstrate that in the inducible system these targets are required for proper execution of the neuronal differentiation. While the study does not reveal conceptually novel molecular mechanisms, its re-discovery of important players in neural differentiation indicates that the expression data sets accompanying the manuscript will be useful and hopefully generally applicable.

We would like to thank Reviewer #3 for his evaluation of our manuscript and acknowledgement of our previous revision.

Minor comments:

1. The authors should not conclude based on global gene expression data that resulting nerve cells express dual neurotransmitter phenotype. Simpler interpretation is that the protocol results in a mixture of nerve cells some of which are cholinergic and other are glutamatergic. To resolve this discrepancy one would have to perform single cell analysis.

This is a very good point and we also wondered about the neurotransmitter type of iNGN cells. We performed immunohistochemistry for vGLUT1 and ChAT to see whether iNGN cells were a mixture of glutamatergic and cholinergic cells or express both (see Figure 2F-H). We found that 100% of the iNGN cells were positive for vGLUT1 and 98% for ChAT. A microscopic single cell analysis revealed >96% co-localization of these two markers. Hence, we have great confidence that the iNGN cells form a homogeneous neuronal population.

2. Similarly, it is an extension to conclude based on the presented data that the cells follow normal developmental trajectory including neural progenitor stage. While several progenitor genes are induced it is unlikely that cells pass through neural plate and early neuroepithelium developmental stages. Indeed expression of the earliest and most general neural progenitor marker Sox1 is not induced during the transition.

We totally agree with Reviewer #3 that we cannot really precise the type of neuronal progenitor types and states. We have clarified this aspect in the revised discussion.